# On the Convergence of Prior-Guided Zeroth-Order Optimization Algorithms

**Shuyu Cheng,    Guoqiang Wu,    Jun Zhu**[*]

Dept. of Comp. Sci. and Tech., BNRist Center, State Key Lab for Intell. Tech. & Sys., Institute for AI,
Tsinghua-Bosch Joint Center for ML, Tsinghua University, Beijing, 100084, China
Pazhou Lab, Guangzhou, 510330, China

`chengsy18@mails.tsinghua.edu.cn, guoqiangwu90@gmail.com, dcszj@tsinghua.edu.cn`

## Abstract

Zeroth-order (ZO) optimization is widely used to handle challenging tasks, such as query-based black-box adversarial attacks and reinforcement learning. Various attempts have been made to integrate prior information into the gradient estimation procedure based on finite differences, with promising empirical results. However, their convergence properties are not well understood. This paper makes an attempt to fill up this gap by analyzing the convergence of prior-guided ZO algorithms under a greedy descent framework with various gradient estimators. We provide a convergence guarantee for the prior-guided random gradient-free (PRGF) algorithms. Moreover, to further accelerate over greedy descent methods, we present a new accelerated random search (ARS) algorithm that incorporates prior information, together with a convergence analysis. Finally, our theoretical results are confirmed by experiments on several numerical benchmarks as well as adversarial attacks. Our code is available at `https://github.com/csy530216/pg-zoo`.

## 1  Introduction

Zeroth-order (ZO) optimization [18] provides powerful tools to deal with challenging tasks, such as query-based black-box adversarial attacks [7, 13], reinforcement learning [22, 17, 9], meta-learning [1], and hyperparameter tuning [23], where the access to gradient information is either not available or too costly. ZO methods only assume an oracle access to *the function value* at any given point, instead of gradients as in first-order methods. The primary goal is to find a solution with as few queries to the function value oracle as possible. Recently, various ZO methods have been proposed in two main categories. One type is to obtain a gradient estimator and plug in some gradient-based methods. [21] analyzes the convergence of such methods with a random gradient-free (RGF) estimator obtained via finite difference along a random direction. The other type is directed search, where the update only depends on comparison between function values at different points. Such methods are robust against monotone transform of the objective function [24, 10, 3]. However, as they do not directly utilize function values, their query complexity is often higher than that of finite difference methods. Therefore, we focus on the first type of methods in this paper. Other methods exist such as CMA-ES [11] which is potentially better on objective functions with a rugged landscape, but lacks a general convergence guarantee.

ZO methods are usually less efficient than first-order algorithms, as they typically take $O(d)$ queries to reach a given precision, where $d$ is the input dimension (see Table 1 in [10]). Specifically, for the methods in [21], the oracle query count is $O(d)$ times larger than that of their corresponding schemes using gradients. This inefficiency stems from random search along uniformly distributed directions. To improve, various attempts have been made to augment random search with (extra)

---

[*]J.Z. is the Corresponding Author. G.W. is now with School of Software, Shandong University.

35th Conference on Neural Information Processing Systems (NeurIPS 2021).

prior information. For instance, [14, 19] use a time-dependent prior (i.e., the gradient estimated in the last iteration), while [16, 8, 4] use surrogate gradients obtained from other sources.[2] Among these methods, [14, 16, 8, 19] propose objective functions to describe the quality of a gradient estimator for justification or optimizing its hyperparameters; for example, [19] uses a subspace estimator that maximizes its squared cosine similarity with the true gradient as the objective, and finds a better descent direction than both the prior direction and the randomly sampled direction. However, all these methods treat gradient estimation and the optimization algorithm separately, and it remains unclear *whether are these gradient estimators and the corresponding prior-exploiting optimization methods theoretically sound?* and *what role does the prior play in accelerating convergence?*

In this paper, we attempt to answer these questions by establishing a formal connection between convergence rates and the quality of gradient estimates. Further, we develop a more efficient ZO algorithm with prior inspired by a theoretical analysis. First, we present a greedy descent framework of ZO methods and provide its convergence rate under smooth convex optimization, which is positively related to the squared cosine similarity between the gradient estimate and true gradient. As shown by [19] and our results, given some finite-difference queries, the optimal estimator maximizing the squared cosine similarity is the projection of true gradient on the subspace spanned by those queried directions. In the case with prior information, a natural such estimator is in the same form as the one in [19]. We call it PRGF estimator and analyze its convergence rate.[3] Our results show that no matter what the prior is, the convergence rate of PRGF is at least the same as that of the RGF baseline [21, 15], and it could be significantly improved if given a useful prior.[4] Such results shed light on exploring prior information in ZO optimization to accelerate convergence.

Then, as a concrete example, we apply the analysis to History-PRGF [19], which uses historical information (i.e., gradient estimate in the last iteration) as the prior. [19] presents an analysis on linear functions, yet still lacking a convergence rate. We analyze on general $L$-smooth functions, and find that when the learning rate is smaller than the optimal value $1/L$ in smooth optimization, the expected squared cosine similarity could converge to a larger value, which compensates for the slowdown of convergence brought by the inappropriate choice of learning rate. We also show that History-PRGF admits a convergence rate independent of learning rate as long as it is in a fairly wide range.

Finally, to further accelerate greedy descent methods, we present Prior-Guided ARS (PARS), a new variant of Accelerated Random Search (ARS) [21] to utilize prior information. Technically, PARS is a non-trivial extension of ARS, as directly replacing the gradient estimator in ARS by the PRGF estimator would lose the convergence analysis. Thus, we present necessary extensions to ARS, and show that PARS has a convergence rate no worse than that of ARS and admits potential acceleration given a good prior. In particular, when the prior is chosen as the historical information, the resulting History-PARS is robust to learning rate in experiments. To our knowledge, History-PARS is the first ARS-based method that is empirically robust while retaining the convergence rate as ARS. Our experiments on numerical benchmarks and adversarial attacks confirm the theoretical results.

## 2 Setups

**Assumptions on the problem class**   We consider unconstrained optimization, where the objective function $f : \mathbb{R}^d \rightarrow \mathbb{R}$ is convex and $L$-smooth for $L \geq 0$. Optionally, we require $f$ to be $\tau$-strongly convex for $\tau > 0$. We leave definitions of these concepts to Appendix A.1.

**Directional derivative oracle**   In ZO optimization, we follow the finite difference approach, which makes more use of the queried function values than direct search and provides better gradient approximations than alternatives [2]. In particular, we consider the forward difference method:

$$g_\mu(v; x) := \frac{f(x + \mu v) - f(x)}{\mu} \approx \nabla f(x)^\top v, \tag{1}$$

---

[2]Some work [7, 14, 25] restricts the random search to a more effective subspace reflecting the prior knowledge. But, this eliminates the possibility of convergence to the optimal solution.

[3]Note that the estimator is different from the P-RGF estimator in [8].

[4]In this paper, RGF and PRGF could refer to either a gradient estimator or the greedy descent algorithm with the corresponding estimator, depending on the context.

---

**Algorithm 1** Greedy descent framework

---

**Input:** $L$-smooth convex function $f$; initialization $x_0$; upper bound $\hat{L}$ ($\hat{L} \geq L$); iteration number $T$.
**Output:** $x_T$ as the approximate minimizer of $f$.
 1: **for** $t = 0$ to $T - 1$ **do**
 2:     Let $v_t$ be a random vector s.t. $\|v_t\| = 1$;
 3:     $x_{t+1} \leftarrow x_t - \frac{1}{\hat{L}} g_t$, where $g_t \leftarrow \nabla f(x_t)^\top v_t \cdot v_t$;
 4: **end for**
 5: **return** $x_T$.

---

where $v$ is a vector with unit $\ell_2$ norm $\|v\| = 1$ and $\mu$ is a small positive step. As long as the objective function is smooth, the error between the finite difference and the directional derivative could be uniformly bounded, as shown in the following proposition (see Appendix A.2 for its proof).

**Proposition 1.** *If $f$ is $L$-smooth, then for any $(x, v)$ with $\|v\| = 1$, $|g_\mu(v; x) - \nabla f(x)^\top v| \leq \frac{1}{2} L \mu$.*

Thus, in smooth optimization, the error brought by finite differences to the convergence bound can be analyzed in a principled way, and its impact tends to zero as $\mu \to 0$. We also choose $\mu$ as small as possible in practice. Hence, in the following analysis we directly assume the **directional derivative oracle**: suppose that we can obtain $\nabla f(x)^\top v$ for any $(x, v)$ in which $\|v\| = 1$ with one query.

## 3   Greedy descent framework and PRGF algorithm

We now introduce a greedy descent framework in ZO optimization which can be implemented with various gradient estimators. We first provide a general analysis, followed by a concrete example.

### 3.1   The greedy descent framework and general analysis

In first-order smooth convex optimization, a sufficient single-step decrease of the objective can guarantee convergence (see Chapter 3.2 in [5]). Inspired by this fact, we design the update in an iteration to greedily seek for maximum decrease. Suppose we are currently at $x$, and want to update along the direction $v$. Without loss of generality, assume $\|v\| = 1$ and $\nabla f(x)^\top v > 0$. To choose a suitable step size $\eta$ that minimizes $f(x - \eta v)$, we note that

$$f(x - \eta v) \leq f(x) - \eta \nabla f(x)^\top v + \frac{1}{2} L \eta^2 := F(\eta) \tag{2}$$

by smoothness of $f$. For the r.h.s, we have $F(\eta) = f(x) - \frac{(\nabla f(x)^\top v)^2}{2L}$ when $\eta = \frac{\nabla f(x)^\top v}{L}$, which minimizes $F(\eta)$. Thus, choosing such $\eta$ could lead to a largest guaranteed decrease of $f(x - \eta v)$ from $f(x)$. In practice, the value of $L$ is often unknown, but we can verify that as long as $0 < \eta \leq \frac{\nabla f(x)^\top v}{L}$, then $f(x - \eta v) \leq F(\eta) < f(x)$, i.e., we can guarantee decrease of the objective (regardless of the direction of $v$ if $\nabla f(x)^\top v > 0$). Based on the above discussion, we further allow $v$ to be random and present the greedy descent framework in Algorithm 1.

**Remark 1.** *If $v_t \sim \mathcal{U}(\mathbb{S}^{d-1})$, i.e. $v_t$ is uniformly sampled from $\mathbb{S}^{d-1}$ (the unit sphere in $\mathbb{R}^d$), then Algorithm 1 is similar to the simple random search in [21] except that $v_t$ is sampled from a Gaussian distribution there.*

**Remark 2.** *In general, $v_t$ could depend on the history (i.e., the randomness before sampling $v_t$). For example, $v_t$ can be biased towards a vector $p_t$ that corresponds to prior information, and $p_t$ depends on the history since it depends on $x_t$ or the optimization trajectory.*

*Theoretically speaking, $v_t$ is sampled from the conditional probability distribution $\Pr(\cdot | \mathcal{F}_{t-1})$ where $\mathcal{F}_{t-1}$ is a sub $\sigma$-algebra modelling the historical information. $\mathcal{F}_{t-1}$ is important in our theoretical analysis since it tells how to perform conditional expectation given the history.*

*We always require that $\mathcal{F}_{t-1}$ includes all the randomness before iteration $t$ to ensure that Lemma 1 (and thus Theorems 1 and 2) and Theorem 5 hold. For further theoretical analysis of various implementations of the framework, $\mathcal{F}_{t-1}$ remains to be specified by possibly also including some randomness in iteration $t$ (see e.g. Example 2 as an implementation of Algorithm 1).[5]*

---

[5] In mathematical language, if $\mathcal{F}_{t-1}$ includes (and only includes) the randomness brought by random vectors $\{x_1, x_2, \ldots, x_n\}$, then $\mathcal{F}_{t-1}$ is the $\sigma$-algebra generated by $\{x_1, x_2, \ldots, x_n\}$: $\mathcal{F}_{t-1}$ is the smallest $\sigma$-algebra s.t. $x_i$ is $\mathcal{F}_{t-1}$-measurable for all $1 \leq i \leq n$.

By Remark 2, we introduce $\mathbb{E}_t[\cdot] := \mathbb{E}[\cdot|\mathcal{F}_{t-1}]$ to denote the conditional expectation given the history. In Algorithm 1, under a suitable choice of $\mathcal{F}_{t-1}$, $\mathbb{E}_t[\cdot]$ roughly means only taking expectation w.r.t. $v_t$. We let $\overline{v} := v/\|v\|$ denote the $\ell_2$ normalization of vector $v$. We let $x^*$ denote one of the minimizers of $f$ (we assume such minimizer exists), and $\delta_t := f(x_t) - f(x^*)$. Thanks to the descent property in Algorithm 1, we have the following lemma on single step progress.

**Lemma 1** (Proof in Appendix B.1). *Let* $C_t := \left(\overline{\nabla f(x_t)}^\top v_t\right)^2$ *and* $L' := \frac{L}{1-(1-\frac{L}{\hat{L}})^2}$, *then in Algorithm 1, we have*

$$\mathbb{E}_t[\delta_{t+1}] \leq \delta_t - \frac{\mathbb{E}_t[C_t]}{2L'}\|\nabla f(x_t)\|^2. \tag{3}$$

We note that $L' = \hat{L}/(2-L/\hat{L})$, so $\frac{\hat{L}}{2} \leq L' \leq \hat{L}$ and $L' \geq L$.

To obtain a bound on $\mathbb{E}[\delta_T]$, one of the classical proofs (see e.g., Theorem 1 in [20]) requires us to take expectation on both sides of Eq. (3). Allowing the distribution of $v_t$ to be dependent on the history leads to additional technical difficulty: in the r.h.s of Eq. (3), $\mathbb{E}_t[C_t]$ becomes a random variable that is not independent of $\|\nabla f(x_t)\|^2$. Thus, we cannot simplify the term $\mathbb{E}[\mathbb{E}_t[C_t]\|\nabla f(x_t)\|^2]$ if we take expectation.[6] By using other techniques, we obtain the following main theorems on convergence rate.

**Theorem 1** (Algorithm 1, smooth and convex; proof in Appendix B.1). *Let* $R := \max_{x:f(x)\leq f(x_0)} \|x - x^*\|$ *and suppose* $R < \infty$. *Then, in Algorithm 1, we have*

$$\mathbb{E}[\delta_T] \leq \frac{2L'R^2 \sum_{t=0}^{T-1} \mathbb{E}\left[\frac{1}{\mathbb{E}_t[C_t]}\right]}{T(T+1)}. \tag{4}$$

**Theorem 2** (Algorithm 1, smooth and strongly convex; proof in Appendix B.1). *If $f$ is also $\tau$-strongly convex, then we have*

$$\mathbb{E}\left[\frac{\delta_T}{\exp\left(-\frac{\tau}{L'}\sum_{t=0}^{T-1}\mathbb{E}_t[C_t]\right)}\right] \leq \delta_0. \tag{5}$$

**Remark 3.** *In our results, the convergence rate depends on $\mathbb{E}_t[C_t]$ in a more complicated way than only depending on $\mathbb{E}[C_t] = \mathbb{E}[\mathbb{E}_t[C_t]]$. For concrete cases, one may need to study the concentration properties of $\mathbb{E}_t[C_t]$ besides its expectation, as in the proofs of Theorems 3 and 4 when we analyze History-PRGF, a special implementation in the greedy descent framework.*

*In the strongly convex case, currently we cannot directly obtain a final bound of $\mathbb{E}[\delta_T]$. However, the form of Eq. (5) is still useful since it roughly tells us that $\delta_T$ converges as the denominator $\exp\left(-\frac{\tau}{L'}\sum_{t=0}^{T-1}\mathbb{E}_t[C_t]\right)$. For History-PRGF, we will turn this intuition into a rigorous theorem (Theorem 4) since in that case we can prove that the denominator has a nice concentration property.*

**Remark 4.** *If we have a lower bound of $\mathbb{E}_t[C_t]$, e.g. $\mathbb{E}_t[C_t] \geq a > 0$, then we directly obtain that $\mathbb{E}[\delta_T] \leq \frac{2L'R^2}{a(T+1)}$ for Theorem 1, and $\mathbb{E}[\delta_T] \leq \delta_0 \exp(-\frac{\tau}{L'}aT)$ for Theorem 2. These could recover the ideal convergence rate for RGF estimator and the worst-case convergence rate for PRGF estimator, as explained in Examples 1 and 2.*

From Theorems 1 and 2, we see that a larger value of $C_t$ would lead to a better bound. To find a good choice of $v_t$ in Algorithm 1, it becomes natural to discuss the following problem. Suppose in an iteration in Algorithm 1, we query the directional derivative oracle at $x_t$ along $q$ directions $\{u_i\}_{i=1}^q$ (maybe randomly chosen) and obtain the values of $\{\nabla f(x_t)^\top u_i\}_{i=1}^q$. We could use this information to construct a vector $v_t$. What is the $v_t$ that maximizes $C_t$ s.t. $\|v_t\| = 1$? To answer this question, we give the following proposition based on Proposition 1 in [19] and additional justification.

**Proposition 2** (Optimality of subspace estimator; proof in Appendix B.2). *In one iteration of Algorithm 1, if we have queried $\{\nabla f(x_t)^\top u_i\}_{i=1}^q$, then the optimal $v_t$ maximizing $C_t$ s.t. $\|v_t\| = 1$ should be in the following form: $v_t = \overline{\nabla f(x_t)_A}$, where $A := \mathrm{span}\{u_1, u_2, \ldots, u_q\}$ and $\nabla f(x_t)_A$ denotes the projection of $\nabla f(x_t)$ onto $A$.*

---

[6]It is also the difficulty we faced in our very preliminary attempts of the theoretical analysis when $f$ is not convex, and we leave its solution or workaround in the non-convex case as future work.

Note that in Line 3 of Algorithm 1, we have $g_t = \nabla f(x_t)^\top \overline{\nabla f(x_t)_A} \cdot \overline{\nabla f(x_t)_A} = \nabla f(x_t)_A$. Therefore, the gradient estimator $g_t$ is equivalent to the projection of the gradient to the subspace $A$, which justifies its name of subspace estimator. Next we discuss some special cases of the subspace estimator. We leave detailed derivation in following examples to Appendix B.3.

**Example 1** (RGF). *$u_i \sim \mathcal{U}(\mathbb{S}^{d-1})$ for $i = 1, 2, \ldots, q$. Without loss of generality, we assume they are orthonormal (e.g., via Gram-Schmidt orthogonalization).[7] The corresponding estimator $g_t = \sum_{i=1}^q \nabla f(x_t)^\top u_i \cdot u_i$ ($v_t = \overline{g_t}$). When $q = 1$, the estimator is similar to the random gradient-free oracle in [21]. With $q \geq 1$, it is essentially the same as the stochastic subspace estimator with columns from Haar-distributed random orthogonal matrix [15] and similar to the orthogonal ES estimator [9]. In theoretical analysis, we let $\mathcal{F}_{t-1}$ only include the randomness before iteration $t$, and then we can prove that $\mathbb{E}_t[C_t] = \frac{q}{d}$. By Theorems 1 and 2, the convergence rate is $\mathbb{E}[\delta_T] \leq \frac{2L'R^2 \frac{d}{q}}{T+1}$ for smooth convex case, and $\mathbb{E}[\delta_T] \leq \delta_0 \exp(-\frac{\tau}{L'}\frac{q}{d}T)$ for smooth and strongly convex case. The bound is the same as that in [15]. Since the query complexity in each iteration is proportional to $q$, the bound for total query complexity is indeed independent of $q$.*

**Example 2** (PRGF). *With slight notation abuse, we assume the subspace in Proposition 2 is spanned by $\{p_1, \ldots, p_k, u_1, \ldots, u_q\}$, so each iteration takes $q + k$ queries. Let $p_1, \cdots, p_k$ be $k$ non-zero vectors corresponding to the prior message (e.g. the historical update, or the gradient of a surrogate model), and $u_i \sim \mathcal{U}(\mathbb{S}^{d-1})$ for $i = 1, 2, \ldots, q$. We note that intuitively we cannot construct a better subspace since the only extra information we know is the $k$ priors. In our analysis, we assume $k = 1$ for convenience, and we change the original notation $p_1$ to $p_t$ to explicitly show the dependence of $p_t$ on the history. We note that $p_t$ could also depend on extra randomness in iteration $t$ (see e.g. the specification of $p_t$ in Appendix D.1.1). For convenience of theoretical analysis, we require that $p_t$ is determined before sampling $\{u_1, u_2, \ldots, u_q\}$, and let $\mathcal{F}_{t-1}$ also include the extra randomness of $p_t$ in iteration $t$ (not including the randomness of $\{u_1, u_2, \ldots, u_q\}$) besides the randomness before iteration $t$. Then $p_t$ is always $\mathcal{F}_{t-1}$-measurable, i.e. determined by the history. Without loss of generality, we assume $\{p_t, u_1, \ldots, u_q\}$ are orthonormal (e.g., via Gram-Schmidt orthogonalization). The corresponding estimator $g_t = \nabla f(x_t)^\top p_t \cdot p_t + \sum_{i=1}^q \nabla f(x_t)^\top u_i \cdot u_i$ ($v_t = \overline{g_t}$), which is similar to the estimator in [19]. By [19] (the expected drift of $X_t^2$ in its Theorem 1), we have*

**Lemma 2** (Proof in Appendix B.3.4). *In Algorithm 1 with PRGF estimator, $\mathbb{E}_t[C_t] = D_t + \frac{q}{d-1}(1 - D_t)$ where $D_t := \left(\overline{\nabla f(x_t)}^\top p_t\right)^2$.*

*Hence $\mathbb{E}_t[C_t] \geq \frac{q}{d}$ holds. By Remark 4, PRGF admits a guaranteed convergence rate of RGF and is potentially better given a good prior (if $D_t$ is large), but it costs an additional query per iteration. This shows soundness of the PRGF algorithm. For further theoretical analysis, we need to bound $D_t$. This could be done when using the historical prior introduced in Section 3.2 (see Lemma 3). Bounding $D_t$ is usually challenging when a general prior is adopted, but if the prior is an approximate gradient (such case appears in [16]), it may be possible. We leave related investigation as future work.*

### 3.2 Analysis on the PRGF algorithm with the historical prior

We apply the above analysis to a concrete example of the History-PRGF estimator [19], which considers the historical prior in the PRGF estimator. In this case, Lemma 1, Theorem 1 and Theorem 2 will manifest themselves by clearly stating the convergence rate which is robust to the learning rate.

Specifically, History-PRGF considers the historical prior as follows: we choose $p_t = \overline{g_{t-1}}$, i.e., we let the prior be the direction of the previous gradient estimate.[8] This is equivalent to letting $p_t = v_{t-1}$. Thus, in Algorithm 1, $v_t = \overline{\nabla f(x_t)_A} = \overline{\nabla f(x_t)^\top v_{t-1} \cdot v_{t-1} + \sum_{i=1}^q \nabla f(x_t)^\top u_i \cdot u_i}$. In this form we require $\{v_{t-1}, u_1, \ldots, u_q\}$ to be orthonormal, so we first determine $v_{t-1}$, and then

---

[7]The computational complexity of Gram-Schmidt orthogonalization over $q$ vectors in $\mathbb{R}^d$ is $O(q^2 d)$. Therefore, with a moderate value of $q$ (e.g. $q \in [10, 20]$ in our experiments), its cost is usually much smaller than that brought by $O(q)$ function evaluations used to approximate the directional derivatives. We note that when using a numerical computing framework, for orthogonalization one could also adopt an efficient implementation, by calling the QR decomposition procedure such as `torch.linalg.qr` in PyTorch.

[8]One can also utilize multiple historical priors (e.g. the last $k$ updates with $k > 1$, as proposed and experimented in [19]), but here we only analyze the $k = 1$ case.

sample $\{u_i\}_{i=1}^q$ in $A_\perp$, the $(d-1)$-dimensional subspace of $\mathbb{R}^d$ perpendicular to $v_{t-1}$, and then do Gram-Schmidt orthonormalization on $\{u_i\}_{i=1}^q$.

To study the convergence rate, we first study evolution of $C_t$ under a general $L$-smooth function. This extends the analysis on linear functions (corresponding to $L = 0$) in [19]. Under the framework of Algorithm 1, intuitively, the change of the gradient should be smaller when the objective function is very smooth ($L$ is small) or the learning rate is small ($\hat{L}$ is large). Since we care about the cosine similarity between the gradient and the prior, we prove the following lemma:

**Lemma 3** (Proof in Appendix B.4). *In History-PRGF ($p_t = v_{t-1}$), we have $D_t \geq (1 - {^L}\!/\!_{\hat{L}})^2 C_{t-1}$.*

When $\hat{L} = L$, i.e., using the optimal learning rate, Lemma 3 does not provide a useful bound, since an optimal learning rate in smooth optimization could find an approximate minimizer along the update direction, so the update direction may be useless in next iteration. In this case, the historical prior does not provide acceleration. Hence, Lemma 3 explains the empirical findings in [19] that past directions can be less useful when the learning rate is larger. However, in practice we often use a conservative learning rate, for the following reasons: 1) We usually do not know $L$, and the cost of tuning learning rate could be large; 2) Even if $L$ is known, it only provides a global bound, so a fixed learning rate could be too conservative in the smoother local regions. In scenarios where $\hat{L}$ is too conservative ($\hat{L} > L$), History-PRGF could bring more acceleration over RGF.

By Lemma 3, we can assume $D_t = (1 - {^L}\!/\!_{\hat{L}})^2 C_{t-1}$ to obtain a lower bound of quantities about $C_t$. Meanwhile, since $D_t$ means quality of the prior, the construction in Example 2 tells us relationship between $C_t$ and $D_t$. Then we have full knowledge of the evolution of $C_t$, and thus $\mathbb{E}_t[C_t]$. In Appendix B.4, we discuss about evolution of $\mathbb{E}[C_t]$ and show that $\mathbb{E}[C_t] \to O(\frac{q}{d}\frac{L'}{L})$ if $\frac{q}{d} \leq \frac{L}{\hat{L}} \leq 1$. Therefore, by Lemma 1, assuming $\mathbb{E}_t[C_t]$ concentrates well around $\mathbb{E}[C_t]$ and hence $\frac{\mathbb{E}_t[C_t]}{L'} \approx \frac{q}{dL}$, PRGF could recover the single step progress with optimal learning rate ($\hat{L} = L$), since Eq. (3) only depends on $\frac{\mathbb{E}_t[C_t]}{L'}$ which is constant w.r.t. $L'$ now. While the above discussion is informal, based on Theorems 1 and 2, we prove following theorems which show that convergence rate of History-PRGF is robust to choice of learning rate.

**Theorem 3** (History-PRGF, smooth and convex; proof in Appendix B.5.1). *In the setting of Theorem 1, assuming $d \geq 4$, $\frac{q}{d-1} \leq \frac{L}{\hat{L}} \leq 1$ and $T > \left\lceil \frac{d}{q} \right\rceil$ ($\lceil \cdot \rceil$ denotes the ceiling function), we have*

$$\mathbb{E}[\delta_T] \leq \left( \frac{32}{q} + 2 \right) \frac{2L\frac{d}{q}R^2}{T - \left\lceil \frac{d}{q} \right\rceil + 1}. \tag{6}$$

*Sketch of the proof.* The idea of the proof of Theorem 3 is to show that for the random variable $\mathbb{E}_t[C_t]$, its standard deviation $\sqrt{\mathrm{Var}[\mathbb{E}_t[C_t]]}$ is small relative to its expectation $\mathbb{E}[\mathbb{E}_t[C_t]] = \mathbb{E}[C_t]$. By Chebyshev's inequality, we can bound $\mathbb{E}\left[\frac{1}{\mathbb{E}_t[C_t]}\right]$ in Theorem 1 with $\frac{1}{\mathbb{E}[\mathbb{E}_t[C_t]]} = \frac{1}{\mathbb{E}[C_t]}$. In the actual proof we replace $C_t$ that appears above with a lower bound $E_t$.

**Theorem 4** (History-PRGF, smooth and strongly convex; proof in Appendix B.5.3). *Under the same conditions as in Theorem 2, then assuming $d \geq 4$, $\frac{q}{d-1} \leq \frac{L}{\hat{L}} \leq 1$, $\frac{q}{d} \leq 0.2\frac{L}{\tau}$, and $T \geq 5\frac{d}{q}$, we have*

$$\mathbb{E}[\delta_T] \leq 2 \exp \left( -0.1\frac{q}{d}\frac{\tau}{L}T \right) \delta_0. \tag{7}$$

This result seems somewhat surprising since Theorem 2 does not directly give a bound of $\mathbb{E}[\delta_T]$.

*Sketch of the proof.* The goal is to show that the denominator in the l.h.s of Eq. (5) in Theorem 2, $\exp(-\frac{\tau}{L}\sum_{t=0}^{T-1}\mathbb{E}_t[C_t])$, concentrates very well. Indeed, the probability that $\exp(-\frac{\tau}{L}\sum_{t=0}^{T-1}\mathbb{E}_t[C_t])$ is larger than $\exp(-0.1\frac{q}{d}\frac{\tau}{L}T)$ is very small so that its influence can be bounded by another $\exp(-0.1\frac{q}{d}\frac{\tau}{L}T)\delta_0$, leading to the coefficient 2 in Eq. (7). In our actual analysis we replace $C_t$ that appears above with a lower bound $E_t$.

**Remark 5.** *As stated in Example 2, using RGF with the optimal learning rate, we have $\mathbb{E}[\delta_T] \leq \frac{2L\frac{d}{q}R^2}{T+1}$ for smooth and convex case, and $\mathbb{E}[\delta_T] \leq \exp\left(-\frac{q}{d}\frac{\tau}{L}T\right)\delta_0$ for smooth and strongly convex*

*case. Therefore, History-PRGF with a suboptimal learning rate $\frac{1}{\hat{L}}$ under the condition $\frac{q}{d-1}\frac{1}{L} \leq \frac{1}{\hat{L}} \leq \frac{1}{L}$ could reach similar convergence rate to RGF with optimal learning rate (up to constant factors), which indicates that History-PRGF is more robust to learning rate than RGF.*

**Remark 6.** *We note that the constants in the bounds are loose and have a large potential to be improved in future work, and empirically the convergence rate of History-PRGF is often not worse than RGF using the optimal learning rate (see Fig. 2).*

As a sidenote, we discuss how to set $q$ in History-PRGF. The iteration complexity given by Theorems 3 and 4 is proportional to $\frac{1}{q}$ if we ignore the constants such as $\frac{32}{q} + 2$ in Eq. (6) by Remark 6. Meanwhile, we recall that each iteration of PRGF requires $q + 1$ queries to the directional derivative oracle, so the total query complexity is roughly proportional to $\frac{q+1}{q}$. Hence, a very small $q$ (e.g. $q = 1$) is suboptimal. On the other hand, Theorems 3 and 4 require $\frac{1}{\hat{L}} \in \left[\frac{q}{d-1}\frac{1}{L}, \frac{1}{L}\right]$, so to enable robustness of History-PRGF to a wider range of the choice of learning rate, $q$ should not be too large. In summary, it is desirable to set $q$ to a moderate value.

Note that if we adopt line search in Algorithm 1, then one can adapt the learning rate in a huge range and reach the convergence guarantee with the optimal learning rate under weak assumptions. Nevertheless, it is still an intriguing fact that History-PRGF could perform similarly to methods adapting the learning rate, while its mechanism is very different. Meanwhile, History-PRGF is easier to be implemented and parallelized compared with methods like line search, since its implementation is the same as that of the RGF baseline except that it records and uses the historical prior.

## 4   Extension of ARS framework and PARS algorithm

To further accelerate greedy descent methods, we extend our analysis to a new variant of Accelerated Random Search (ARS) [21] by incorporating prior information, under the smooth and convex setting.[9]

By delving into the proof in [21], we present our extension to ARS in Algorithm 2, state its convergence guarantee in Theorem 5 and explain its design in the proof sketch in Appendix C.1.

---

**Algorithm 2** Extended accelerated random search framework[10]

---

**Input:** $L$-smooth convex function $f$; initialization $x_0$; $\hat{L} \geq L$; iteration number $T$; $\gamma_0 > 0$.
**Output:** $x_T$ as the approximate minimizer of $f$.
1: $m_0 \leftarrow x_0$;
2: **for** $t = 0$ to $T - 1$ **do**
3:     Find a $\theta_t > 0$ such that $\theta_t \leq \frac{\mathbb{E}_t\left[\left(\nabla f(y_t)^\top v_t\right)^2\right]}{\hat{L} \cdot \mathbb{E}_t[\|g_2(y_t)\|^2]}$ where $y_t$, $v_t$ and $g_2(y_t)$ are defined in following steps:
4:     Step 1: $y_t \leftarrow (1 - \alpha_t)x_t + \alpha_t m_t$, where $\alpha_t$ is a positive root of the equation $\alpha_t^2 = \theta_t(1 - \alpha_t)\gamma_t$; $\gamma_{t+1} \leftarrow (1 - \alpha_t)\gamma_t$;
5:     Step 2: Let $v_t$ be a random vector s.t. $\|v_t\| = 1$; $g_1(y_t) \leftarrow \nabla f(y_t)^\top v_t \cdot v_t$;
6:     Step 3: Let $g_2(y_t)$ be an unbiased estimator of $\nabla f(y_t)$, i.e., $\mathbb{E}_t[g_2(y_t)] = \nabla f(y_t)$;
7:     $x_{t+1} \leftarrow y_t - \frac{1}{\hat{L}}g_1(y_t), m_{t+1} \leftarrow m_t - \frac{\theta_t}{\alpha_t}g_2(y_t)$;
8: **end for**
9: **return** $x_T$.

---

**Theorem 5** (Proof in Appendix C.1). *In Algorithm 2, if $\theta_t$ is $\mathcal{F}_{t-1}$-measurable (see Appendix C.1 for more explanation), then we have*

$$\mathbb{E}\left[(f(x_T) - f(x^*))\left(1 + \frac{\sqrt{\gamma_0}}{2}\sum_{t=0}^{T-1}\sqrt{\theta_t}\right)^2\right] \leq f(x_0) - f(x^*) + \frac{\gamma_0}{2}\|x_0 - x^*\|^2. \quad (8)$$

---

[9]The procedure of ARS requires knowledge of the strong convexity parameter $\tau$ ($\tau$ can be 0), but for clarity we only discuss the case $\tau = 0$ here (i.e., we do not consider strong convexity), and leave the strongly convex case to Appendix C.6.

[10]Keys in this extension are: 1) We require $\mathbb{E}_t[g_2(y_t)] = \nabla f(y_t)$. Thus, $g_2(y_t)$ could not be the PRGF estimator as it is biased towards the prior; 2) To accelerate convergence, we need to find an appropriate $\theta_t$ since it appears in Eq. (8). If we set $\theta_t$ to the value in ARS baseline, then no potential acceleration is guaranteed.

**Remark 7.** *If we let $g_1(y_t)$ be the RGF estimator in Example 1 and let $g_2(y_t) = d/q \cdot g_1(y_t)$, we can show that $\mathbb{E}[g_2(y_t)] = \nabla f(y_t)$ and $\theta_t$ could be chosen as $\frac{q^2}{\hat{L}d^2}$ since $\frac{\mathbb{E}_t\left[\left(\nabla f(y_t)^\top v_t\right)^2\right]}{\hat{L}\cdot\mathbb{E}_t[\|g_2(y_t)\|^2]} = \frac{q^2}{\hat{L}d^2}$. Then roughly, the convergence rate $\propto q$, so the total query complexity is independent of $q$. When $q = 1$, ARS baseline is recovered. For convenience we call the algorithm ARS regardless of the value of $q$.*

**Remark 8.** *If we have a uniform constant lower bound $\theta > 0$ such that $\forall t, \theta_t \geq \theta$, then we have*

$$\mathbb{E}\left[f(x_T) - f(x^*)\right] \leq \left(1 + \frac{\sqrt{\gamma_0}}{2}T\sqrt{\theta}\right)^{-2}\left(f(x_0) - f(x^*) + \frac{\gamma_0}{2}\|x_0 - x^*\|^2\right). \tag{9}$$

Next we present Prior-Guided ARS (PARS) by specifying the choice of $g_1(y_t)$ and $g_2(y_t)$ in Algorithm 2 when prior information $p_t \in \mathbb{R}^d$ is available. Since we want to maximize the value of $\theta_t$, regarding $g_1(y_t)$ we want to maximize $\mathbb{E}_t\left[\left(\nabla f(y_t)^\top v_t\right)^2\right]$. By Proposition 2 and Example 2, it is natural to let $g_1(y_t)$ be the PRGF estimator for $\nabla f(y_t)$. Then by Lemma 2, we have $\mathbb{E}_t\left[\left(\nabla f(y_t)^\top v_t\right)^2\right] = \|\nabla f(y_t)\|^2(D_t + \frac{q}{d-1}(1 - D_t))$, where $D_t := \left(\overline{\nabla f(y_t)}^\top p_t\right)^2$. The remaining problem is to construct $g_2(y_t)$, an unbiased estimator of $\nabla f(y_t)$ ($\mathbb{E}_t[g_2(y_t)] = \nabla f(y_t)$), and make $\mathbb{E}_t[\|g_2(y_t)\|^2]$ as small as possible. We leave the construction of $g_2(y_t)$ in Appendix C.2. Finally, we calculate the following expression which appears in Line 3 of Algorithm 2 to complete the description of PARS:

$$\frac{\mathbb{E}_t\left[\left(\nabla f(y_t)^\top v_t\right)^2\right]}{\hat{L}\cdot\mathbb{E}_t[\|g_2(y_t)\|^2]} = \frac{D_t + \frac{q}{d-1}(1 - D_t)}{\hat{L}\left(D_t + \frac{d-1}{q}(1 - D_t)\right)}. \tag{10}$$

Since $D_t \geq 0$, the right-hand side is larger than $q^2/\hat{L}d^2$ (by Remark 7, this value corresponds to the value of $\theta_t$ in ARS), so by Remark 8 PARS is guaranteed a convergence rate of ARS.

In implementation of PARS, we note that there remain two problems to solve. The first is that $D_t$ is not accessible through one oracle query, since $D_t = \left(\overline{\nabla f(y_t)}^\top p_t\right)^2 = \left(\nabla f(y_t)^\top p_t/\|\nabla f(y_t)\|\right)^2$, and $\|\nabla f(y_t)\|$ requires estimation. Fortunately, the queries used to construct $g_1(y_t)$ and $g_2(y_t)$ can also be used to estimate $D_t$. With a moderate value of $q$, we can prove that considering the error brought by estimation of $D_t$, a modified version of PARS is guaranteed to converge with high probability. We leave related discussion to Appendix C.3. The second is that Line 3 of Algorithm 2 has a subtlety that $y_t$ depends on $\theta_t$, so we cannot directly determine an optimal $\theta_t$ satisfying $\theta_t \leq \frac{\mathbb{E}_t\left[\left(\nabla f(y_t)^\top v_t\right)^2\right]}{\hat{L}\cdot\mathbb{E}_t[\|g_2(y_t)\|^2]}$. Theoretically, we can guess a conservative estimate of $\theta_t$ and verify this inequality, but in practice we adopt a more aggressive strategy to find an approximate solution of $\theta_t$. We leave the actual implementation, named PARS-Impl, in Appendix C.4.

In PARS, if we adopt the historical prior as in Section 3.2, i.e., letting $p_t$ be the previous PRGF gradient estimator $\overline{g_1(y_{t-1})}$, then we arrive at a novel algorithm named History-PARS. Here, we note that unlike the case in History-PRGF, it is more difficult to derive the evolution of $\theta_t$ theoretically, so we currently cannot prove theorems corresponding to Theorem 3. However, History-PARS can be guaranteed the convergence rate of ARS, which is desirable since if we adopt line search in ARS to reach robustness against learning rate (e.g. in [24]), currently there is no convergence guarantee. We present the actual implementation History-PARS-Impl in Appendix C.5 and empirically verify that History-PARS-Impl is robust to learning rate in Section 5.

## 5 Experiments

### 5.1 Numerical benchmarks

We first experiment on several closed-form test functions to support our theoretical claims. We leave more details of experimental settings to Appendix D.1.

First, we present experimental results when a general useful prior is provided. The prior-guided methods include PRGF, PARS (refers to PARS-Impl) and PARS-Naive (simply replacing the RGF estimator in ARS with the PRGF estimator). We adopt the setting in Section 4.1 of [16] in which the

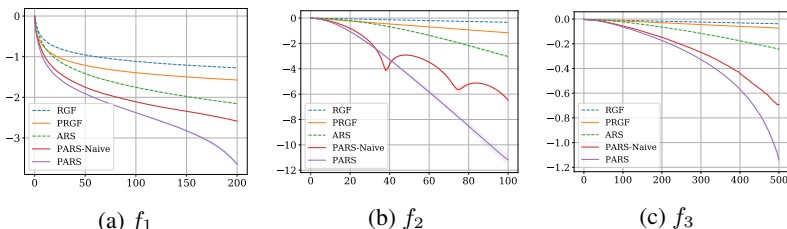

$$(a)\ f_1 \qquad\qquad (b)\ f_2 \qquad\qquad (c)\ f_3$$

Figure 1: Experimental results using biased gradient as the prior (best viewed in color).

prior is a biased version of the true gradient. Our test functions are as follows: 1) $f_1$ is the "worst-case smooth convex function" used to construct the lower bound complexity of first-order optimization, as in [21]; 2) $f_2$ is a simple smooth and strongly convex function with a worst-case initialization: $f_2(x) = \frac{1}{d} \sum_{i=1}^{d} \left( i \cdot (x^{(i)})^2 \right)$, where $x_0^{(1)} = d, x_0^{(i)} = 0$ for $i \geq 2$; and 3) $f_3$ is the Rosenbrock function ($f_8$ in [12]) which is a well-known non-convex function used to test the performance of optimization problems. For $f_1$ and $f_2$, we set $\hat{L}$ to ground truth value $L$; for $f_3$, we search $\hat{L}$ for best performance for each algorithm. We set $d = 256$ for all test functions and set $q$ such that each iteration of these algorithms costs 11 queries[11] to the directional derivative oracle.[12] We plot the experimental results in Fig. 1, where the horizontal axis represents the number of iterations divided by $\lfloor d/11 \rfloor$, and the vertical axis represents $\log_{10} \frac{f(x_{\text{current}}) - f(x^*)}{f(x_0) - f(x^*)}$. Methods without using the prior information are shown with dashed lines. We also plot the 95% confidence interval in the colored region. The results show that for these functions (which have ill-conditioned Hessians), ARS-based methods perform better than the methods based on greedy descent. Importantly, the utilization of the prior could significantly accelerate convergence for both greedy descent and ARS. We note that the performance of our proposed PARS algorithm is better than PARS-Naive which naively replaces the gradient estimator in the original ARS with the PRGF estimator, demonstrating the value of our algorithm design with convergence analysis.

Next, we verify the properties of History-PRGF and History-PARS, i.e., the historical-prior-guided algorithms. In this part we set $d = 500$. We first verify that they are robust against learning rate on $f_1$ and $f_2$, and plot the results in Fig. 2(a)(b).[13] In the legend, for example, 'RGF' means RGF using the optimal learning rate ($\hat{L} = L$), and 'RGF-0.02' means that the learning rate is set to 0.02 times of the optimal one ($\hat{L} = 50L$). We note that for PRGF and PARS, $q = 10$, so $\frac{q}{d} = 0.02$. From Fig. 2(a)(b), we see that: 1) when using the optimal learning rate, the performance of prior-guided algorithms is not worse than that of its corresponding baseline; and 2) the performance of prior-guided algorithms under the sub-optimal learning rate such that $\frac{q}{d} \leq \frac{L}{\hat{L}} \leq 1$ is at least comparable to that of its corresponding baseline with optimal learning rate. However, for baseline algorithms (RGF and ARS), the convergence rate significantly degrades if a smaller learning rate is set. In summary, we verify our claims that History-PRGF and History-PARS are robust to learning rate if $\frac{q}{d} \leq \frac{L}{\hat{L}} \leq 1$. Moreover, we show that they can provide acceleration over baselines with optimal learning rate on functions with varying local smoothness. We design a new test function as follows:

$$f_4(x) = \begin{cases} \frac{1}{2} r^2, & r \leq 1 \\ r - \frac{1}{2}, & r > 1 \end{cases}, r = \sqrt{f_2(x)}, \text{ where } x_0^{(1)} = 5\sqrt{d}, x_0^{(i)} = 0 \text{ for } i \geq 2. \qquad (11)$$

We note that $f_4$ in regions far away from the origin is more smooth than in the region near the origin, and the global smoothness parameter is determined by the worst-case situation (the region near the origin). Therefore, baseline methods using an optimal learning rate could also manifest sub-optimal performance. Fig. 2(c) shows the results. We can see that when utilizing the historical prior, the algorithm could show behaviors of adapting to the local smoothness, thus accelerating convergence when the learning rate is locally too conservative.

---

[11]That is, for prior-guided algorithms we set $q = 10$, and for other algorithms (RGF and ARS) we set $q = 11$.

[12]The directional derivative is approximated by finite differences. In PARS, 2 additional queries to the directional derivative oracle per iteration are required to find $\theta_t$ (see Appendix C.4).

[13]In Fig. 2, the setting of ARS-based methods are different from that in Fig. 1 as explained in Appendix D.1, which leads to many differences of the ARS curves between Fig. 1 and Fig. 2.

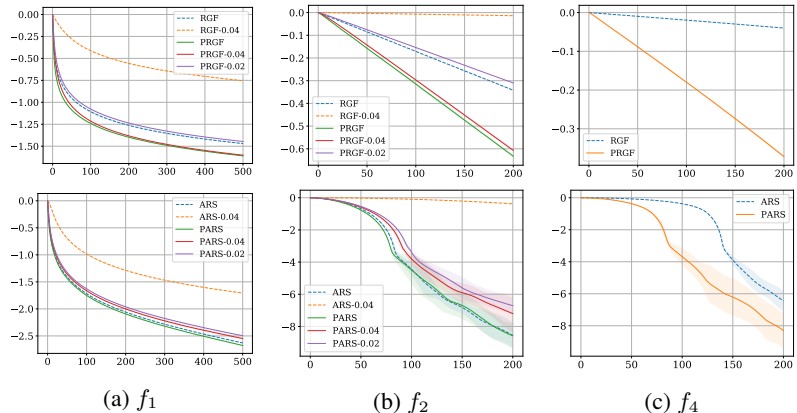

<p style="text-align:center">(a) $f_1$        (b) $f_2$        (c) $f_4$</p>

Figure 2: Experimental results using the historical prior.

## 5.2 Black-box adversarial attacks

In this section, we perform ZO optimization on real-world problems. We conduct score-based black-box targeted adversarial attacks on 500 images from MNIST and leave more details of experimental settings to Appendix D.2. In view of optimization, this corresponds to performing constrained maximization over $\{f_i\}_{i=1}^{500}$ respectively, where $f_i$ denotes the loss function to maximize in attacks. For each image $i$, we record the number of queries of $f_i$ used in optimization until the attack succeeds (when using the C&W loss function [6], this means $f_i > 0$). For each optimization method, we report the median query number over these images (smaller is better) in Table 1. The subscript of the method name indicates the learning rate $1/\hat{L}$. For all methods we set $q$ to 20. Since [8] has shown that using PRGF estimator with transfer-based prior significantly outperforms using RGF estimator in adversarial attacks, for prior-guided algorithms here we only include the historical prior case.

Table 1: Attack results on MNIST.

| METHOD | MEDIAN QUERY | METHOD | MEDIAN QUERY |
|---|---|---|---|
| $\text{RGF}_{0.2}$ | 777 | $\text{ARS}_{0.2}$ | 735 |
| $\text{RGF}_{0.1}$ | 1596 | $\text{ARS}_{0.1}$ | 1386 |
| $\text{History-PRGF}_{0.2}$ | 484 | $\text{History-PARS}_{0.2}$ | 484 |
| $\text{History-PRGF}_{0.1}$ | 572 | $\text{History-PARS}_{0.1}$ | 550 |
| $\text{History-PRGF}_{0.05}$ | 704 | $\text{History-PARS}_{0.05}$ | 726 |

We found that in this task, ARS-based methods perform comparably to RGF-based ones. This could be because 1) the numbers of iterations until success of attacks are too small to show the advantage of ARS; 2) currently ARS is not guaranteed to converge faster than RGF under non-convex problems. We leave more evaluation of ARS-based methods in adversarial attacks and further improvement of their performance as future work. Experimental results show that History-PRGF is more robust to learning rate than RGF. However, a small learning rate could still lead to its deteriorated performance due to non-smoothness of the objective function. The same statement holds for ARS-based algorithms.

## 6 Conclusion and discussion

In this paper, we present a convergence analysis on existing prior-guided ZO optimization algorithms including PRGF and History-PRGF. We further propose a novel prior-guided ARS algorithm with convergence guarantee. Experimental results confirm our theoretical analysis.

Our limitations lie in: 1) we adopt a directional derivative oracle in our analysis, so the error on the convergence bound brought by finite-difference approximation has not been clearly stated; and 2) our implementation of PARS in practice requires an approximate solution of $\theta_t$, and the accuracy and influence of this approximation is not well studied yet. We leave these as future work. Other future work includes extension of the theoretical analysis to non-convex cases, and more empirical studies in various application tasks.

## Acknowledgements

This work was supported by the National Key Research and Development Program of China (No. 2020AAA0104304), NSFC Projects (Nos. 61620106010, 62061136001, 61621136008, 62076147, U19B2034, U19A2081, U1811461), Beijing NSF Project (No. JQ19016), Beijing Academy of Artificial Intelligence (BAAI), Tsinghua-Huawei Joint Research Program, a grant from Tsinghua Institute for Guo Qiang, Tiangong Institute for Intelligent Computing, and the NVIDIA NVAIL Program with GPU/DGX Acceleration.

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
