# A Supplemental materials for Section 2

## A.1 Definitions in convex optimization

**Definition 1** (Convexity). *A differentiable function $f$ is convex if for every $x, y \in \mathbb{R}^d$,*

$$f(y) \geq f(x) + \nabla f(x)^\top (y - x).$$

**Definition 2** (Smoothness). *A differentiable function $f$ is L-smooth for some positive constant $L$ if its gradient is L-Lipschitz; namely, for every $x, y \in \mathbb{R}^d$, we have*

$$\|\nabla f(x) - \nabla f(y)\| \leq L\|x - y\|.$$

**Corollary 1.** *If $f$ is L-smooth, then for every $x, y \in \mathbb{R}^d$,*

$$|f(y) - f(x) - \nabla f(x)^\top (y - x)| \leq \frac{1}{2}L\|y - x\|^2. \tag{12}$$

*Proof.* See Lemma 3.4 in [2]. $\qquad\qquad\square$

**Definition 3** (Strong convexity). *A differentiable function $f$ is $\tau$-strongly convex for some positive constant $\tau$, if for all $x, y \in \mathbb{R}^d$,*

$$f(y) \geq f(x) + \nabla f(x)^\top (y - x) + \frac{\tau}{2}\|y - x\|^2.$$

## A.2 Proof of Proposition 1

**Proposition 1.** *If $f$ is L-smooth, then for any $(x, v)$ with $\|v\| = 1$, $|g_\mu(v; x) - \nabla f(x)^\top v| \leq \frac{1}{2}L\mu$.*

*Proof.* In Eq. (12), setting $y = x + \mu v$ and dividing both sides by $\mu$, we complete the proof. $\qquad\square$

# B Supplemental materials for Section 3

## B.1 Proofs of Lemma 1, Theorem 1 and Theorem 2

**Lemma 1.** *Let $C_t := \left(\overline{\nabla f(x_t)}^\top v_t\right)^2$ and $L' := \frac{L}{1-\left(1-\frac{L}{\hat{L}}\right)^2}$, then in Algorithm 1, we have*

$$\mathbb{E}_t[\delta_{t+1}] \leq \delta_t - \frac{\mathbb{E}_t[C_t]}{2L'}\|\nabla f(x_t)\|^2. \tag{13}$$

*Proof.* We have

$$\delta_{t+1} - \delta_t = f(x_{t+1}) - f(x_t) \tag{14}$$

$$= f\left(x_t - \frac{1}{\hat{L}}\nabla f(x_t)^\top v_t \cdot v_t\right) - f(x_t) \tag{15}$$

$$\leq -\frac{1}{\hat{L}}\left(\nabla f(x_t)^\top v_t\right)^2 + \frac{1}{2}L \cdot \left(\frac{1}{\hat{L}}\nabla f(x_t)^\top v_t\right)^2 \tag{16}$$

$$= -\frac{1}{2L'}\left(\nabla f(x_t)^\top v_t\right)^2. \tag{17}$$

Hence,

$$\mathbb{E}_t[\delta_{t+1}] - \delta_t \leq -\frac{1}{2L'}\mathbb{E}_t\left[\left(\nabla f(x_t)^\top v_t\right)^2\right] = -\frac{\mathbb{E}_t[C_t]}{2L'}\|\nabla f(x_t)\|^2, \tag{18}$$

where the last equality holds because $x_t$ is $\mathcal{F}_{t-1}$-measurable since we require $\mathcal{F}_{t-1}$ to include all the randomness before iteration $t$ in Remark 2 (so $\|\nabla f(x_t)\|^2$ is also $\mathcal{F}_{t-1}$-measurable). $\qquad\square$

**Remark 9.** *The proof actually does not require $f$ to be convex. It only requires $f$ to be L-smooth.*

**Remark 10.** *From the proof we see that $\delta_{t+1} - \delta_t \leq 0$, so $f(x_{t+1}) \leq f(x_t)$. Hence, the sequence $\{f(x_t)\}_{t \geq 0}$ is non-increasing in Algorithm 1.*

**Theorem 1** (Algorithm 1, smooth and convex). *Let $R := \max_{x:f(x) \leq f(x_0)} \|x - x^*\|$ and suppose $R < \infty$. Then, in Algorithm 1, we have*

$$\mathbb{E}[\delta_T] \leq \frac{2L'R^2 \sum_{t=0}^{T-1} \mathbb{E}\left[\frac{1}{\mathbb{E}_t[C_t]}\right]}{T(T+1)}. \tag{19}$$

*Proof.* Since $f$ is convex, we have

$$\delta_t = f(x_t) - f(x^*) \leq \nabla f(x_t)^\top (x_t - x^*) \leq \|\nabla f(x_t)\| \cdot \|x_t - x^*\| \leq R\|\nabla f(x_t)\|, \tag{20}$$

where the last inequality follows from the definition of $R$ and the fact that $f(x_t) \leq f(x_0)$ (since $\delta_{t+1} \leq \delta_t$ for all $t$). The following proof is adapted from the proof of Theorem 3.2 in [1]. Define $\Phi_t := t(t+1)\delta_t$. By Lemma 1, we have

$$\mathbb{E}_t[\Phi_{t+1}] - \Phi_t = (t+1)(t+2)\mathbb{E}_t[\delta_{t+1}] - t(t+1)\delta_t \tag{21}$$

$$= (t+1)(t+2)(\mathbb{E}_t[\delta_{t+1}] - \delta_t) + 2(t+1)\delta_t \tag{22}$$

$$\leq -(t+1)(t+2)\frac{\mathbb{E}_t[C_t]}{2L'}\|\nabla f(x_t)\|^2 + 2(t+1)R\|\nabla f(x_t)\| \tag{23}$$

$$\leq \frac{(2(t+1)R)^2}{4(t+1)(t+2)\frac{\mathbb{E}_t[C_t]}{2L'}} \tag{24}$$

$$= \frac{2L'(t+1)R^2}{(t+2)\mathbb{E}_t[C_t]} \tag{25}$$

$$\leq \frac{2L'R^2}{\mathbb{E}_t[C_t]}, \tag{26}$$

where Eq. (24) follows from the fact that $-at^2 + bt \leq \frac{b^2}{4a}$ for $a > 0$. Hence

$$\mathbb{E}[\Phi_{t+1}] - \mathbb{E}[\Phi_t] = \mathbb{E}[\mathbb{E}_t[\Phi_{t+1}] - \Phi_t] \leq 2L'R^2\mathbb{E}\left[\frac{1}{\mathbb{E}_t[C_t]}\right]. \tag{27}$$

Since $\Phi_0 = 0$, we have $\mathbb{E}[\Phi_T] \leq 2L'R^2 \sum_{t=0}^{T-1} \mathbb{E}\left[\frac{1}{\mathbb{E}_t[C_t]}\right]$. Therefore,

$$\mathbb{E}[\delta_T] = \frac{\mathbb{E}[\Phi_T]}{T(T+1)} \leq \frac{2L'R^2 \sum_{t=0}^{T-1} \mathbb{E}\left[\frac{1}{\mathbb{E}_t[C_t]}\right]}{T(T+1)}. \tag{28}$$

$\square$

**Remark 11.** *By inspecting the proof, we note that Theorem 1 still holds if we replace the fixed initialization $x_0$ in Algorithm 1 with a random initialization $x_0'$ for which $f(x_0') \leq f(x_0)$ always holds. We formally summarize this in the following proposition. This proposition will be useful in the proof of Theorem 3.*

**Proposition 3.** *Let $x_{\text{fix}}$ be a fixed vector, $R := \max_{x:f(x) \leq f(x_{\text{fix}})} \|x - x^*\|$ and suppose $R < \infty$. Then, in Algorithm 1, using a random initialization $x_0$, if $f(x_0) \leq f(x_{\text{fix}})$ always hold, we have*

$$\mathbb{E}[\delta_T] \leq \frac{2L'R^2 \sum_{t=0}^{T-1} \mathbb{E}\left[\frac{1}{\mathbb{E}_t[C_t]}\right]}{T(T+1)}. \tag{29}$$

*Proof.* By Remark 10, $f(x_t) \leq f(x_0)$. We note that $\|x_t - x^*\| \leq R$ since $f(x_t) \leq f(x_0) \leq f(x_{\text{fix}})$. The remaining proof is the same as the proof of Theorem 1. $\square$

Next we state the proof regarding the convergence guarantee of Algorithm 1 under smooth and strongly convex case.

**Theorem 2** (Algorithm 1, smooth and strongly convex). *In Algorithm 1, if we further assume that $f$ is $\tau$-strongly convex, then we have*

$$\mathbb{E}\left[\frac{\delta_T}{\exp\left(-\frac{\tau}{L'}\sum_{t=0}^{T-1}\mathbb{E}_t[C_t]\right)}\right] \leq \delta_0. \tag{30}$$

*Proof.* Since $f$ is $\tau$-strongly convex, we have

$$\delta_t = f(x_t) - f(x^*) \leq \nabla f(x_t)^\top(x_t - x^*) - \frac{\tau}{2}\|x_t - x^*\|^2 \tag{31}$$

$$\leq \|\nabla f(x_t)\| \cdot \|x_t - x^*\| - \frac{\tau}{2}\|x_t - x^*\|^2 \tag{32}$$

$$\leq \frac{\|\nabla f(x_t)\|^2}{2\tau}. \tag{33}$$

Therefore we have

$$\|\nabla f(x_t)\|^2 \geq 2\tau\delta_t. \tag{34}$$

By Lemma 1 and Eq. (34) we have

$$\mathbb{E}_t[\delta_{t+1}] \leq \delta_t - \frac{\mathbb{E}_t[C_t]\tau}{L'}\delta_t = \left(1 - \frac{\tau}{L'}\mathbb{E}_t[C_t]\right)\delta_t. \tag{35}$$

Let $\alpha_t := \frac{\tau}{L'}\mathbb{E}_t[C_t]$, then $\mathbb{E}_t[\delta_{t+1}] \leq (1-\alpha_t)\delta_t$. We have

$$\delta_0 = \mathbb{E}[\delta_0] \geq \mathbb{E}\left[\frac{1}{1-\alpha_0}\mathbb{E}_0[\delta_1]\right] = \mathbb{E}\left[\mathbb{E}_0\left[\frac{\delta_1}{1-\alpha_0}\right]\right] = \mathbb{E}\left[\frac{\delta_1}{1-\alpha_0}\right]$$

$$\geq \mathbb{E}\left[\frac{\mathbb{E}_1[\delta_2]}{(1-\alpha_0)(1-\alpha_1)}\right] = \mathbb{E}\left[\mathbb{E}_1\left[\frac{\delta_2}{(1-\alpha_0)(1-\alpha_1)}\right]\right] = \mathbb{E}\left[\frac{\delta_2}{(1-\alpha_0)(1-\alpha_1)}\right]$$

$$\geq \cdots$$

$$\geq \mathbb{E}\left[\frac{\delta_T}{\prod_{t=0}^{T-1}(1-\alpha_t)}\right].$$

Since $\exp(-x) \geq 1 - x \geq 0$ when $0 \leq x \leq 1$, the proof is completed. $\square$

**Remark 12.** *Indeed, the proof does not require $f$ to be strongly convex or convex. It only requires the Polyak-Łojasiewicz condition (Eq. (34)) which is weaker than strong convexity [14, 8, 7].*

### B.2 Proof of Proposition 2

We note that in Algorithm 1, $\|v_t\| = 1$. If $v_t \in A$, then we have the following lemma by Proposition 1 in [10].

**Lemma 4.** *Let $u_1, u_2, \ldots, u_q$ be $q$ fixed vectors in $\mathbb{R}^d$ and $A := \operatorname{span}\{u_1, u_2, \ldots, u_q\}$ be the subspace spanned by $u_1, u_2, \ldots, u_q$. Let $\nabla f(x_t)_A$ denote the projection of $\nabla f(x_t)$ onto $A$, then $\overline{\nabla f(x_t)_A} = \operatorname{argmax}_{v_t \in A, \|v_t\|=1} C_t$.*

We further note that $\nabla f(x_t)_A$ could be calculated with the values of $\{\nabla f(x_t)^\top u_i\}_{i=1}^q$:

**Lemma 5.** *Let $A := \operatorname{span}\{u_1, u_2, \ldots, u_q\}$ be the subspace spanned by $u_1, u_2, \ldots, u_q$, and suppose $\{u_1, u_2, \ldots, u_q\}$ is linearly independent (if they are not, then we choose a subset of these vectors which is linearly independent). Then $\nabla f(x_t)_A = \sum_{i=1}^q a_i u_i$, where $a := (a_1, a_2, \cdots, a_q)^\top$ is given by $a = G^{-1}b$, where $G$ is a $q \times q$ matrix in which $G_{ij} = u_i^\top u_j$, $b$ is a $q$-dimensional vector in which $b_i = \nabla f(x_t)^\top u_i$.*

*Proof.* Since $\nabla f(x_t)_A \in \operatorname{span}\{u_1, u_2, \ldots, u_q\}$, there exists $a \in \mathbb{R}^q$ such that $\nabla f(x_t)_A = \sum_{i=1}^q a_i u_i$. Since $\nabla f(x_t)_A$ is the projection of $\nabla f(x_t)$ onto $A$ and $u_1, u_2, \ldots, u_q \in A$, $\nabla f(x_t)_A^\top u_i = \nabla f(x_t)^\top u_i$ holds for any $i$. Therefore, $Ga = b$. Since $\{u_1, u_2, \ldots, u_q\}$ is linearly independent and $G$ is corresponding Gram matrix, $G$ is invertible. Hence $a = G^{-1}b$. $\square$

Therefore, if we suppose $v_t \in A$, then the optimal $v_t$ is given by $\overline{\nabla f(x_t)_A}$, which could be calculated from $\{\nabla f(x_t)^\top u_i\}_{i=1}^q$. Now we are ready to prove Proposition 2 through an additional justification.

**Proposition 2** (Optimality of subspace estimator). *In one iteration of Algorithm 1, if we have queried $\{\nabla f(x_t)^\top u_i\}_{i=1}^q$, then the optimal $v_t$ maximizing $C_t$ s.t. $\|v_t\| = 1$ should be in the following form: $v_t = \overline{\nabla f(x_t)_A}$, where $A := \mathrm{span}\{u_1, u_2, \ldots, u_q\}$.*

*Proof.* It remains to justify the assumption that $v_t \in A$. We note that in Line 3 of Algorithm 1, generally it requires 1 additional call to query the value of $\nabla f(x_t)^\top v_t$, but if $v_t \in A$, then we can always save this query by calculating $\nabla f(x_t)^\top v_t$ with the values of $\{\nabla f(x_t)^\top u_i\}_{i=1}^q$, since if $v_t \in A$, then we can write $v_t$ in the form $v_t = \sum_{i=1}^q a_i u_i$, and hence $\nabla f(x_t)^\top v_t = \sum_{i=1}^q a_i \nabla f(x_t)^\top u_i$. Now suppose we finally sample a $v_t \notin A$. Then this additional query of $\nabla f(x_t)^\top v_t$ is necessary. Now we could let $A' := \mathrm{span}\{u_1, u_2, \ldots, u_q, v_t\}$ and calculate $v_t' = \overline{\nabla f(x_t)_{A'}}$. Obviously, $(\overline{\nabla f(x_t)}^\top v_t')^2 \geq (\overline{\nabla f(x_t)}^\top v_t)^2$, suggesting that $v_t'$ is better than $v_t$. Therefore, without loss of generality we can always assume $v_t \in A$, and by Lemma 4 the proof is complete. $\qquad\square$

### B.3 Details regarding RGF and PRGF estimators

#### B.3.1 Construction of RGF estimator

In Example 1, we mentioned that the RGF estimator is given by $v_t = \overline{\nabla f(x_t)}_A$ where $A = \mathrm{span}\{u_1, u_2, \ldots, u_q\}$ $(q > 0)$ and $\forall i, u_i \sim \mathcal{U}(\mathbb{S}_{d-1})$ ($u \sim \mathcal{U}(\mathbb{S}_{d-1})$ means that $u$ is sampled uniformly from the $(d-1)$-dimensional unit sphere, as a normalized $d$-dimensional random vector), and $u_1, u_2, \ldots, u_q$ are sampled independently. Now we present the detailed expression of $v_t$ by explicitly orthogonalizing $\{u_1, u_2, \ldots, u_q\}$:

$$u_1 \sim \mathcal{U}(\mathbb{S}_{d-1});$$
$$u_2 = \overline{(\mathbf{I} - u_1 u_1^\top)\xi_2}, \xi_2 \sim \mathcal{U}(\mathbb{S}_{d-1});$$
$$u_3 = \overline{(\mathbf{I} - u_1 u_1^\top - u_2 u_2^\top)\xi_3}, \xi_3 \sim \mathcal{U}(\mathbb{S}_{d-1});$$
$$\ldots$$
$$u_q = \overline{\left(\mathbf{I} - \sum_{i=1}^{q-1} u_i u_i^\top\right)\xi_q}, \xi_q \sim \mathcal{U}(\mathbb{S}_{d-1}).$$

Then we let $v_t = \overline{\sum_{i=1}^q \nabla f(x_t)^\top u_i \cdot u_i}$. Since $g_t = \nabla f(x_t)^\top \overline{\nabla f(x_t)_A} \cdot \overline{\nabla f(x_t)_A} = \nabla f(x_t)_A$, we have $g_t = \sum_{i=1}^q \nabla f(x_t)^\top u_i \cdot u_i$. Therefore, when using the RGF estimator, each iteration in Algorithm 1 costs $q$ queries to the directional derivative oracle.

#### B.3.2 Properties of RGF estimator

In this section we show that for RGF estimator with $q$ queries, $\mathbb{E}_t[C_t] = \frac{q}{d}$. We first state a simple proposition here.

**Proposition 4.** *If $v_t = \overline{\sum_{i=1}^q \nabla f(x_t)^\top u_i \cdot u_i}$ and $u_1, u_2, \ldots, u_q$ are orthonormal, then*

$$\left(\overline{\nabla f(x_t)}^\top v_t\right)^2 = \sum_{i=1}^q \left(\overline{\nabla f(x_t)}^\top u_i\right)^2. \tag{36}$$

*Proof.* Since $\nabla f(x_t)^\top v_t \cdot v_t := g_t = \sum_{i=1}^q \nabla f(x_t)^\top u_i \cdot u_i$, we have $\overline{\nabla f(x_t)}^\top v_t \cdot v_t = \sum_{i=1}^q \overline{\nabla f(x_t)}^\top u_i \cdot u_i$. Taking inner product with $\overline{\nabla f(x_t)}$ to both sides, we obtain the result. $\qquad\square$

By Proposition 4,

$$\mathbb{E}_t[C_t] = \mathbb{E}_t\left[\left(\overline{\nabla f(x_t)}^\top v_t\right)^2\right] = \sum_{i=1}^q \mathbb{E}_t\left[\left(\overline{\nabla f(x_t)}^\top u_i\right)^2\right] = \sum_{i=1}^q \left(\overline{\nabla f(x_t)}^\top \mathbb{E}_t[u_i u_i^\top]\overline{\nabla f(x_t)}\right). \tag{37}$$

In RGF, $u_i$ is independent of the history, so in this section we directly write $\mathbb{E}[u_i u_i^\top]$ instead of $\mathbb{E}_t[u_i u_i^\top]$.

For $i = 1$, since $u_1 \sim \mathcal{U}(\mathbb{S}_{d-1})$, we have $\mathbb{E}[u_1 u_1^\top] = \frac{\mathbf{I}}{d}$. (Explanation: the distribution of $u_1$ is symmetric, hence $\mathbb{E}[u_1 u_1^\top]$ should be something like $a\mathbf{I}$; since $\mathrm{Tr}(\mathbb{E}[u_1 u_1^\top]) = \mathbb{E}[u_1^\top u_1] = 1$, $a = 1/\mathrm{Tr}(\mathbf{I}) = 1/d$.)

For $i = 2$, we have $\mathbb{E}[u_2 u_2^\top | u_1] = \frac{\mathbf{I} - u_1 u_1^\top}{d-1}$. (See Section A.2 in [4] for the proof.) Therefore, $\mathbb{E}[u_2 u_2^\top] = \mathbb{E}[\mathbb{E}[u_2 u_2^\top | u_1]] = \frac{\mathbf{I} - \mathbb{E}[u_1 u_1^\top]}{d-1} = \frac{\mathbf{I}}{d}$.

Then by induction, we have that $\forall 1 \leq i \leq q$, $\mathbb{E}[u_i u_i^\top] = \frac{\mathbf{I}}{d}$. Hence by Eq. (37), $\mathbb{E}_t[C_t] = \frac{q}{d}$.

### B.3.3 Construction of PRGF estimator

In Example 2, we mentioned that the PRGF estimator is given by $v_t = \overline{\nabla f(x_t)}_A$ where $A = \mathrm{span}\{p_t, u_1, u_2, \ldots, u_q\}$ $(q > 0)$, where $p_t$ is a vector corresponding to the prior message which is available at the beginning of iteration $t$, and $\forall i, u_i \sim \mathcal{U}(\mathbb{S}_{d-1})$ $(u_1, u_2, \ldots, u_q$ are sampled independently). Now we present the detailed expression of $v_t$ by explicitly orthogonalizing $\{p_t, u_1, u_2, \ldots, u_q\}$. We note that here we leave $p_t$ unchanged (we only normalize it, i.e. $p_t \leftarrow \frac{p_t}{\|p_t\|}$ if $\|p_t\| \neq 1$) and make $\{u_1, u_2, \ldots, u_q\}$ orthogonal to $p_t$. Specifically, given a positive integer $q \leq d - 1$,

$$u_1 = \overline{(\mathbf{I} - p_t p_t^\top)\xi_1}, \xi_1 \sim \mathcal{U}(\mathbb{S}_{d-1});$$
$$u_2 = \overline{(\mathbf{I} - p_t p_t^\top - u_1 u_1^\top)\xi_2}, \xi_2 \sim \mathcal{U}(\mathbb{S}_{d-1});$$
$$u_3 = \overline{(\mathbf{I} - p_t p_t^\top - u_1 u_1^\top - u_2 u_2^\top)\xi_3}, \xi_3 \sim \mathcal{U}(\mathbb{S}_{d-1});$$
$$\ldots$$
$$u_q = \overline{\left(\mathbf{I} - p_t p_t^\top - \sum_{i=1}^{q-1} u_i u_i^\top\right)\xi_q}, \xi_q \sim \mathcal{U}(\mathbb{S}_{d-1}).$$

Then we let $v_t = \overline{\nabla f(x_t)^\top p_t \cdot p_t + \sum_{i=1}^q \nabla f(x_t)^\top u_i \cdot u_i}$. Since $g_t = \nabla f(x_t)^\top \overline{\nabla f(x_t)_A} \cdot \overline{\nabla f(x_t)_A} = \nabla f(x_t)_A$, we have $g_t = \nabla f(x_t)^\top p_t \cdot p_t + \sum_{i=1}^q \nabla f(x_t)^\top u_i \cdot u_i$. Therefore, when using the PRGF estimator, each iteration in Algorithm 1 costs $q + 1$ queries to the directional derivative oracle.

### B.3.4 Properties of PRGF estimator

Here we prove Lemma 2 in the main article (its proof appears in [10]; we prove it in our language here), but for later use we give a more useful formula here, which can derive Lemma 2. Let $D_t := \left(\overline{\nabla f(x_t)}^\top p_t\right)^2$. We have

**Proposition 5.** *For $t \geq 1$,*
$$C_t = D_t + (1 - D_t)\xi_t^2, \tag{38}$$

*where $\xi_t^2 := \sum_{i=1}^q \xi_{ti}^2$, $\xi_{ti} := \overline{\nabla f(x_t)_H}^\top u_i$[14] in which $e_H := e - p_t p_t^\top e$ denotes the projection of the vector $e$ onto the $(d-1)$-dimensional subspace $H$, of which $p_t$ is a normal vector.*

*Proof.* By Proposition 4, we have
$$C_t = \left(\overline{\nabla f(x_t)}^\top v_t\right)^2 = D_t + \sum_{i=1}^q \left(\overline{\nabla f(x_t)}^\top u_i\right)^2. \tag{39}$$

By the definition of $u_1, u_2, \ldots, u_q$, they are in the subspace $H$. Therefore
$$\left(\overline{\nabla f(x_t)}^\top u_i\right)^2 = \left(\overline{\nabla f(x_t)}_H^\top u_i\right)^2 = \|\overline{\nabla f(x_t)}_H\|^2 \left(\overline{\nabla f(x_t)_H}^\top u_i\right)^2 = (1 - D_t)\left(\overline{\nabla f(x_t)_H}^\top u_i\right)^2. \tag{40}$$

---

[14]Note that in different iterations, $\{u_i\}$ are different. Hence here we explicitly show this dependency on $t$ in the subscript of $\xi$.

By Eq. (39) and Eq. (40), the proposition is proved. □

Next we state $\mathbb{E}_t[\xi_t^2]$, the conditional expectation of $\xi_t^2$ given the history $\mathcal{F}_{t-1}$. We can also derive it in the similar way as in Section B.3.2, but for later use let us describe the distribution of $\xi_t^2$ in a more convenient way. We note that the conditional distribution of $u_i$ is the uniform distribution from the unit sphere in the $(d-1)$-dimensional subspace $H$. Since $\xi_{ti} := \overline{\nabla f(x_t)_H}^\top u_i$, $\xi_{ti}$ is indeed the inner product between one fixed unit vector and one uniformly random sampled unit vector in $H$. Indeed, $\xi_t^2$ is equal to $\left\| \left( \overline{\nabla f(x_t)_H} \right)_{A'} \right\|^2$ where $A' := \mathrm{span}(u_1, u_2, \cdots, u_q)$ is a random $q$-dimensional subspace of $H$. Therefore, $\xi_t^2$ is equal to the squared norm of the projection of a fixed unit vector in $H$ to a random $q$-dimensional subspace of $H$. By the discussion in the proof of Lemma 5.3.2 in [15], we can view a random projection acting on a fixed vector as a fixed projection acting on a random vector. Therefore, we state the following proposition.

**Proposition 6.** *The conditional distribution of $\xi_t^2$ given $\mathcal{F}_{t-1}$ is the same as the distribution of $\sum_{i=1}^{q} z_i^2$, where $(z_1, z_2, \ldots, z_{d-1})^\top \sim \mathcal{U}(\mathbb{S}^{d-2})$, where $\mathbb{S}^{d-2}$ is the unit sphere in $\mathbb{R}^{d-1}$.*

Then it is straightforward to prove the following proposition.

**Proposition 7.** $\mathbb{E}_t[\xi_t^2] = \frac{q}{d-1}$.

*Proof.* By symmetry, $\mathbb{E}[z_i^2] = \mathbb{E}[z_j^2] \; \forall i,j$. Since $\mathbb{E}[\sum_{i=1}^{d-1} z_i^2] = 1$, $\mathbb{E}[z_i^2] = \frac{1}{d-1}$. Hence by Proposition 6, $\mathbb{E}_t[\xi_t^2] = \mathbb{E}[\sum_{i=1}^{q} z_i^2] = \frac{q}{d-1}$. □

Now we reach Lemma 2.

**Lemma 2.** *In Algorithm 1 with PRGF estimator,*

$$\mathbb{E}_t[C_t] = D_t + \frac{q}{d-1}(1 - D_t), \tag{41}$$

*where $D_t := \left( \overline{\nabla f(x_t)}^\top p_t \right)^2$.*

*Proof.* Since $D_t$ is $\mathcal{F}_{t-1}$-measurable, by Proposition 5 and Proposition 7, we have

$$\mathbb{E}_t[C_t] = D_t + (1 - D_t)\mathbb{E}_t[\xi_t^2] = D_t + \frac{q}{d-1}(1 - D_t).$$

□

Finally, we note that Proposition 6 implies that $\xi_t^2$ is independent of the history (indeed, for all $i$, $\xi_{ti}^2$ is independent of the history). For convenience, in the following, when we need the conditional expectation (given some historical information) of quantities only related to $\xi_t^2$, we could directly write the expectation without conditioning. For example, we directly write $\mathbb{E}[\xi_t^2]$ instead of $\mathbb{E}_t[\xi_t^2]$, and write $\mathrm{Var}[\xi_t^2]$ instead of the conditional variance $\mathrm{Var}_t[\xi_t^2]$.

## B.4 Proof of Lemma 3 and evolution of $\mathbb{E}[C_t]$

In this section, we discuss the key properties of History-PRGF before presenting the theorems in Section B.5. First we mention that while in History-PRGF we choose the prior $p_t$ to be $v_{t-1}$, we can choose $p_0$ as any fixed normalized vector. We first present a lemma which is useful for the proof of Lemma 3.

**Lemma 6** (Proof in Section B.4.1). *Let $a$, $b$ and $c$ be vectors in $\mathbb{R}^d$, $\|a\| = \|c\| = 1$, $B := \{b : \|b - a\| \leq k \cdot a^\top c\}$, $0 \leq k \leq 1$, $a^\top c \geq 0$. Then $\min_{b \in B} \bar{b}^\top c \geq \min_{b \in B} b^\top c = (1 - k)a^\top c$.*

**Lemma 3.** *In History-PRGF ($p_t = v_{t-1}$), we have*

$$D_t \geq \left( 1 - \frac{L}{\hat{L}} \right)^2 C_{t-1}. \tag{42}$$

*Proof.* In History-PRGF $p_t = v_{t-1}$, so by the definitions of $D_t$ and $C_t$ we are going to prove

$$\left(\overline{\nabla f(x_t)}^\top v_{t-1}\right)^2 \geq \left(1 - \frac{L}{\hat{L}}\right)^2 \left(\overline{\nabla f(x_{t-1})}^\top v_{t-1}\right)^2. \tag{43}$$

Without loss of generality, assume $\nabla f(x_{t-1})^\top v_{t-1} \geq 0$. Since $f$ is $L$-smooth, we have

$$\|\nabla f(x_t) - \nabla f(x_{t-1})\| \leq L\|x_t - x_{t-1}\| = \frac{L}{\hat{L}}\nabla f(x_{t-1})^\top v_{t-1}, \tag{44}$$

which is equivalent to

$$\left\|\frac{\nabla f(x_t)}{\|\nabla f(x_{t-1})\|} - \overline{\nabla f(x_{t-1})}\right\| \leq \frac{L}{\hat{L}}\overline{\nabla f(x_{t-1})}^\top v_{t-1}. \tag{45}$$

Let $a = \overline{\nabla f(x_{t-1})}$, $b = \frac{\nabla f(x_t)}{\|\nabla f(x_{t-1})\|}$, $c = v_{t-1}$. By Lemma 6 we have

$$\overline{\nabla f(x_t)}^\top v_{t-1} \geq \left(1 - \frac{L}{\hat{L}}\right)\overline{\nabla f(x_{t-1})}^\top v_{t-1}. \tag{46}$$

By the definition of $v_t$, the right-hand side is non-negative. Taking square on both sides, the proof is completed. $\qquad\square$

When considering the lower bound related to $C_t$, we can replace the inequality with equality in Lemma 3. Therefore, by Proposition 5 and Lemma 3, we now have full knowledge of evolution of $C_t$. We summarize the above discussion in the following proposition. We define $a' := \left(1 - \frac{L}{\hat{L}}\right)^2$ in the following.

**Proposition 8.** *Let $a' := \left(1 - \frac{L}{\hat{L}}\right)^2$. Then in History-PRGF, we have*

$$C_t \geq a'C_{t-1} + (1 - a'C_{t-1})\xi_t^2. \tag{47}$$

*Proof.* By Proposition 5, $C_t = (1 - \xi_t^2)D_t + \xi_t^2$. By Lemma 3, $D_t \geq a'C_{t-1}$. Since $\xi_t^2 \leq 1$, we obtain the result. $\qquad\square$

As an appetizer, we discuss the evolution of $\mathbb{E}[C_t]$ here using Lemma 2 and Lemma 3 in the following proposition.

**Proposition 9.** *Suppose $\frac{q}{d-1} = k\frac{L}{\hat{L}}$ ($k > 0$), then in History-PRGF, $\mathbb{E}[C_t] \geq (1 - e^{-n})\frac{2}{2+k}\frac{q}{d-1}\frac{1}{1-a'}$ for $t \geq n\frac{d-1}{q}$.*

*Proof.* By Eq. (47), we have

$$\mathbb{E}[C_t] = \mathbb{E}[\mathbb{E}_t[C_t]] \geq \mathbb{E}[a'C_{t-1} + (1 - a'C_{t-1})\mathbb{E}_t[\xi_t^2]] \tag{48}$$

$$= \mathbb{E}[a'C_{t-1} + (1 - a'C_{t-1})\mathbb{E}[\xi_t^2]] \tag{49}$$

$$= \left(1 - \frac{q}{d-1}\right)a'\mathbb{E}[C_{t-1}] + \frac{q}{d-1}. \tag{50}$$

Letting $a := a'(1 - \frac{q}{d-1})$, $b := \frac{q}{d-1}$, then $\mathbb{E}[C_t] \geq a\mathbb{E}[C_{t-1}] + b$ and $0 \leq a < 1$. We have $\mathbb{E}[C_t] - \frac{b}{1-a} \geq a(\mathbb{E}[C_{t-1}] - \frac{b}{1-a}) \geq a^2(\mathbb{E}[C_{t-2}] - \frac{b}{1-a}) \geq \ldots \geq a^t(\mathbb{E}[C_0] - \frac{b}{1-a})$, hence $\mathbb{E}[C_t] \geq \frac{b}{1-a} - a^t(\frac{b}{1-a} - \mathbb{E}[C_0]) \geq (1 - a^t)\frac{b}{1-a}$.

Since $1 - a = 1 - (1 - \frac{q}{d-1})(1 - \frac{L}{\hat{L}})^2 = 1 - (1 - k\frac{L}{\hat{L}})(1 - \frac{L}{\hat{L}})^2$, noting that

$$\frac{1 - (1 - \frac{L}{\hat{L}})^2}{1 - (1 - k\frac{L}{\hat{L}})(1 - \frac{L}{\hat{L}})^2} = \frac{\frac{L}{\hat{L}} + \frac{L}{\hat{L}}(1 - \frac{L}{\hat{L}})}{\frac{L}{\hat{L}} + \frac{L}{\hat{L}}(1 - \frac{L}{\hat{L}}) + k\frac{L}{\hat{L}}(1 - \frac{L}{\hat{L}})^2} \geq \frac{2}{2 + k}, \tag{51}$$

we have $\frac{1-a'}{1-a} \geq \frac{2}{2+k}$. Meanwhile, $a \leq 1 - \frac{q}{d-1}$. Therefore, if $t \geq n\frac{d-1}{q}$, we have

$$a^t \leq (1 - \frac{q}{d-1})^{n\frac{d-1}{q}} \leq \exp(-\frac{q}{d-1})^{n\frac{d-1}{q}} = e^{-n}. \tag{52}$$

Since $\frac{1-a'}{1-a} \geq \frac{2}{2+k}$ and $a^t \leq e^{-n}$, we have

$$\mathbb{E}[C_t] \geq (1 - a^t)\frac{b}{1-a} \geq \frac{2}{2+k}(1 - e^{-n})\frac{1}{1-a'}\frac{q}{d-1}. \tag{53}$$

$\square$

**Corollary 2.** *In History-PRGF,* $\liminf_{t\to\infty} \mathbb{E}[C_t] \geq \frac{2}{2+k}\frac{q}{d-1}\frac{1}{1-a'}$.

Recalling that $L' := \frac{L}{1-a'}$, the propositions above tell us that $\mathbb{E}[C_t]$ tends to $O\left(\frac{q}{d}\frac{L'}{L}\right)$ in a fast rate, as long as $k$ is small, e.g. when $\frac{q}{d} \leq \frac{L}{\hat{L}}$ (which means that the chosen learning rate $\frac{1}{\hat{L}}$ is not too small compared with the optimal learning rate $\frac{1}{L}$: $\frac{1}{\hat{L}} \geq \frac{q}{d}\frac{1}{L}$). If $\mathbb{E}[C_t] \approx \frac{q}{d}\frac{L'}{L}$, then $\frac{\mathbb{E}[C_t]}{L'}$ is not dependent on $L'$ (and thus independent of $\hat{L}$). By Lemma 1, Theorem 1 and Theorem 2, this roughly means that the convergence rate is robust to the choice of $\hat{L}$, i.e. robust to the choice of learning rate. Specifically, History-PRGF with $\hat{L} > L$ (but $\hat{L}$ is not too large) could roughly recover the performance of RGF with $\hat{L} = L$, since $\frac{\mathbb{E}[C_t]}{L'} \approx \frac{q}{d}$ where $\frac{q}{d}$ is the value of $\mathbb{E}_t[C_t]$ when using the RGF estimator.

### B.4.1 Proof of Lemma 6

In this section, we first give a lemma for the proof of Lemma 6.

**Lemma 7.** *Let $a$ and $b$ be vectors in $\mathbb{R}^d$, $\|a\| = 1$, $\|b\| \geq 1$. Then $\|\bar{b} - a\| \leq \|b - a\|$.*

*Proof.*

$$\|b - a\|^2 - \|\bar{b} - a\|^2 = \|b - \bar{b}\|^2 + 2(b - \bar{b})^\top(\bar{b} - a) \tag{54}$$
$$\geq 2(b - \bar{b})^\top(\bar{b} - a) \tag{55}$$
$$= 2(\|b\| - 1)\bar{b}^\top(\bar{b} - a) \tag{56}$$
$$= 2(\|b\| - 1)(1 - \bar{b}^\top a) \tag{57}$$
$$\geq 0. \tag{58}$$

$\square$

Then, the detailed proof of **Lemma 6** is as follows.

*Proof.* $\forall b \in B$, $b^\top c = a^\top c - (a - b)^\top c \geq a^\top c - \|a - b\|\|c\| \geq (1 - k)a^\top c$, and both equality holds when $b = a - k \cdot a^\top c \cdot c$.

**Case 1:** $\|b\| \geq 1$   By Lemma 7 we have $\|\bar{b} - a\| \leq \|b - a\|$, hence if $b \in B$, then $\bar{b} \in B$, so when $\|b\| \geq 1$ we have $\bar{b}^\top c \geq \min_{b \in B} b^\top c$.

**Case 2:** $\|b\| < 1$   $\forall b \in B$, if $\|b\| \leq 1$, then $\bar{b}^\top c = \frac{b^\top c}{\|b\|} \geq b^\top c \geq \min_{b \in B} b^\top c$.

The proof of the lemma is completed. $\square$

## B.5 Proofs of Theorem 3 and Theorem 4

### B.5.1 Proof of Theorem 3

As mentioned above, we define $a' := \left(1 - \frac{L}{\hat{L}}\right)^2$ to be used in the proofs. In the analysis, we first try to replace the inequality in Lemma 3 with equality. To do that, similar to Eq. (47), we define $\{E_t\}_{t=0}^{T-1}$ as follows: $E_0 = 0$, and

$$E_t = a'E_{t-1} + (1 - a'E_{t-1})\xi_t^2, \tag{59}$$

where $\xi_t^2$ is defined in Proposition 5.

First, we give the following lemmas, which is useful for the proof of Theorem 3.

**Lemma 8** (Upper-bounded variance; proof in Section B.5.2). *If $d \geq 4$, then $\forall t$, $\mathrm{Var}[\mathbb{E}_t[E_t]] \leq \frac{1}{1-(a')^2}\frac{2q}{(d-1)^2}$.*

**Lemma 9** (Lower-bounded expectation; proof in Section B.5.2). *If $\frac{q}{d-1} \leq \frac{L}{\tilde{L}}$ and $t \geq \frac{d-1}{q}$, then*

$$\mathbb{E}[E_t] \geq \frac{1}{2}\frac{1}{1-a'}\frac{q}{d-1}. \tag{60}$$

**Lemma 10** (Proof in Section B.5.2). *If a random variable $X \geq B > 0$ satisfies that $\mathbb{E}[X] \geq \mu B$, $\mathrm{Var}[X] \leq (\sigma B)^2$, then*

$$\mathbb{E}\left[\frac{1}{X}\right] \leq \frac{1}{\mu B}\left(\frac{4\sigma^2}{\mu} + 2\right). \tag{61}$$

Then, we provide the proof of Theorem 3 in the following.

**Theorem 3** (History-PRGF, smooth and convex). *In the setting of Theorem 1, when using the History-PRGF estimator, assuming $d \geq 4$, $\frac{q}{d-1} \leq \frac{L}{\tilde{L}} \leq 1$ and $T > \lceil \frac{d}{q} \rceil$ ($\lceil \cdot \rceil$ denotes the ceiling function), we have*

$$\mathbb{E}[f(x_T)] - f(x^*) \leq \left(\frac{32}{q} + 2\right)\frac{2L\frac{d}{q}R^2}{T - \lceil \frac{d}{q} \rceil + 1}. \tag{62}$$

*Proof.* Since $E_0 = 0 \leq C_0$, and if $E_{t-1} \leq C_{t-1}$, then

$$E_t = a'E_{t-1} + (1 - a'E_{t-1})\xi_t^2 \tag{63}$$
$$\leq a'C_{t-1} + (1 - a'C_{t-1})\xi_t^2 \tag{64}$$
$$\leq C_t, \tag{65}$$

in which the first inequality is because $\xi_t^2 \leq 1$ and the second inequality is due to Eq. (47). Therefore by mathematical induction we have that $\forall t$, $E_t \leq C_t$.

Next, if $d \geq 4$, $\frac{q}{d-1} \leq \frac{L}{\tilde{L}}$ and $t \geq \frac{d-1}{q}$, by Lemma 8 and Lemma 9, if we set $B = \frac{q}{d-1}$, then $\mathbb{E}[\mathbb{E}_t[E_t]] = \mathbb{E}[E_t] \geq \frac{1}{2}\frac{1}{1-a'}B$, and $\mathrm{Var}[\mathbb{E}_t[E_t]] \leq \frac{2}{q}\frac{1}{1-(a')^2}B^2$. Meanwhile, if $t \geq 1$, then $\mathbb{E}_t[E_t] = a'(1 - \frac{q}{d-1})E_{t-1} + \frac{q}{d-1} \geq B$. Therefore, by Lemma 10 we have

$$\mathbb{E}\left[\frac{1}{\mathbb{E}_t[E_t]}\right] \leq \frac{1}{\frac{1}{2}\frac{1}{1-a'}\frac{q}{d-1}}\left(\frac{4\frac{2}{q}\frac{1}{1-(a')^2}}{\frac{1}{2}\frac{1}{1-a'}} + 2\right) \tag{66}$$

$$= \frac{d-1}{q}(1-a')\left(\frac{32}{q}\frac{1-a'}{1-(a')^2} + 2\right) \tag{67}$$

$$\leq \frac{d}{q}(1-a')\left(\frac{32}{q} + 2\right) \tag{68}$$

$$= \frac{d}{q}\frac{L}{L'}\left(\frac{32}{q} + 2\right). \tag{69}$$

Since $E_t \leq C_t$, $\mathbb{E}_t[E_t] \leq \mathbb{E}_t[C_t]$. Let $s := \lceil \frac{d}{q} \rceil$, then $\forall t \geq s$, $\mathbb{E}\left[\frac{1}{\mathbb{E}_t[C_t]}\right] \leq \mathbb{E}\left[\frac{1}{\mathbb{E}_t[E_t]}\right] \leq \frac{d}{q}\frac{L}{L'}\left(\frac{32}{q} + 2\right)$. Now imagine that we run History-PRGF algorithm with $x_s$ as the random initialization in Algorithm 1, and set $p_0$ to $v_{s-1}$. Then quantities in iteration $t$ (e.g. $x_t$, $v_t$, $C_t$) in the imaginary setting have the same distribution as quantities in iteration $t + s$ (e.g. $x_{t+s}$, $v_{t+s}$, $C_{t+s}$) in the original algorithm (indeed, the quantities before iteration $t$ in the imaginary setting have the same joint distribution as the quantities from iteration $s$ to iteration $t + s - 1$ in the original algorithm), and $\mathcal{F}_{t-1}$ in the imaginary setting corresponds to $\mathcal{F}_{t+s-1}$ in the original algorithm. Now we apply Proposition 3 to the imaginary setting, and we note that if we set $x_{\mathrm{fix}}$ to the original $x_0$,

then the condition in Proposition 3 holds (since by Remark 10, $f(x_s) \leq f(x_0)$). Since quantities in iteration $t$ in the original algorithm correspond to quantities in iteration $t - s$ in the imaginary setting, Proposition 3 tells us that if $T > s$, we have

$$\mathbb{E}[f(x_T)] - f(x^*) \leq \frac{2L'R^2 \sum_{t=s}^{T-1} \mathbb{E}\left[\frac{1}{\mathbb{E}_t[C_t]}\right]}{(T-s)(T-s+1)}, \tag{70}$$

so

$$\mathbb{E}[f(x_T)] - f(x^*) \leq \left(\frac{32}{q} + 2\right) \frac{2L\frac{d}{q}R^2}{T - \left\lceil\frac{d}{q}\right\rceil + 1}. \tag{71}$$

$\square$

### B.5.2 Proofs of Lemma 8, 9 and 10

In this section, we first present **Lemma 11** for the proof of Lemma 8.

**Lemma 11.** *Suppose $d \geq 3$, then $\mathrm{Var}[\xi_t^2] < \frac{2q}{(d-1)^2}$.*

*Proof.* For convenience, denote $D := d - 1$ in the following. By Proposition 6, the distribution of $\xi_t^2$ is the same as the distribution of $\sum_{i=1}^q z_i^2$, where $(z_1, \cdots, z_q)^\top \sim \mathcal{U}(\mathbb{S}^{D-1})$. We note that the distribution of $z$ is the same as the distribution of $\frac{x}{\|x\|}$, where $x \sim \mathcal{N}(0, \mathbf{I})$. Therefore,

$$\mathbb{E}\left[\left(\sum_{i=1}^q z_i^2\right)^2\right] = \mathbb{E}\left[\left(\sum_{i=1}^q \frac{x_i^2}{\|x\|^2}\right)^2\right] = \mathbb{E}\left[\frac{\left(\sum_{i=1}^q x_i^2\right)^2}{\left(\sum_{i=1}^D x_i^2\right)^2}\right]. \tag{72}$$

By Theorem 1 in [6], $\frac{\sum_{i=1}^q x_i^2}{\|x\|^2}$ and $\|x\|^2$ are independently distributed. Therefore, $\frac{\left(\sum_{i=1}^q x_i^2\right)^2}{\left(\sum_{i=1}^D x_i^2\right)^2}$ and $\left(\sum_{i=1}^D x_i^2\right)^2$ are independently distributed, which implies

$$\mathbb{E}\left[\frac{\left(\sum_{i=1}^q x_i^2\right)^2}{\left(\sum_{i=1}^D x_i^2\right)^2}\right] = \frac{\mathbb{E}\left[\left(\sum_{i=1}^q x_i^2\right)^2\right]}{\mathbb{E}\left[\left(\sum_{i=1}^D x_i^2\right)^2\right]}. \tag{73}$$

We note that $\sum_{i=1}^q x_i^2$ follows the chi-squared distribution with $q$ degrees of freedom. Therefore, $\mathbb{E}\left[\sum_{i=1}^q x_i^2\right] = q$, and $\mathrm{Var}\left[\sum_{i=1}^q x_i^2\right] = 2q$. Therefore, $\mathbb{E}\left[\left(\sum_{i=1}^q x_i^2\right)^2\right] = \mathbb{E}\left[\sum_{i=1}^q x_i^2\right]^2 + \mathrm{Var}\left[\sum_{i=1}^q x_i^2\right] = q(q+2)$. Hence

$$\mathrm{Var}\left[\sum_{i=1}^q z_i^2\right] = \mathbb{E}\left[\left(\sum_{i=1}^q z_i^2\right)^2\right] - \mathbb{E}\left[\sum_{i=1}^q z_i^2\right]^2 = \frac{q(q+2)}{D(D+2)} - \frac{q^2}{D^2} = \frac{2q(D-q)}{D^2(D+2)} < \frac{2q}{D^2}. \tag{74}$$

Since $D = d - 1$, the proof is complete. $\square$

Then, the detailed proof of **Lemma 8** is as follows.

*Proof.* By the law of total variance, using Proposition 7 and Lemma 11, we have

$$\text{Var}[E_t] = \mathbb{E}[\text{Var}[E_t|E_{t-1}]] + \text{Var}[\mathbb{E}[E_t|E_{t-1}]] \tag{75}$$

$$= \mathbb{E}[(1 - a'E_{t-1})^2 \text{Var}[\xi_t^2]] + \text{Var}[a'(1 - \mathbb{E}[\xi_t^2])E_{t-1}] \tag{76}$$

$$= \text{Var}[\xi_t^2]\mathbb{E}[(1 - a'E_{t-1})^2] + (a')^2(1 - \mathbb{E}[\xi_t^2])^2\text{Var}[E_{t-1}] \tag{77}$$

$$= \text{Var}[\xi_t^2](\mathbb{E}[(1 - a'E_{t-1})]^2 + \text{Var}[1 - a'E_{t-1}]) + (a')^2(1 - \mathbb{E}[\xi_t^2])^2\text{Var}[E_{t-1}] \tag{78}$$

$$= \text{Var}[\xi_t^2](\mathbb{E}[(1 - a'E_{t-1})]^2 + (a')^2\text{Var}[E_{t-1}]) + (a')^2(1 - \mathbb{E}[\xi_t^2])^2\text{Var}[E_{t-1}] \tag{79}$$

$$= (a')^2(\text{Var}[\xi_t^2] + (1 - \mathbb{E}[\xi_t^2])^2)\text{Var}[E_{t-1}] + \text{Var}[\xi_t^2]\mathbb{E}[(1 - a'E_{t-1})]^2 \tag{80}$$

$$\leq (a')^2(\text{Var}[\xi_t^2] + (1 - \mathbb{E}[\xi_t^2])^2)\text{Var}[E_{t-1}] + \text{Var}[\xi_t^2] \tag{81}$$

$$\leq (a')^2\left(\frac{2q}{(d-1)^2} + \left(1 - \frac{q}{d-1}\right)^2\right)\text{Var}[E_{t-1}] + \frac{2q}{(d-1)^2}. \tag{82}$$

If $d \geq 4$, then $\frac{2q}{(d-1)^2} + (1 - \frac{q}{d-1})^2 = 1 - \frac{q}{(d-1)^2}(2(d-1) - q - 2) \leq 1 - \frac{q}{(d-1)^2}(d - q) \leq 1$. Therefore we have

$$\text{Var}[E_t] \leq (a')^2\text{Var}[E_{t-1}] + \frac{2q}{(d-1)^2}. \tag{83}$$

Letting $a := (a')^2, b := \frac{2q}{(d-1)^2}$, then $\text{Var}[E_t] \leq a\text{Var}[E_{t-1}] + b$ and $0 \leq a < 1$. We have $\text{Var}[E_t] - \frac{b}{1-a} \leq a(\text{Var}[E_{t-1}] - \frac{b}{1-a}) \leq a^2(\text{Var}[E_{t-2}] - \frac{b}{1-a}) \leq \ldots \leq a^t(\text{Var}[E_0] - \frac{b}{1-a})$, hence $\text{Var}[E_t] \leq \frac{b}{1-a} - a^t(\frac{b}{1-a} - \text{Var}[E_0]) = (1 - a^t)\frac{b}{1-a} \leq \frac{b}{1-a} = \frac{1}{1-(a')^2}\frac{2q}{(d-1)^2}$.

Finally, since $\text{Var}[\mathbb{E}_t[E_t]] = \text{Var}[\mathbb{E}[E_t|E_{t-1}]] \leq \text{Var}[E_t]$, the proof is completed. □

The detailed proof of **Lemma 9** is as follows.

*Proof.* Similar to the proof of Proposition 9, letting $a := a'(1 - \frac{q}{d-1})$ and $b := \frac{q}{d-1}$, then $\mathbb{E}[E_t] = (1 - a^t)\frac{b}{1-a}$, and $\frac{1-a'}{1-a} \geq \frac{2}{3}$. Meanwhile, since $\frac{q}{d-1} \leq \frac{L}{L}$, $a \leq (1 - \frac{q}{d-1})^3$. Therefore, if $t \geq \frac{d-1}{q}$, we have

$$a^t \leq (1 - \frac{q}{d-1})^{3\frac{d-1}{q}} \leq \exp(-\frac{q}{d-1})^{3\frac{d-1}{q}} = e^{-3}. \tag{84}$$

Since $\frac{1-a'}{1-a} \geq \frac{2}{3}$ and $a^t \leq e^{-3}$, we have

$$\mathbb{E}[E_t] = (1 - a^t)\frac{b}{1-a} \geq \frac{2}{3}(1 - e^{-3})\frac{1}{1-a'}\frac{q}{d-1} \geq \frac{1}{2}\frac{1}{1-a'}\frac{q}{d-1}. \tag{85}$$

□

The detailed proof of **Lemma 10** is as follows.

*Proof.* By Chebyshev's Inequality, we have

$$\Pr(X < \frac{1}{2}\mathbb{E}[X]) \leq \frac{\text{Var}[X]}{(\frac{1}{2}\mathbb{E}[X])^2} \leq \frac{4\sigma^2}{\mu^2}. \tag{86}$$

Hence

$$\mathbb{E}\left[\frac{1}{X}\right] \leq \frac{1}{B}\Pr\left(X < \frac{1}{2}\mathbb{E}[X]\right) + \frac{1}{\frac{1}{2}\mathbb{E}[X]} \leq \frac{1}{B}\left(\frac{4\sigma^2}{\mu^2} + \frac{2}{\mu}\right) = \frac{1}{\mu B}\left(\frac{4\sigma^2}{\mu} + 2\right). \tag{87}$$

□

### B.5.3 Proof of Theorem 4

As in Section B.5.1, similar to Eq. (47), we define $\{E_t\}_{t=0}^{T-1}$ as follows: $E_0 = 0$, and

$$E_t = a'E_{t-1} + (1 - a'E_{t-1})\zeta_t^2, \tag{88}$$

where $\zeta_t^2 := \sum_{i=1}^q \zeta_{ti}^2$, $\zeta_{ti}^2 := \min\{\xi_{ti}^2, \frac{1}{d-1}\}$ and $\xi_{ti}$ is defined in Proposition 5. Here we use $\zeta_t^2$ instead of $\xi_t^2$ in Eq. (88) to obtain a tighter bound when using McDiarmid's inequality in the proof of Lemma 12.

Here we first give Lemma 12 for the proof of Theorem 4.

**Lemma 12** (Proof in Section B.5.4). $\Pr\left(\frac{1}{T}\sum_{t=0}^{T-1}\mathbb{E}_t[E_t] < 0.1\frac{q}{d}\frac{1}{1-a'}\right) \leq \exp(-0.02T)$.

Then, we provide the proof of Theorem 4 in the following.

**Theorem 4** (History-PRGF, smooth and strongly convex). *Under the same conditions as in Theorem 2 ($f$ is $\tau$-strongly convex), when using the History-PRGF estimator, assuming $d \geq 4$, $\frac{q}{d-1} \leq \frac{L}{\tilde{L}} \leq 1$, $\frac{q}{d} \leq 0.2\frac{L}{\tau}$, and $T \geq 5\frac{d}{q}$, we have*

$$\mathbb{E}[\delta_T] \leq 2\exp(-0.1\frac{q}{d}\frac{\tau}{L}T)\delta_0. \tag{89}$$

*Proof.* Since $E_0 = 0 \leq C_0$, and if $E_{t-1} \leq C_{t-1}$, then

$$E_t = a'E_{t-1} + (1 - a'E_{t-1})\sum_{i=1}^q \zeta_{ti}^2 \tag{90}$$

$$\leq a'E_{t-1} + (1 - a'E_{t-1})\sum_{i=1}^q \xi_{ti}^2 \tag{91}$$

$$= a'E_{t-1} + (1 - a'E_{t-1})\xi_t^2 \tag{92}$$

$$\leq a'C_{t-1} + (1 - a'C_{t-1})\xi_t^2 \tag{93}$$

$$\leq C_t, \tag{94}$$

in which the second inequality is due to that $\xi_t^2 \leq 1$ and the third inequality is due to Eq. (47). Therefore by mathematical induction we have that $\forall t, E_t \leq C_t$.

Then, since $\forall t, E_t \leq C_t$, we have $\frac{1}{T}\sum_{t=0}^{T-1}\mathbb{E}_t[E_t] \leq \frac{1}{T}\sum_{t=0}^{T-1}\mathbb{E}_t[C_t]$. Therefore, by Lemma 12 we have $\Pr\left(\frac{1}{T}\sum_{t=0}^{T-1}\mathbb{E}_t[C_t] < 0.1\frac{q}{d}\frac{1}{1-a'}\right) \leq \exp(-0.02T)$. Let $k_T = \exp\left(\frac{\tau}{L'}\sum_{t=0}^{T-1}\mathbb{E}_t[C_t]\right)$. Since $\frac{1}{1-a'} = \frac{L'}{L}$,

$$\Pr\left(k_T < \exp\left(0.1\frac{q}{d}\frac{\tau}{L}T\right)\right) \leq \exp(-0.02T). \tag{95}$$

Meanwhile, let $\delta_t := f(x_t) - f(x^*)$, Theorem 2 tells us that

$$\mathbb{E}[\delta_T k_T] \leq \delta_0. \tag{96}$$

Noting that $\delta_0 \geq \delta_T$, we have

$$\delta_0 \geq \mathbb{E}[\delta_T k_T] \tag{97}$$

$$\geq \mathbb{E}[\delta_T k_T 1_{k_T \geq \exp\left(0.1\frac{q}{d}\frac{\tau}{L}T\right)}] \tag{98}$$

$$\geq \exp\left(0.1\frac{q}{d}\frac{\tau}{L}T\right)\mathbb{E}[\delta_T 1_{k_T \geq \exp\left(0.1\frac{q}{d}\frac{\tau}{L}T\right)}] \tag{99}$$

$$= \exp\left(0.1\frac{q}{d}\frac{\tau}{L}T\right)\left(\mathbb{E}[\delta_T] - \mathbb{E}[\delta_T 1_{k_T < \exp\left(0.1\frac{q}{d}\frac{\tau}{L}T\right)}]\right) \tag{100}$$

$$\geq \exp\left(0.1\frac{q}{d}\frac{\tau}{L}T\right)\left(\mathbb{E}[\delta_T] - \delta_0\mathbb{E}[1_{k_T < \exp\left(0.1\frac{q}{d}\frac{\tau}{L}T\right)}]\right) \tag{101}$$

$$= \exp\left(0.1\frac{q}{d}\frac{\tau}{L}T\right)\left(\mathbb{E}[\delta_T] - \delta_0\Pr\left(k_T < \exp\left(0.1\frac{q}{d}\frac{\tau}{L}T\right)\right)\right) \tag{102}$$

$$\geq \exp\left(0.1\frac{q}{d}\frac{\tau}{L}T\right)\left(\mathbb{E}[\delta_T] - \delta_0\exp(-0.02T)\right), \tag{103}$$

in which $1_B$ denotes the indicator function of the event $B$ ($1_B = 1$ when $B$ happens and $1_B = 0$ when $B$ does not happen). By rearranging we have

$$\mathbb{E}[\delta_T] \leq \exp\left(-0.1\frac{q}{d}\frac{\tau}{L}T\right)\delta_0 + \exp(-0.02T)\delta_0. \tag{104}$$

When $\frac{q}{d} \leq 0.2\frac{L}{\tau}$, $0.1\frac{q}{d}\frac{\tau}{L}T \leq 0.02T$, and hence $\exp\left(-0.1\frac{q}{d}\frac{\tau}{L}T\right) \geq \exp(-0.02T)$. Therefore, $\mathbb{E}[\delta_T] \leq 2\exp\left(-0.1\frac{q}{d}\frac{\tau}{L}T\right)\delta_0$. The proof of the theorem is completed. $\qquad\square$

### B.5.4 Proof of Lemma 12

In this section, we first give two lemmas for the proof of Lemma 12.

**Lemma 13.** *If $d \geq 4$, then $\mathbb{E}[\zeta_t^2] \geq 0.3\frac{q}{d-1}$.*

*Proof.* We note that the distribution of $u_i$ is the uniform distribution from the unit sphere in the $(d-1)$-dimensional subspace $A$. Since $\xi_{ti} := \overline{\nabla f(x_t)_A}^\top u_i$, $\xi_{ti}$ is indeed the inner product between one fixed unit vector and one uniformly random sampled unit vector. Therefore, its distribution is the same as $z_1$, where $(z_1, z_2, \ldots, z_{d-1})$ are uniformly sampled from $\mathbb{S}^{d-2}$, i.e. the unit sphere in $\mathbb{R}^{d-1}$. Now it suffices to prove that $\mathbb{E}[\min\{z_1^2, \frac{1}{d-1}\}] \geq \frac{0.3}{d-1}$.

Let $p(\cdot)$ denote the probability density function of $z_1$. For convenience let $D := d - 1$. We have $p(0) = \frac{S_{D-2}}{S_{D-1}}$, where $S_{D-1}$ denotes the surface area of $\mathbb{S}_{D-1}$. Since $S_{D-1} = \frac{2\pi^{\frac{D}{2}}}{\Gamma(\frac{D}{2})}$ where $\Gamma(\cdot)$ is the Gamma function, and by [11] we have $\frac{\Gamma(\frac{D}{2})}{\Gamma(\frac{D-1}{2})} \leq \sqrt{\frac{D-1}{2}}$, we have $p(0) \leq \sqrt{\frac{D-1}{2\pi}} \leq \sqrt{\frac{d-1}{2\pi}}$. Meanwhile, we have $p(x) = p(0) \cdot \frac{(\sqrt{1-x^2})^{D-2}}{\sqrt{1-x^2}} = p(0) \cdot (\sqrt{1-x^2})^{D-3}$. If $d \geq 4$, then $D \geq 3$, and we have $\forall x \in [-1, 1], p(0) \geq p(x)$. Therefore,

$$\Pr\left(z_1^2 \geq \frac{1}{d-1}\right) = 1 - \Pr\left(|z_1| < \frac{1}{\sqrt{d-1}}\right) \geq 1 - \frac{2}{\sqrt{d-1}}p(0) = 1 - \sqrt{\frac{2}{\pi}} \geq 0.2. \tag{105}$$

Similarly we have

$$\Pr\left(z_1^2 \geq \frac{0.25}{d-1}\right) = 1 - \Pr\left(|z_1| < \frac{0.5}{\sqrt{d-1}}\right) \geq 1 - \frac{1}{\sqrt{d-1}}p(0) = 1 - \sqrt{\frac{1}{2\pi}} \geq 0.6. \tag{106}$$

Let ${z_1'}^2 := \min\{z_1^2, \frac{1}{d-1}\}$. Then $\Pr\left({z_1'}^2 \geq \frac{1}{d-1}\right) \geq 0.2$ and $\Pr\left({z_1'}^2 \geq \frac{0.25}{d-1}\right) \geq 0.6$. Then

$$\mathbb{E}[{z_1'}^2] \geq \frac{1}{d-1}\Pr\left({z_1'}^2 \geq \frac{1}{d-1}\right) + \frac{0.25}{d-1}\Pr\left(\frac{1}{d-1} \geq {z_1'}^2 \geq \frac{0.25}{d-1}\right) \tag{107}$$

$$= \frac{0.75}{d-1}\Pr\left({z_1'}^2 \geq \frac{1}{d-1}\right) + \frac{0.25}{d-1}\Pr\left({z_1'}^2 \geq \frac{0.25}{d-1}\right) \tag{108}$$

$$\geq \frac{0.3}{d-1}. \tag{109}$$

Hence $\mathbb{E}[\zeta_{ti}^2] \geq \frac{0.3}{d-1}$. By the definition of $\zeta_t^2$ the lemma is proved. $\qquad\square$

**Lemma 14.** *If $\frac{q}{d-1} \leq \frac{L}{L}$, $T \geq 5\frac{d}{q}$, then $\frac{1}{T}\sum_{t=0}^{T-1}\mathbb{E}[E_t] \geq 0.2\frac{\frac{q}{d-1}}{1-a'}$.*

*Proof.* By Eq. (88) and Lemma 13, we have $\mathbb{E}_t[E_t] \geq \left(1 - 0.3\frac{q}{d-1}\right)a'E_{t-1} + 0.3\frac{q}{d-1}$. Taking expectation to both sides, we have

$$\mathbb{E}[E_t] \geq \left(1 - 0.3\frac{q}{d-1}\right)a'\mathbb{E}[E_{t-1}] + 0.3\frac{q}{d-1}. \tag{110}$$

Let $a := \left(1 - 0.3\frac{q}{d-1}\right)a'$, $b := 0.3\frac{q}{d-1}$, then $\mathbb{E}[E_t] \geq a\mathbb{E}[E_{t-1}] + b$ and $0 \leq a < 1$. We have $\mathbb{E}[E_t] - \frac{b}{1-a} \geq a(\mathbb{E}[E_{t-1}] - \frac{b}{1-a}) \geq a^2(\mathbb{E}[E_{t-2}] - \frac{b}{1-a}) \geq \ldots \geq a^t(\mathbb{E}[E_0] - \frac{b}{1-a})$, hence

$\mathbb{E}[E_t] \geq \frac{b}{1-a} - a^t(\frac{b}{1-a} - \mathbb{E}[E_0]) = (1-a^t)\frac{b}{1-a}$. Hence we have

$$\frac{1}{T}\sum_{t=0}^{T-1}\mathbb{E}[E_t] \geq \frac{b}{1-a}\left(1 - \frac{1-a^T}{(1-a)T}\right) \geq \frac{b}{1-a}\left(1 - \frac{1}{(1-a)T}\right). \tag{111}$$

Since $1 - a = 1 - (1 - 0.3\frac{q}{d-1})(1-\frac{L}{\bar{L}})^2 \leq 1 - (1 - 0.3\frac{L}{\bar{L}})(1-\frac{L}{\bar{L}})^2$, noting that

$$\frac{1 - (1-\frac{L}{\bar{L}})^2}{1 - (1 - 0.3\frac{L}{\bar{L}})(1-\frac{L}{\bar{L}})^2} = \frac{\frac{L}{\bar{L}} + \frac{L}{\bar{L}}(1-\frac{L}{\bar{L}})}{\frac{L}{\bar{L}} + \frac{L}{\bar{L}}(1-\frac{L}{\bar{L}}) + 0.3\frac{L}{\bar{L}}(1-\frac{L}{\bar{L}})^2} \geq \frac{2}{2.3}, \tag{112}$$

we have $\frac{1-a'}{1-a} \geq \frac{2}{2.3}$. Letting $T \geq 5\frac{d}{q}$, then $T \geq 5\frac{q}{\frac{q}{d}} \geq 5\frac{1}{\frac{L}{\bar{L}}} \geq 5\frac{1}{1-\sqrt{a'}} \geq 5\frac{1}{1-a}$. By Eq. (111) we have

$$\frac{1}{T}\sum_{t=0}^{T-1}\mathbb{E}[E_t] \geq \frac{2}{2.3}\frac{b}{1-a'}\frac{4}{5} = \frac{2.4}{11.5}\frac{\frac{q}{d-1}}{1-a'} \geq 0.2\frac{\frac{q}{d-1}}{1-a'}. \tag{113}$$

$\square$

Finally, the detailed proof of **Lemma 12** is as follows.

*Proof.* Let $\overline{E} := \frac{1}{T}\sum_{t=0}^{T-1}\mathbb{E}_t[E_t]$. We note that $\{\zeta_1^2, \zeta_2^2, \ldots, \zeta_{T-1}^2\}$ are independent, and $\overline{E}$ is a function of them. Now suppose that the value of $\zeta_t^2$ is changed by $\Delta\zeta_t^2$, while the value of $\{\zeta_1^2, \ldots, \zeta_{t-1}^2, \zeta_{t+1}^2, \ldots, \zeta_{T-1}^2\}$ are unchanged. Then

$$\Delta E_s = 0, \qquad\qquad\qquad\qquad\qquad 0 \leq s \leq t-1; \tag{114}$$
$$\Delta E_s = (1 - a'E_{t-1})\Delta\zeta_t^2 \leq \Delta\zeta_t^2, \qquad\qquad s = t; \tag{115}$$
$$\Delta E_s = (1 - \zeta_s^2)a'\Delta E_{s-1} \leq a'\Delta E_{s-1}, \qquad\qquad s \geq t+1. \tag{116}$$

Therefore, for $s \geq t$, $\Delta E_s \leq (a')^{s-t}\Delta E_t \leq (a')^{s-t}\Delta\zeta_t^2$; for $s < t$, $\Delta E_s = 0$. By Eq. (88), $\mathbb{E}_s[E_s] = a'(1 - \mathbb{E}[\zeta_s^2])E_{s-1} + \mathbb{E}[\zeta_s^2]$, so $\Delta\mathbb{E}_s[E_s] \leq a'\Delta E_{s-1} \leq \Delta E_{s-1}$. Hence

$$\Delta\overline{E} = \frac{1}{T}\sum_{s=0}^{T-1}\Delta\mathbb{E}_s[E_s] \leq \frac{1}{T}\sum_{s=t+1}^{T-1}(a')^{s-1-t}\Delta\zeta_t^2 \leq \frac{1}{T}\frac{1}{1-a'}\Delta\zeta_t^2. \tag{117}$$

Since $\zeta_{ti}^2 := \min\{\xi_{ti}^2, \frac{1}{d-1}\}$, $0 \leq \zeta_t^2 \leq \frac{q}{d-1}$. Therefore $\Delta\overline{E} \leq \frac{1}{T}\frac{1}{1-a'}\frac{q}{d-1}$. Therefore, by McDiarmid's inequality, we have

$$\Pr(\overline{E} < \mathbb{E}[\overline{E}] - \epsilon) \leq \exp\left(-\frac{2\epsilon^2}{T\left(\frac{1}{T}\frac{1}{1-a'}\frac{q}{d-1}\right)^2}\right) \leq \exp\left(-2T\left(\epsilon(1-a')\frac{d-1}{q}\right)^2\right). \tag{118}$$

Let $\epsilon = 0.1\frac{\frac{q}{d-1}}{1-a'}$, we have $\Pr(\overline{E} < \mathbb{E}[\overline{E}] - 0.1\frac{\frac{q}{d-1}}{1-a'}) \leq \exp(-0.02T)$. By Lemma 14, $\mathbb{E}[\overline{E}] \geq 0.2\frac{\frac{q}{d-1}}{1-a'}$. Noting that $\frac{q}{d} \leq \frac{q}{d-1}$, the proof is completed. $\square$

# C  Supplemental materials for Section 4

## C.1  Proof of Theorem 5

**Theorem 5.** *In Algorithm 2, if $\theta_t$ is $\mathcal{F}_{t-1}$-measurable, we have*

$$\mathbb{E}\left[(f(x_T) - f(x^*))\left(1 + \frac{\sqrt{\gamma_0}}{2}\sum_{t=0}^{T-1}\sqrt{\theta_t}\right)^2\right] \leq f(x_0) - f(x^*) + \frac{\gamma_0}{2}\|x_0 - x^*\|^2. \tag{119}$$

To help understand the design of Algorithm 2, we present the proof sketch below, where the part which is the same as the original proof in [12] is omitted.

*Sketch of the proof.* Since $x_{t+1} = y_t - \frac{1}{L}g_1(y_t)$ and $g_1(y_t) = \nabla f(y_t)^\top v_t \cdot v_t$, by Lemma 1,

$$\mathbb{E}_t[f(x_{t+1})] \le f(y_t) - \frac{\mathbb{E}_t\left[\left(\nabla f(y_t)^\top v_t\right)^2\right]}{2L'} \le f(y_t) - \frac{\mathbb{E}_t\left[\left(\nabla f(y_t)^\top v_t\right)^2\right]}{2\hat{L}}. \tag{120}$$

Define $\rho_t := \frac{\gamma_t}{2}\|m_t - x^*\|^2 + f(x_t) - f(x^*)$. The same as in original proof, we have

$$\rho_{t+1} = \frac{\gamma_{t+1}}{2}\|m_t - x^*\|^2 - \alpha_t g_2(y_t)^\top(m_t - x^*) + \frac{\theta_t}{2}\|g_2(y_t)\|^2 + f(x_{t+1}) - f(x^*). \tag{121}$$

Then we derive $\mathbb{E}_t[\rho_{t+1}]$. We mentioned in Remark 2 that the notation $\mathbb{E}_t[\cdot]$ means the conditional expectation $\mathbb{E}[\cdot|\mathcal{F}_{t-1}]$, where $\mathcal{F}_{t-1}$ is a sub $\sigma$-algebra modelling the historical information, and we require that $\mathcal{F}_{t-1}$ includes all the randomness before iteration $t$. Therefore, $\gamma_t$ and $m_t$ are $\mathcal{F}_{t-1}$-measurable. The assumption in Theorem 5 requires that $\theta_t$ is $\mathcal{F}_{t-1}$-measurable. Since $\alpha_t$ is determined by $\gamma_t$ and $\theta_t$ (through a Borel function), $\alpha_t$ is also $\mathcal{F}_{t-1}$-measurable. We have

$$\mathbb{E}_t[\rho_{t+1}]$$
$$= \frac{\gamma_{t+1}}{2}\|m_t - x^*\|^2 - \alpha_t\mathbb{E}_t[g_2(y_t)]^\top(m_t - x^*) + \frac{\theta_t}{2}\mathbb{E}_t[\|g_2(y_t)\|^2] + \mathbb{E}_t[f(x_{t+1})] - f(x^*) \tag{122}$$

$$= \frac{\gamma_{t+1}}{2}\|m_t - x^*\|^2 - \alpha_t\nabla f(y_t)^\top(m_t - x^*) + \frac{\theta_t}{2}\mathbb{E}_t[\|g_2(y_t)\|^2] + \mathbb{E}_t[f(x_{t+1})] - f(x^*) \tag{123}$$

$$\le \frac{\gamma_{t+1}}{2}\|m_t - x^*\|^2 - \alpha_t\nabla f(y_t)^\top(m_t - x^*) + \frac{\mathbb{E}_t\left[\left(\nabla f(y_t)^\top v_t\right)^2\right]}{2\hat{L}} + \mathbb{E}_t[f(x_{t+1})] - f(x^*) \tag{124}$$

$$\le \frac{\gamma_{t+1}}{2}\|m_t - x^*\|^2 - \alpha_t\nabla f(y_t)^\top(m_t - x^*) + f(y_t) - f(x^*) \tag{125}$$

$$\le (1 - \alpha_t)\rho_t, \tag{126}$$

where the first equality is because $m_t$, $\alpha_t$ and $\theta_t$ are $\mathcal{F}_{t-1}$-measurable, the second equality is because $\mathbb{E}_t[g_2(y_t)] = \nabla f(y_t)$, the first inequality is because $\theta_t \le \frac{\mathbb{E}_t\left[\left(\nabla f(y_t)^\top v_t\right)^2\right]}{\hat{L}\cdot\mathbb{E}_t[\|g_2(y_t)\|^2]}$, the second inequality is because of Eq. (120), and the last inequality is the same as in original proof. By the similar reasoning to the proof of Theorem 2, we have $\mathbb{E}\left[\frac{\rho_T}{\prod_{t=0}^{T-1}(1-\alpha_t)}\right] \le \rho_0$. By the original proof, $\prod_{t=0}^{T-1}(1-\alpha_t) \le \frac{1}{\left(1+\frac{\sqrt{\gamma_0}}{2}\sum_{t=0}^{T-1}\sqrt{\theta_t}\right)^2}$, completing the proof. $\qquad\square$

From the proof sketch, we see that

- The requirement that $\mathbb{E}_t[g_2(y_t)] = \nabla f(y_t)$ is to ensure that Eq. (123) holds.

- The constraint on $\theta_t$ in Line 3 of Algorithm 2 is to ensure that Eq. (124) holds.

- The choice of $g_1(y_t)$ ($g_1(y_t) = \nabla f(y_t)^\top v_t \cdot v_t$) and update of $x_t$ ($x_{t+1} = y_t - \frac{1}{L}g_1(y_t)$) is the same as in Algorithm 1, i.e. the greedy descent framework. This is since Eq. (125) requires that $\mathbb{E}_t[f(x_{t+1})]$ decreases as much as possible from $f(y_t)$.

- From Eq. (121) to Eq. (122), we require $\mathbb{E}_t[\alpha_t g_2(y_t)^\top(m_t - x)] = \alpha_t\mathbb{E}_t[g_2(y_t)]^\top(m_t - x)$ and $\mathbb{E}_t[\theta_t\|g_2(y_t)\|^2] = \theta_t\mathbb{E}_t[\|g_2(y_t)\|^2]$. Therefore, to make the two above identities hold, by the property of "pulling out known factors" in taking conditional expectation, we require that $m_t$, $\alpha_t$ and $\theta_t$ are $\mathcal{F}_{t-1}$-measurable. Since we make sure in Remark 2 that $\mathcal{F}_{t-1}$ always includes all the randomness before iteration $t$, and $\alpha_t$ is determined by $\gamma_t$ and $\theta_t$, it suffices to let $\theta_t$ be $\mathcal{F}_{t-1}$-measurable. We note that "being $\mathcal{F}_{t-1}$-measurable" means "being determined by the history, i.e. fixed given the history".

Now we present the complete proof of Theorem 5.

*Proof.* Since $x_{t+1} = y_t - \frac{1}{L}g_1(y_t)$ and $g_1(y_t) = \nabla f(y_t)^\top v_t \cdot v_t$, by Lemma 1,

$$\mathbb{E}_t[f(x_{t+1})] \le f(y_t) - \frac{\mathbb{E}_t\left[\left(\nabla f(y_t)^\top v_t\right)^2\right]}{2L'} \tag{127}$$

$$\le f(y_t) - \frac{\mathbb{E}_t\left[\left(\nabla f(y_t)^\top v_t\right)^2\right]}{2\hat{L}}. \tag{128}$$

For an arbitrary fixed $x$, define $\rho_t(x) := \frac{\gamma_t}{2}\|m_t - x\|^2 + f(x_t) - f(x)$. Then

$$\rho_{t+1}(x) = \frac{\gamma_{t+1}}{2}\|m_{t+1} - x\|^2 + f(x_{t+1}) - f(x) \tag{129}$$

$$= \frac{\gamma_{t+1}}{2}\|m_t - x\|^2 - \frac{\gamma_{t+1}\theta_t}{\alpha_t}g_2(y_t)^\top(m_t - x) + \frac{\gamma_{t+1}\theta_t^2}{2\alpha_t^2}\|g_2(y_t)\|^2 + f(x_{t+1}) - f(x) \tag{130}$$

$$= \frac{\gamma_{t+1}}{2}\|m_t - x\|^2 - \alpha_t g_2(y_t)^\top(m_t - x) + \frac{\theta_t}{2}\|g_2(y_t)\|^2 + f(x_{t+1}) - f(x). \tag{131}$$

We make sure in Remark 2 that $\mathcal{F}_{t-1}$ always includes all the randomness before iteration $t$. Therefore, $\gamma_t$ and $m_t$ are $\mathcal{F}_{t-1}$-measurable. The assumption in Theorem 5 requires that $\theta_t$ is $\mathcal{F}_{t-1}$-measurable. Since $\alpha_t$ is determined by $\gamma_t$ and $\theta_t$ (through a Borel function), $\alpha_t$ is also $\mathcal{F}_{t-1}$-measurable. Since $m_t$, $\alpha_t$ and $\theta_t$ are $\mathcal{F}_{t-1}$-measurable, we have $\mathbb{E}_t[\alpha_t g_2(y_t)^\top(m_t - x)] = \alpha_t \mathbb{E}_t[g_2(y_t)]^\top(m_t - x)$ and $\mathbb{E}_t[\theta_t\|g_2(y_t)\|^2] = \theta_t \mathbb{E}_t[\|g_2(y_t)\|^2]$. Hence

$$\mathbb{E}_t[\rho_{t+1}(x)] = \frac{\gamma_{t+1}}{2}\|m_t - x^*\|^2 - \alpha_t \mathbb{E}_t[g_2(y_t)]^\top(m_t - x^*) + \frac{\theta_t}{2}\mathbb{E}_t[\|g_2(y_t)\|^2] + \mathbb{E}_t[f(x_{t+1})] - f(x^*) \tag{132}$$

$$= \frac{\gamma_{t+1}}{2}\|m_t - x\|^2 - \alpha_t \nabla f(y_t)^\top(m_t - x) + \frac{\theta_t}{2}\mathbb{E}_t[\|g_2(y_t)\|^2] + \mathbb{E}_t[f(x_{t+1})] - f(x) \tag{133}$$

$$\le \frac{\gamma_{t+1}}{2}\|m_t - x\|^2 - \alpha_t \nabla f(y_t)^\top(m_t - x) + \frac{\mathbb{E}_t\left[\left(\nabla f(y_t)^\top v_t\right)^2\right]}{2\hat{L}} + \mathbb{E}_t[f(x_{t+1})] - f(x) \tag{134}$$

$$\le \frac{\gamma_{t+1}}{2}\|m_t - x\|^2 - \alpha_t \nabla f(y_t)^\top(m_t - x) + f(y_t) - f(x) \tag{135}$$

$$= \frac{\gamma_{t+1}}{2}\|m_t - x\|^2 - \nabla f(y_t)^\top(\alpha_t m_t - \alpha_t x) + f(y_t) - f(x) \tag{136}$$

$$= \frac{\gamma_{t+1}}{2}\|m_t - x\|^2 + \nabla f(y_t)^\top(-y_t + (1 - \alpha_t)x_t + \alpha_t x) + f(y_t) - f(x) \tag{137}$$

$$\le \frac{\gamma_{t+1}}{2}\|m_t - x\|^2 + f((1 - \alpha_t)x_t + \alpha_t x) - f(x) \tag{138}$$

$$\le \frac{\gamma_{t+1}}{2}\|m_t - x\|^2 + (1 - \alpha_t)f(x_t) - (1 - \alpha_t)f(x) \tag{139}$$

$$= (1 - \alpha_t)\left(\frac{\gamma_t}{2}\|m_t - x\|^2 + f(x_t) - f(x)\right) \tag{140}$$

$$= (1 - \alpha_t)\rho_t(x). \tag{141}$$

Therefore,

$$\rho_0(x) = \mathbb{E}[\rho_0(x)] \ge \mathbb{E}\left[\frac{\mathbb{E}_0[\rho_1(x)]}{1 - \alpha_0}\right] = \mathbb{E}\left[\mathbb{E}_0\left[\frac{\rho_1(x)}{1 - \alpha_0}\right]\right] = \mathbb{E}\left[\frac{\rho_1(x)}{1 - \alpha_0}\right]$$

$$\ge \mathbb{E}\left[\frac{\mathbb{E}_1[\rho_2(x)]}{(1 - \alpha_0)(1 - \alpha_1)}\right] = \mathbb{E}\left[\mathbb{E}_1\left[\frac{\rho_2(x)}{(1 - \alpha_0)(1 - \alpha_1)}\right]\right] = \mathbb{E}\left[\frac{\rho_2(x)}{(1 - \alpha_0)(1 - \alpha_1)}\right]$$

$$\ge \dots$$

$$\ge \mathbb{E}\left[\frac{\rho_T(x)}{\prod_{t=0}^{T-1}(1 - \alpha_t)}\right].$$

We have $\rho_T(x) \geq f(x_T) - f(x)$. To prove the theorem, let $x = x^*$. The remaining is to give an upper bound of $\prod_{t=0}^{T-1}(1 - \alpha_t)$. Let $\psi_k := \prod_{t=0}^{k-1}(1 - \alpha_t)$ and $a_k := \frac{1}{\sqrt{\psi_k}}$, we have

$$a_{k+1} - a_k = \frac{1}{\sqrt{\psi_{k+1}}} - \frac{1}{\sqrt{\psi_k}} = \frac{\sqrt{\psi_k} - \sqrt{\psi_{k+1}}}{\sqrt{\psi_k \psi_{k+1}}} = \frac{\psi_k - \psi_{k+1}}{\sqrt{\psi_k \psi_{k+1}}(\sqrt{\psi_k} + \sqrt{\psi_{k+1}})} \tag{142}$$

$$\geq \frac{\psi_k - \psi_{k+1}}{\sqrt{\psi_k \psi_{k+1}}(2\sqrt{\psi_k})} \tag{143}$$

$$= \frac{\psi_k - (1 - \alpha_k)\psi_k}{2\psi_k \sqrt{\psi_{k+1}}} = \frac{\alpha_k}{2\sqrt{\psi_{k+1}}} = \frac{\sqrt{\gamma_{k+1}\theta_k}}{2\sqrt{\psi_{k+1}}} = \frac{\sqrt{\theta_k}}{2}\sqrt{\frac{\gamma_{k+1}}{\psi_{k+1}}} \tag{144}$$

$$= \frac{\sqrt{\gamma_0 \theta_k}}{2}. \tag{145}$$

Since $\psi_0 = 1$, $a_0 = 1$. Hence $a_T \geq 1 + \frac{\sqrt{\gamma_0}}{2}\sum_{t=0}^{T-1}\sqrt{\theta_t}$. Therefore, $\psi_T \leq \frac{1}{\left(1 + \frac{\sqrt{\gamma_0}}{2}\sum_{t=0}^{T-1}\sqrt{\theta_t}\right)^2}$.
The proof is completed. $\qquad\square$

### C.2 Construction of $g_2(y_t)$

We first note that in PARS, the specification of $\mathcal{F}_{t-1}$ is similar to that in Example 2. That is, we suppose that $p_t$ is determined before sampling $\{u_1, u_2, \ldots, u_q\}$, but it could depend on extra randomness in iteration $t$. We let $\mathcal{F}_{t-1}$ also includes the extra randomness of $p_t$ in iteration $t$ (not including the randomness of $\{u_1, u_2, \ldots, u_q\}$) besides the randomness before iteration $t$. Meanwhile, we note that the assumption in Theorem 5 requires that $\theta_t$ is $\mathcal{F}_{t-1}$-measurable, and this is satisfied if the algorithm to find $\theta_t$ in Algorithm 2 is deterministic given randomness in $\mathcal{F}_{t-1}$ (does not use $\{u_1, u_2, \ldots, u_q\}$ in iteration $t$). Since $\mathcal{F}_{t-1}$ includes randomness before iteration $t$, if $\theta_t$ is $\mathcal{F}_{t-1}$-measurable, we can show that $y_t$ is $\mathcal{F}_{t-1}$-measurable.

We also note that in Section 4 and Appendix C, we let $D_t := \left(\overline{\nabla f(y_t)}^{\top} p_t\right)^2$, which is different from the previous definition $D_t := \left(\overline{\nabla f(x_t)}^{\top} p_t\right)^2$ in Section 3 and Appendix B. This is because in ARS-based algorithms, we care about gradient estimation at $y_t$ instead of that at $x_t$.

In Algorithm 2, we need to construct $g_2(y_t)$ as an unbiased estimator of $\nabla f(y_t)$ satisfying $\mathbb{E}_t[g_2(y_t)] = \nabla f(y_t)$. Since Theorem 5 tells us that a larger $\theta_t$ could potentially accelerate convergence, by Line 3 of Algorithm 2, we want to make $\mathbb{E}_t[\|g_2(y_t)\|^2]$ as small as possible. To save queries, we hope that we can reuse the queries $\nabla f(y_t)^{\top} p_t$ and $\{\nabla f(y_t)^{\top} u_i\}_{i=1}^{q}$ used in the process of constructing $g_1(y_t)$.

Here we adopt the construction process in Section B.3.3, and leave the discussion of alternative ways in Section C.2.1. We note that if we let $H$ be the $(d-1)$-dimensional subspace perpendicular to $p_t$, then

$$\nabla f(y_t) = \nabla f(y_t)^{\top} p_t \cdot p_t + (\mathbf{I} - p_t p_t^{\top})\nabla f(y_t) = \nabla f(y_t)^{\top} p_t \cdot p_t + \nabla f(y_t)_H. \tag{146}$$

Therefore, we need to obtain an unbiased estimator of $\nabla f(y_t)_H$. This is straightforward since we can utilize $\{u_i\}_{i=1}^{q}$ which is uniformly sampled from the $(d-1)$-dimensional space $H$.

**Proposition 10.** *For any* $1 \leq i \leq q$, $\mathbb{E}_t[\nabla f(y_t)^{\top} u_i \cdot u_i] = \frac{1}{d-1}\nabla f(y_t)_H$.

*Proof.* We have $\mathbb{E}_t[u_i u_i^{\top}] = \frac{\mathbf{I} - p_t p_t^{\top}}{d-1}$ (See Section A.2 in [4] for the proof.). Therefore,

$$\mathbb{E}_t[\nabla f(y_t)^{\top} u_i \cdot u_i] = \frac{\mathbf{I} - p_t p_t^{\top}}{d-1}\nabla f(y_t) = \frac{1}{d-1}\nabla f(y_t)_H. \tag{147}$$

$\qquad\square$

Therefore,

$$g_2(y_t) = \nabla f(y_t)^{\top} p_t \cdot p_t + \frac{d-1}{q}\sum_{i=1}^{q}\nabla f(y_t)^{\top} u_i \cdot u_i \tag{148}$$

satisfies that $\mathbb{E}_t[g_2(y_t)] = \nabla f(y_t)$. Then

$$\mathbb{E}_t[\|g_2(y_t)\|^2] = \|\nabla f(y_t)\|^2 \mathbb{E}_t\left[\left\|\overline{\nabla f(y_t)}^\top p_t \cdot p_t + \frac{d-1}{q}\sum_{i=1}^q \overline{\nabla f(y_t)}^\top u_i \cdot u_i\right\|^2\right] \tag{149}$$

$$= \|\nabla f(y_t)\|^2 \left(\left(\overline{\nabla f(y_t)}^\top p_t\right)^2 + \frac{(d-1)^2}{q^2}\sum_{i=1}^q \mathbb{E}_t\left[\left(\overline{\nabla f(y_t)}^\top u_i\right)^2\right]\right) \tag{150}$$

$$= \|\nabla f(y_t)\|^2 \left(D_t + \frac{d-1}{q}(1 - D_t)\right), \tag{151}$$

where the last equality is due to the fact that $\mathbb{E}_t[u_i u_i^\top] = \frac{\mathbf{I} - p_t p_t^\top}{d-1}$ (hence $\mathbb{E}_t\left[\left(\overline{\nabla f(y_t)}^\top u_i\right)^2\right] = \frac{1-D_t}{d-1}$). Meanwhile, if we adopt an RGF estimator as $g_2(y_t)$, then $\mathbb{E}_t[\|g_2(y_t)\|^2] = \frac{d}{q}\|\nabla f(y_t)\|^2$. Noting that $D_t + \frac{d-1}{q}(1 - D_t) < \frac{d}{q}$, our proposed unbiased estimator results in a smaller $\mathbb{E}_t[\|g_2(y_t)\|^2]$ especially when $D_t$ is closed to 1, since it utilizes the prior information.

Finally, using $g_2(y_t)$ in Eq. (148), when calculating the following expression which appears in Line 3 of Algorithm 2, the term $\|\nabla f(y_t)\|^2$ would be cancelled:

$$\frac{\mathbb{E}_t\left[\left(\nabla f(y_t)^\top v_t\right)^2\right]}{\hat{L} \cdot \mathbb{E}_t[\|g_2(y_t)\|^2]} = \frac{\mathbb{E}_t\left[\left(\overline{\nabla f(y_t)}^\top v_t\right)^2\right]}{\hat{L}\left(D_t + \frac{d-1}{q}(1 - D_t)\right)} = \frac{D_t + \frac{q}{d-1}(1 - D_t)}{\hat{L}\left(D_t + \frac{d-1}{q}(1 - D_t)\right)}, \tag{152}$$

where the last equality is due to Lemma 2.

### C.2.1 Alternative way to construct $g_2(y_t)$

Instead of using the orthogonalization process in Section B.3.3, when constructing $g_1(y_t)$ as the PRGF estimator, we can also first sample $q$ orthonormal vectors $\{u_i\}_{i=1}^q$ uniformly from $\mathbb{S}_{d-1}$, and then let $p_t$ be orthogonal to them with $\{u_i\}_{i=1}^q$ unchanged. Then we can construct $g_2(y_t)$ using this set of $\{\nabla f(y_t)^\top u_i\}_{i=1}^q$ and $\nabla f(y_t)^\top p_t$.

**Example 3** (RGF). *Since $\{u_i\}_{i=1}^q$ are uniformly sampled from $\mathbb{S}_{d-1}$, we can directly use them to construct an unbiased estimator of $\nabla f(y_t)$. We let $g_2(y_t) = \frac{d}{q}\sum_{i=1}^q \nabla f(y_t)^\top u_i \cdot u_i$. We show that it is an unbiased estimator of $\nabla f(y_t)$, and $\mathbb{E}_t[\|g_2(y_t)\|^2] = \frac{d}{q}\|\nabla f(y_t)\|^2$.*

*Proof.* In Section B.3.2 we show that $\mathbb{E}[u_i u_i^\top] = \frac{\mathbf{I}}{d}$. Therefore

$$\mathbb{E}_t[g_2(y_t)] = \frac{d}{q}\sum_{i=1}^q \mathbb{E}_t[u_i u_i^\top]\nabla f(y_t) = \frac{d}{q}\sum_{i=1}^q \frac{1}{d}\nabla f(y_t) = \nabla f(y_t).$$

$$\mathbb{E}_t[\|g_2(y_t)\|^2] = \frac{d^2}{q^2}\sum_{i=1}^q \mathbb{E}_t[(\nabla f(y_t)^\top u_i)^2] = \frac{d^2}{q^2}\sum_{i=1}^q \nabla f(y_t)^\top \mathbb{E}_t[u_i u_i^\top]\nabla f(y_t)$$

$$= \frac{d^2}{q^2}\sum_{i=1}^q \frac{1}{d}\|\nabla f(y_t)\|^2 = \frac{d}{q}\|\nabla f(y_t)\|^2.$$

$\square$

*We see that $\mathbb{E}_t[\|g_2(y_t)\|^2]$ here is larger than Eq. (151).*

**Example 4** (Variance reduced RGF). *To reduce the variance of RGF estimator, we could use $p_t$ to construct a control variate. Here we use $p_t$ to refer to the original $p_t^{ori}$ before orthogonalization so that it is fixed w.r.t. randomness of $\{u_1, \ldots, u_q\}$ (then it requires one additional query to obtain $\nabla f(y_t)^\top p_t^{ori}$). Specifically, we can let $g_2(y_t) = \frac{d}{q}\sum_{i=1}^q (\nabla f(y_t)^\top u_i \cdot u_i - (\nabla f(y_t)^\top p_t \cdot p_t)^\top u_i \cdot u_i) + \nabla f(y_t)^\top p_t \cdot p_t$. We show that it is unbiased, and $\mathbb{E}_t[\|g_2(y_t)\|^2] = \|\nabla f(y_t)\|^2 \left(D_t + \frac{d}{q}(1 - D_t)\right)$.*

*Proof.*

$$\mathbb{E}_t[(\nabla f(y_t)^\top p_t \cdot p_t)^\top u_i \cdot u_i] = \mathbb{E}_t[u_i u_i^\top] \nabla f(y_t)^\top p_t \cdot p_t = \frac{1}{d} \nabla f(y_t)^\top p_t \cdot p_t.$$

Therefore,

$$\mathbb{E}_t[g_2(y_t)] = \mathbb{E}_t \left[ \frac{d}{q} \sum_{i=1}^q \nabla f(y_t)^\top u_i \cdot u_i \right] = \nabla f(y_t).$$

Let $\nabla f(y_t)_H := \nabla f(y_t) - \nabla f(y_t)^\top p_t \cdot p_t$. We define that $\mathrm{Var}[x]$ for a stochastic vector $x$ is such that $\mathrm{Var}[x] = \sum_i \mathrm{Var}[x_i]$. Then for any stochastic vector $x$, $\mathbb{E}[\|x\|^2] = \|\mathbb{E}[x]\|^2 + \mathrm{Var}[x]$. We have $\mathrm{Var}_t[g_2(y_t)] = \mathrm{Var}_t \left[ \frac{d}{q} \sum_{i=1}^q \nabla f(y_t)_H^\top u_i \cdot u_i \right]$. Let $g_2'(y_t) := \frac{d}{q} \sum_{i=1}^q \nabla f(y_t)_H^\top u_i \cdot u_i$. Then $\mathbb{E}_t[g_2'(y_t)] = \nabla f(y_t)_H$, $\mathbb{E}_t[\|g_2'(y_t)\|^2] = \frac{d}{q}\|\nabla f(y_t)_H\|^2$. Therefore, $\mathrm{Var}_t[g_2(y_t)] = \mathbb{E}_t[\|g_2'(y_t)\|^2] - \|\mathbb{E}_t[g_2'(y_t)]\|^2 = \left(\frac{d}{q} - 1\right)\|\nabla f(y_t)_H\|^2 = (1 - D_t)\left(\frac{d}{q} - 1\right)\|\nabla f(y_t)\|^2$. Hence,

$$\mathbb{E}_t[\|g_2(y_t)\|^2] = \|\mathbb{E}_t[g_2(y_t)]\|^2 + \mathrm{Var}_t[g_2(y_t)] = \left(1 + (1 - D_t)\left(\frac{d}{q} - 1\right)\right)\|\nabla f(y_t)\|^2$$

$$= \left(D_t + \frac{d}{q}(1 - D_t)\right)\|\nabla f(y_t)\|^2.$$

$\square$

*We see that $\mathbb{E}_t[\|g_2(y_t)\|^2]$ here is comparable but slightly worse (slightly larger) than Eq. (151).*

In summary, we still favor Eq. (148) as the construction of $g_2(y_t)$ due to its simplicity and smallest value of $\mathbb{E}_t[\|g_2(y_t)\|^2]$ among the ones we propose.

### C.3  Estimation of $D_t$ and proof of convergence of PARS using this estimator

In zeroth-order optimization, $D_t$ is not accessible since $D_t = \left(\frac{\nabla f(y_t)^\top p_t}{\|\nabla f(y_t)\|}\right)^2$. For the term inside the square, while the numerator can be obtained from the oracle, we do not have access to the denominator. Therefore, our task is to estimate $\|\nabla f(y_t)\|^2$.

To save queries, it is ideal to reuse the oracle query results used to obtain $v_t$ and $g_2(y_t)$: $\nabla f(y_t)^\top p_t$ and $\{\nabla f(y_t)^\top u_i\}_{i \in \{1,2,\dots,q\}}$. Again, we suppose $p_t, u_1, \cdots, u_q$ are obtained from the process in Section B.3.3. By Eq. (146), we have

$$\|\nabla f(y_t)\|^2 = (\nabla f(y_t)^\top p_t)^2 + \|\nabla f(y_t)_H\|^2. \tag{153}$$

Since $\{u_i\}_{i=1}^q$ is uniformly sampled from the $(d-1)$-dimensional space $H$,

**Proposition 11.** *For any $1 \le i \le q$, $\mathbb{E}_t[(\nabla f(y_t)^\top u_i)^2] = \frac{1}{d-1}\|\nabla f(y_t)_H\|^2$.*

*Proof.* By Proposition 10, $\mathbb{E}_t[\nabla f(y_t)^\top u_i \cdot u_i] = \frac{1}{d-1}\nabla f(y_t)_H$. Therefore,

$$\mathbb{E}_t[(\nabla f(y_t)^\top u_i)^2] = \nabla f(y_t)^\top \mathbb{E}_t[\nabla f(y_t)^\top u_i \cdot u_i] = \frac{1}{d-1}\nabla f(y_t)^\top \nabla f(y_t)_H \tag{154}$$

$$= \frac{1}{d-1}\|\nabla f(y_t)_H\|^2. \tag{155}$$

$\square$

Thus, we adopt the following unbiased estimate:

$$\|\nabla f(y_t)_H\|^2 \approx \frac{d-1}{q} \sum_{i=1}^q \left(\nabla f(y_t)^\top u_i\right)^2 \tag{156}$$

By Johnson-Lindenstrauss Lemma (see Lemma 5.3.2 in [15]), this approximation is rather accurate given a moderate value of $q$. Therefore, we have

$$\|\nabla f(y_t)\|^2 \approx \left(\nabla f(y_t)^\top p_t\right)^2 + \frac{d-1}{q} \sum_{i=1}^{q} \left(\nabla f(y_t)^\top u_i\right)^2 \tag{157}$$

and

$$D_t = \frac{(\nabla f(y_t)^\top p_t)^2}{\|\nabla f(y_t)\|^2} \approx \frac{\left(\nabla f(y_t)^\top p_t\right)^2}{\left(\nabla f(y_t)^\top p_t\right)^2 + \frac{d-1}{q} \sum_{i=1}^{q} \left(\nabla f(y_t)^\top u_i\right)^2}. \tag{158}$$

### C.3.1 PARS-Est algorithm with theoretical guarantee

In fact, the estimator of $D_t$ concentrates well around the true value of $D_t$ given a moderate value of $q$. To reach an algorithm with theoretical guarantee, we could adopt a conservative estimate of $D_t$, such that the constraint of $\theta_t$ in Line 3 of Algorithm 2 is satisfied with high probability. We show the prior-guided implementation of Algorithm 2 with estimation of $D_t$ in Algorithm 3, call it PARS-Est, and show that it admits a theoretical guarantee.

---

**Algorithm 3** Prior-guided ARS with a conservative estimator of $D_t$ (PARS-Est)

---

**Input:** $L$-smooth convex function $f$; initialization $x_0$; $\hat{L} \geq L$; Query count per iteration $q$; iteration number $T$; $\gamma_0 > 0$.
**Output:** $x_T$ as the approximate minimizer of $f$.
1:  $m_0 \leftarrow x_0$;
2:  **for** $t = 0$ to $T - 1$ **do**
3:      Obtain the prior $p_t$;
4:      Find a $\theta_t$ such that $\theta_t \leq \theta_t'$ in which $\theta_t'$ is defined in the following steps:
5:      Step 1: $y_t \leftarrow (1 - \alpha_t)x_t + \alpha_t m_t$, where $\alpha_t \geq 0$ is a positive root of the equation $\alpha_t^2 = \theta_t(1 - \alpha_t)\gamma_t$; $\gamma_{t+1} \leftarrow (1 - \alpha_t)\gamma_t$;
6:      Step 2: Sample an orthonormal set of $\{u_i\}_{i=1}^{q}$ in the subspace perpendicular to $p_t$ uniformly, as in Section B.3.3;
7:      Step 3: $\hat{D}_t \leftarrow \frac{\left(\nabla f(y_t)^\top p_t\right)^2}{\left(\nabla f(y_t)^\top p_t\right)^2 + \frac{2(d-1)}{q} \sum_{i=1}^{q}\left(\nabla f(y_t)^\top u_i\right)^2}$; $\theta_t' \leftarrow \frac{\hat{D}_t + \frac{q}{d-1}(1-\hat{D}_t)}{\hat{L}\left(\hat{D}_t + \frac{d-1}{q}(1-\hat{D}_t)\right)}$;
8:      Resample $\{u_i\}_{i=1}^{q}$ and calculate $v_t$ as in Section B.3.3;
9:      $g_1(y_t) \leftarrow \nabla f(y_t)^\top v_t \cdot v_t = \sum_{i=1}^{q} \nabla f(y_t)^\top u_i \cdot u_i + \nabla f(y_t)^\top p_t \cdot p_t$;
10:      $g_2(y_t) \leftarrow \frac{d-1}{q} \sum_{i=1}^{q} \nabla f(y_t)^\top u_i \cdot u_i + \nabla f(y_t)^\top p_t \cdot p_t$;
11:      $x_{t+1} \leftarrow y_t - \frac{1}{\hat{L}} g_1(y_t)$, $m_{t+1} \leftarrow m_t - \frac{\theta_t}{\alpha_t} g_2(y_t)$;
12:  **end for**
13:  **return** $x_T$.

---

**Theorem 6.** *Let*

$$p = \Pr\left(\sum_{i=1}^{q} x_i^2 < \frac{q}{2(d-1)}\right) \tag{159}$$

*where* $(x_1, x_2, ..., x_{d-1})^\top \sim \mathcal{U}(\mathbb{S}_{d-2})$, *i.e. follows a uniform distribution over the unit* $(d-1)$-*dimensional sphere. Then, in Algorithm 3, for any* $\delta \in (0, 1)$, *choosing a* $q$ *such that* $p \leq \frac{\delta}{T}$, *there exists an event* $M$ *such that* $\Pr(M) \geq 1 - \delta$, *and*

$$\mathbb{E}\left[(f(x_T) - f(x^*))\left(1 + \frac{\sqrt{\gamma_0}}{2} \sum_{t=0}^{T-1} \sqrt{\theta_t}\right)^2 \Bigg| M\right] \leq f(x_0) - f(x^*) + \frac{\gamma_0}{2}\|x_0 - x^*\|^2. \tag{160}$$

*Proof.* We first explain the definition of $\mathcal{F}_{t-1}$ in the proof (recall that $\mathbb{E}_t[\cdot]$ is $\mathbb{E}[\cdot|\mathcal{F}_{t-1}]$). Since in Theorem 5 we require $\theta_t$ to be $\mathcal{F}_{t-1}$-measurable, we let $\mathcal{F}_{t-1}$ also includes the randomness in Line 6 of Algorithm 3 in iteration $t$, besides randomness before iteration $t$ and randomness of $p_t$. We note that $\mathcal{F}_{t-1}$ does not include the randomness in Line 8.

Let $M$ be the event that: For each $t \in \{0, 1, ..., T-1\}$, $\frac{d-1}{q} \sum_{i=1}^{q} \left(\nabla f(y_t)^\top u_i\right)^2 \geq \frac{1}{2}\|\nabla f(y_t)_H\|^2$. When $M$ is true, we have that $\forall t$,

$$\hat{D}_t = \frac{\left(\nabla f(y_t)^\top p_t\right)^2}{\left(\nabla f(y_t)^\top p_t\right)^2 + \frac{2(d-1)}{q}\sum_{i=1}^{q}\left(\nabla f(y_t)^\top u_i\right)^2} \tag{161}$$

$$\leq \frac{\left(\nabla f(y_t)^\top p_t\right)^2}{\left(\nabla f(y_t)^\top p_t\right)^2 + \|\nabla f(y_t)_H\|^2} \tag{162}$$

$$= \frac{\left(\nabla f(y_t)^\top p_t\right)^2}{\|\nabla f(y_t)\|^2} = D_t. \tag{163}$$

Therefore,

$$\theta_t \leq \theta_t' = \frac{\hat{D}_t + \frac{q}{d-1}(1-\hat{D}_t)}{\hat{L}\left(\hat{D}_t + \frac{d-1}{q}(1-\hat{D}_t)\right)} \leq \frac{D_t + \frac{q}{d-1}(1-D_t)}{\hat{L}\left(D_t + \frac{d-1}{q}(1-D_t)\right)} = \frac{\mathbb{E}_t\left[\left(\nabla f(y_t)^\top v_t\right)^2\right]}{\hat{L} \cdot \mathbb{E}_t[\|g_2(y_t)\|^2]}. \tag{164}$$

Since $\mathcal{F}_{t-1}$ includes the randomness in Line 6 of Algorithm 3 in iteration $t$, $\mathbb{E}_t[\cdot]$ refer to only taking expectation w.r.t. the randomness of $v_t$ and $g_2(y_t)$ in iteration $t$, i.e. w.r.t. $\{u_1, \ldots, u_q\}$ in Line 8 of Algorithm 3. Since $\{u_1, \ldots, u_q\}$ in Line 8 is independent of $\{u_1, \ldots, u_q\}$ in Line 6, adding $\{u_1, \ldots, u_q\}$ in Line 6 to the history does not change the distribution of $\{u_1, \ldots, u_q\}$ in Line 8 given the history. Therefore according to the analysis in Section C.2, the last equality of Eq. (164) holds, and $\mathbb{E}_t[g_2(y_t)] = \nabla f(y_t)$. By Theorem 5, Eq. (160) is proved.

Next we give a lower bound of $\Pr(M)$. Let us fix $t$. Then

$$\Pr\left(\frac{d-1}{q}\sum_{i=1}^{q}\left(\nabla f(y_t)^\top u_i\right)^2 < \frac{1}{2}\|\nabla f(y_t)_H\|^2\right) = p.$$

Since for different $t$ the events inside the brackets are independent, by union bound we have $\Pr(M) \geq 1 - pT$. Since $p \leq \frac{\delta}{T}$, the proof is completed. $\square$

**Remark 13.** *To save queries, one may think that when constructing $v_t$ and $g_2(y_t)$, we could omit the procedure of resampling $\{u_i\}_{i=1}^{q}$ in Line 8, and reuse $\{u_i\}_{i=1}^{q}$ sampled in Line 6 to utilize the queries of relevant directional derivatives in Line 7. Our theoretical analysis does not support this yet, as explained below.*

*If we reuse $\{u_i\}_{i=1}^{q}$ sampled in Line 6 to construct $v_t$ and $g_2(y_t)$, then both $\theta_t$ and $\{g_2(y_t), v_t\}$ depend on the same set of $\{u_i\}_{i=1}^{q}$. Since Theorem 5 requires $\theta_t$ to be $\mathcal{F}_{t-1}$-measurable, we have to make $\mathcal{F}_{t-1}$ include randomness of this set of $\{u_i\}_{i=1}^{q}$. Then both $g_2(y_t)$ and $v_t$ are fixed given the history $\mathcal{F}_{t-1}$, which is not desired (e.g. $\mathbb{E}_t[g_2(y_t)] = \nabla f(y_t)$ generally does not hold since $\mathbb{E}_t[g_2(y_t)] = g_2(y_t)$ now, making the proof of Theorem 5 fail).*

**Remark 14.** *For given $d$ and $q$, $p$ can be calculated in closed form with the aid of softwares such as Mathematica. When $d = 2000$, if $q = 50$, then $p \approx 7.5 \times 10^{-4}$. If $q = 100$, then $p \approx 3.5 \times 10^{-6}$. Hence $p$ is rather small so that a moderate value of $q$ is enough to let $p \leq \frac{\delta}{T}$.*

*In fact, $p$ can be bounded by $O(\exp(-cq))$ by Johnson-Lindenstrauss Lemma where $c$ is an absolute constant (see Lemma 5.3.2 in [15]). Note that the bound is exponentially decayed w.r.t. $q$ and independent of $d$.*

**Remark 15.** *We give an analysis of the influence of the additional factor 2 in Line 7 of Algorithm 3. Let*

$$\hat{D}_{t2} = \frac{\left(\nabla f(y_t)^\top p_t\right)^2}{\left(\nabla f(y_t)^\top p_t\right)^2 + \frac{2(d-1)}{q} \sum_{i=1}^q \left(\nabla f(y_t)^\top u_i\right)^2},$$

$$\hat{D}_{t1} = \frac{\left(\nabla f(y_t)^\top p_t\right)^2}{\left(\nabla f(y_t)^\top p_t\right)^2 + \frac{d-1}{q} \sum_{i=1}^q \left(\nabla f(y_t)^\top u_i\right)^2},$$

$$\theta_{t2} = \frac{\hat{D}_{t2} + \frac{q}{d-1}(1 - \hat{D}_{t2})}{\hat{L}\left(\hat{D}_{t2} + \frac{d-1}{q}(1 - \hat{D}_{t2})\right)},$$

$$\theta_{t1} = \frac{\hat{D}_{t1} + \frac{q}{d-1}(1 - \hat{D}_{t1})}{\hat{L}\left(\hat{D}_{t1} + \frac{d-1}{q}(1 - \hat{D}_{t1})\right)}.$$

*Then $\hat{D}_{t1} \geq \hat{D}_{t2}$ and $1 - \hat{D}_{t1} \leq 1 - \hat{D}_{t2}$. We have*

$$\frac{\theta_{t2}}{\theta_{t1}} = \frac{\hat{D}_{t2} + \frac{q}{d-1}(1 - \hat{D}_{t2})}{\hat{D}_{t1} + \frac{q}{d-1}(1 - \hat{D}_{t1})} \cdot \frac{\hat{D}_{t1} + \frac{d-1}{q}(1 - \hat{D}_{t1})}{\hat{D}_{t2} + \frac{d-1}{q}(1 - \hat{D}_{t2})} \tag{165}$$

$$\geq \frac{\hat{D}_{t2}}{\hat{D}_{t1}} \cdot \frac{1 - \hat{D}_{t1}}{1 - \hat{D}_{t2}} \tag{166}$$

$$= \frac{\frac{\hat{D}_{t2}}{1 - \hat{D}_{t2}}}{\frac{\hat{D}_{t1}}{1 - \hat{D}_{t1}}} \tag{167}$$

$$= \frac{\frac{\left(\nabla f(y_t)^\top p_t\right)^2}{\frac{2(d-1)}{q} \sum_{i=1}^q (\nabla f(y_t)^\top u_i)^2}}{\frac{\left(\nabla f(y_t)^\top p_t\right)^2}{\frac{d-1}{q} \sum_{i=1}^q (\nabla f(y_t)^\top u_i)^2}} \tag{168}$$

$$= \frac{1}{2}. \tag{169}$$

*Therefore, we have $\theta_{t2} \geq \frac{1}{2}\theta_{t1}$.*

*Meanwhile, since $\hat{D}_{t2} \geq 0$, we have $\theta_{t2} \geq \frac{q^2}{\hat{L}(d-1)^2}$. Hence $\theta_{t2} \geq \max\left\{\frac{1}{2}\theta_{t1}, \frac{q^2}{\hat{L}(d-1)^2}\right\}$.*

### C.4 Approximate solution of $\theta_t$ and implementation of PARS in practice (PARS-Impl)

We note that in Line 3 of Algorithm 2, it is not straightforward to obtain an ideal solution of $\theta_t$, since $y_t$ depends on $\theta_t$. Theoretically speaking, $\theta_t > 0$ satisfying the inequality $\theta_t \leq \frac{\mathbb{E}_t\left[\left(\nabla f(y_t)^\top v_t\right)^2\right]}{\hat{L} \cdot \mathbb{E}_t[\|g_2(y_t)\|^2]}$ always exists, since by Eq. (152), $\frac{\mathbb{E}_t\left[\left(\nabla f(y_t)^\top v_t\right)^2\right]}{\hat{L} \cdot \mathbb{E}_t[\|g_2(y_t)\|^2]} \geq \frac{q^2}{\hat{L}(d-1)^2} := \theta$ always holds, so we can always let $\theta_t = \theta$. However, such estimate of $\theta_t$ is too conservative and does not benefit from a good prior (when $D_t$ is large). Therefore, one can guess a value of $D_t$, and then compute the value of $\theta_t$ by Eq. (152), and then estimate the value of $D_t$ and verify that $\theta_t \leq \frac{\mathbb{E}_t\left[\left(\nabla f(y_t)^\top v_t\right)^2\right]}{\hat{L} \cdot \mathbb{E}_t[\|g_2(y_t)\|^2]}$ holds. If it does not hold, we need to try a smaller $\theta_t$ until the inequality is satisfied. For example, in Algorithm 3, if we implement its Line 4 to Line 7 with a guessing procedure,[15] we could obtain an runnable algorithm with convergence guarantee. However, in practice such procedure could require multiple runs from Line 5 to Line 7 in Algorithm 3, which requires many additional queries; on the other hand, due to the additional factor 2 in Line 7 of Algorithm 3, we would always find a conservative estimate of $\theta_t$.

---

[15]For example, (1) compute $\theta_t'$ with $\theta_t \leftarrow 0$ by running Line 5 to Line 7; (2) we guess $\theta_t \leftarrow \kappa\theta_t'$ to compute a new $\theta_t'$ by rerunning Line 5 to Line 7, where $0 < \kappa < 1$ is a discount factor to obtain a more conservative estimate of $\theta_t$; (3) if $\theta_t \leq \theta_t'$, then we have found $\theta_t$ as required; else, we go to step (2).

In this section, we introduce the algorithm we use to find an approximate solution to find $\theta_t$ in Line 3 of Algorithm 2, which does not have theoretical guarantee but empirically performs well. The full algorithm PARS-Impl is shown in Algorithm 4. It stills follow the PARS framework (Algorithm 2), and our procedure to find $\theta_t$ is shown in Line 5 to Line 7.[16] Next we explain the procedure to find $\theta_t$ in detail.

---

**Algorithm 4** Prior-Guided Accelerated Random Search in implementation (PARS-Impl)

---

**Input:** $L$-smooth convex function $f$; initialization $x_0$; $\hat{L} \geq L$; Query count per iteration $q$ (cannot be too small); iteration number $T$; $\gamma_0 > 0$.
**Output:** $x_T$ as the approximate minimizer of $f$.
1: $m_0 \leftarrow x_0$;
2: $\|\hat{\nabla} f_{-1}\|^2 \leftarrow +\infty$;
3: **for** $t = 0$ to $T - 1$ **do**
4:  Obtain the prior $p_t$;
5:  $y_t^{(0)} \leftarrow x_t$; $\hat{D}_t \leftarrow \frac{(\nabla f(y_t^{(0)})^\top p_t)^2}{\|\hat{\nabla} f_{t-1}\|^2}$; $\theta_t \leftarrow \frac{\hat{D}_t + \frac{q}{d-1}(1-\hat{D}_t)}{\hat{L}\left(\hat{D}_t + \frac{d-1}{q}(1-\hat{D}_t)\right)}$;
6:  $y_t^{(1)} \leftarrow (1 - \alpha_t)x_t + \alpha_t m_t$, where $\alpha_t \geq 0$ is a positive root of the equation $\alpha_t^2 = \theta_t(1 - \alpha_t)\gamma_t$;
7:  $\hat{D}_t \leftarrow \frac{(\nabla f(y_t^{(1)})^\top p_t)^2}{\|\hat{\nabla} f_{t-1}\|^2}$; $\theta_t \leftarrow \frac{\hat{D}_t + \frac{q}{d-1}(1-\hat{D}_t)}{\hat{L}\left(\hat{D}_t + \frac{d-1}{q}(1-\hat{D}_t)\right)}$;
8:  $y_t \leftarrow (1 - \alpha_t)x_t + \alpha_t m_t$, where $\alpha_t \geq 0$ is a positive root of the equation $\alpha_t^2 = \theta_t(1 - \alpha_t)\gamma_t$; $\gamma_{t+1} \leftarrow (1 - \alpha_t)\gamma_t$;
9:  Sample an orthonormal set of $\{u_i\}_{i=1}^q$ in the subspace perpendicular to $p_t$ uniformly, as in Section B.3.3;
10:  $g_1(y_t) \leftarrow \sum_{i=1}^q \nabla f(y_t)^\top u_i \cdot u_i + \nabla f(y_t)^\top p_t \cdot p_t$;
11:  $g_2(y_t) \leftarrow \frac{d-1}{q} \sum_{i=1}^q \nabla f(y_t)^\top u_i \cdot u_i + \nabla f(y_t)^\top p_t \cdot p_t$;
12:  $\|\hat{\nabla} f_t\|^2 \leftarrow \left(\nabla f(y_t)^\top p_t\right)^2 + \frac{d-1}{q} \sum_{i=1}^q \left(\nabla f(y_t)^\top u_i\right)^2$;
13:  $x_{t+1} \leftarrow y_t - \frac{1}{\hat{L}}g_1(y_t)$, $m_{t+1} \leftarrow m_t - \frac{\theta_t}{\alpha_t}g_2(y_t)$;
14: **end for**
15: **return** $x_T$.

---

Specifically, we try to find an approximated solution of $\theta_t$ satisfying the equation $\theta_t = \frac{\mathbb{E}_t\left[\left(\nabla f(y_t)^\top v_t\right)^2\right]}{\hat{L} \cdot \mathbb{E}_t[\|g_2(y_t)\|^2]}$ to find a $\theta_t$ as large as possible and approximately satisfies the inequality $\theta_t \leq \frac{\mathbb{E}_t\left[\left(\nabla f(y_t)^\top v_t\right)^2\right]}{\hat{L} \cdot \mathbb{E}_t[\|g_2(y_t)\|^2]}$. Since $\frac{\mathbb{E}_t\left[\left(\nabla f(y_t)^\top v_t\right)^2\right]}{\hat{L} \cdot \mathbb{E}_t[\|g_2(y_t)\|^2]} = \frac{D_t + \frac{q}{d-1}(1-D_t)}{\hat{L}\left(D_t + \frac{d-1}{q}(1-D_t)\right)}$, we try to solve the equation

$$\theta_t = g(\theta_t) := \frac{D_t + \frac{q}{d-1}(1 - D_t)}{\hat{L}\left(D_t + \frac{d-1}{q}(1 - D_t)\right)}, \tag{170}$$

where $D_t = (\overline{\nabla f(y_t)}^\top p_t)^2$ and $y_t$ depends on $\theta_t$. This corresponds to finding the fixed-point of $g$, so we apply the fixed-point iteration method. Specifically, we first let $\theta_t = 0$, then $y_t = x_t$, and let $\theta_t \leftarrow g(\theta_t)$ (the above corresponding to Line 5 of Algorithm 4); then we calculate $y_t$ again using the new value of $\theta_t$ (corresponding to Line 6), and let $\theta_t \leftarrow g(\theta_t)$ (corresponding to Line 7). We find that two iterations are able to lead to satisfactory performance. Note that then two additional queries to the directional derivative oracle are required to obtain $\nabla f(y_t^{(0)})^\top p_t$ and $\nabla f(y_t^{(1)})^\top p_t$ used in Line 5 and Line 7.

Since $D_t = (\overline{\nabla f(y_t)}^\top p_t)^2 = \frac{(\nabla f(y_t)^\top p_t)^2}{\|\nabla f(y_t)\|^2}$, we need to estimate $\|\nabla f(y_t)\|^2$ as introduced in Section C.3. However, $y_t^{(0)}$ and $y_t^{(1)}$ in Algorithm 4 are different from both $y_t$ and $y_{t-1}$, so to estimate $\|\nabla f(y_t^{(0)})\|^2$ and $\|\nabla f(y_t^{(1)})\|^2$ as in Section C.3, many additional queries are required (since the query results of the directional derivative at $y_{t-1}$ or $y_t$ cannot be reused). Therefore, we introduce one additional approximation: we use the estimate of $\|\nabla f(y_{t-1})\|^2$ as the approximation of $\|\nabla f(y_t^{(0)})\|^2$ and $\|\nabla f(y_t^{(1)})\|^2$. Since the gradient norm itself is relatively large (compared with

---

[16]Line 5 and Line 7 require the query of $\nabla f(y_t^{(0)})^\top p_t$ and $\nabla f(y_t^{(1)})^\top p_t$ respectively, so each iteration of Algorithm 4 requires 2 additional queries to the directional derivative oracle, or requires 4 additional queries to the function value oracle using finite difference approximation of the directional derivative.

e.g. directional derivatives) and in zeroth-order optimization, the single-step update is relatively small, we expect that $\|\nabla f(y_t^{(0)})\|^2$ and $\|\nabla f(y_t^{(1)})\|^2$ are closed to $\|\nabla f(y_{t-1})\|^2$. In Algorithm 4, Line 12 estimates $\|\nabla f(y_t)\|^2$ by Eq. (157), and the estimator is denoted $\|\hat{\nabla} f_t\|^2$. Given this, $\|\nabla f(y_t^{(0)})\|^2$ and $\|\nabla f(y_t^{(1)})\|^2$ are approximated by $\|\hat{\nabla} f_{t-1}\|^2$ for calculations of $\hat{D}_t$ in Line 5 and Line 7 as approximations of $\overline{\left(\nabla f(y_t^{(0)})^\top p_t\right)^2}$ and $\overline{\left(\nabla f(y_t^{(1)})^\top p_t\right)^2}$.

Finally we note that in the experiments, we find that when using Algorithm 4, the error brought by approximation of $\|\nabla f(y_t^{(0)})\|^2$ and $\|\nabla f(y_t^{(1)})\|^2$ sometimes makes the performance of the algorithm not robust, especially when $q$ is small (e.g. $q = 10$), which could lead the algorithm to divergence. Therefore, we propose two tricks to suppress the influence of approximation error (we note that in practice, the second trick is more important, while the first trick is often not necessary given the application of the second trick):

- To reduce the variance of $\|\hat{\nabla} f_t\|$ when $q$ is small, we let

$$\|\hat{\nabla} f_t^{\mathrm{avg}}\|^2 = \frac{1}{k} \sum_{s=t-k+1}^{t} \|\hat{\nabla} f_s\|^2, \tag{171}$$

  and use $\|\hat{\nabla} f_{t-1}^{\mathrm{avg}}\|^2$ to replace $\|\hat{\nabla} f_{t-1}\|^2$ in Line 5 and Line 7. In our experiments we choose $k = 10$. Compared with $\|\hat{\nabla} f_{t-1}\|^2$, using $\|\hat{\nabla} f_{t-1}^{\mathrm{avg}}\|^2$ to estimate $\|\nabla f(y_t^{(0)})\|^2$ and $\|\nabla f(y_t^{(1)})\|^2$ could reduce the variance at the cost of increased bias. We note that the increased bias sometimes brings problems, so one should be careful when applying this trick.

- Although $D_t \leq 1$, It is possible that $\hat{D}_t$ in Line 5 and Line 7 is larger than 1, which could lead to a negative $\theta_t$. Therefore, a clipping of $\hat{D}_t$ is required. In our experiments, we observe that a $\hat{D}_t$ which is less than but very close to 1 (when caused by the accidental large approximation error) could also lead to instability of optimization, perhaps because that it leads to a too large value of $\theta_t$ used to determine $y_t$ and to update $m_t$. Therefore, we let $\hat{D}_t \leftarrow \min\{\hat{D}_t, B_{\mathrm{ub}}\}$ in Line 5 and Line 7 before calculating $\theta_t$, where $0 < B_{\mathrm{ub}} \leq 1$ is fixed. In our experiments we set $B_{\mathrm{ub}}$ to 0.6.

We leave a more systematic study of the approximation error as future work.

### C.5 Implementation of History-PARS in practice (History-PARS-Impl)

In PARS, when using a specific prior instead of the prior from a general source, we can utilize some properties of the prior. When using the historical prior ($p_t = v_{t-1}$), we find that $D_t$ is usually similar to $D_{t-1}$, and intuitively it happens when the smoothness of the objective function does not change quickly along the optimization trajectory. Therefore, the best value of $\theta_t$ should also be similar to the best value of $\theta_{t-1}$. Based on this observation, we can directly use $\theta_{t-1}$ as the value of $\theta_t$ in iteration $t$, and the value of $\theta_{t-1}$ is obtained with $y_{t-1}$ in iteration $t-1$. Following this thread, we present our implementation of History-PARS, i.e. History-PARS-Impl, in Algorithm 5.

### C.6 Full version of Algorithm 2 considering the strong convexity parameter and its convergence theorem

In fact, the ARS algorithm proposed in [12] requires knowledge of the strong convexity parameter $\tau$ of the objective function, and the original algorithm depends on $\tau$. The ARS algorithm has a convergence rate for general smooth convex functions, and also have another potentially better convergence rate if $\tau > 0$. In previous sections, for simplicity, we suppose $\tau = 0$ and illustrate the corresponding extension in Algorithm 2. In fact, for the general case $\tau \geq 0$, the original ARS can also be extended to allow for incorporation of prior information. We present the extension to ARS with $\tau \geq 0$ in Algorithm 6. Note that our modification is similar to that in Algorithm 2. For Algorithm 6, we can also provide its convergence guarantee as shown in Theorem 7. Note that after considering the strong convexity parameter in the algorithm, we have an additional convergence guarantee, i.e. Eq. (173). In the corresponding PARS algorithm, we have $\theta_t \geq \frac{q^2}{\hat{L}d^2}$, so the convergence rate of PARS is not worse than that of ARS and admits improvement given a good prior.

---

**Algorithm 5** History-PARS in implementation (History-PARS-Impl)

---

**Input:** $L$-smooth convex function $f$; initialization $x_0$; $\hat{L} \geq L$; Query count per iteration $q$ (cannot be too small); iteration number $T$; $\gamma_0 > 0$.
**Output:** $x_T$ as the approximate minimizer of $f$.
1: $m_0 \leftarrow x_0$;
2: $\theta_{-1} \leftarrow$ a very small positive number close to 0;
3: $v_{-1} \sim \mathcal{U}(\mathbb{S}_{d-1})$;
4: **for** $t = 0$ to $T - 1$ **do**
5:  $\quad y_t \leftarrow (1 - \alpha_t)x_t + \alpha_t m_t$, where $\alpha_t \geq 0$ is a positive root of the equation $\alpha_t^2 = \theta_{t-1}(1 - \alpha_t)\gamma_t$; $\gamma_{t+1} \leftarrow (1 - \alpha_t)\gamma_t$;
6:  $\quad$ Sample an orthonormal set $\{u_i\}_{i=1}^q$ in the subspace perpendicular to $v_{t-1}$, as in Section B.3.3 with $p_t = v_{t-1}$;
7:  $\quad g_1(y_t) \leftarrow \sum_{i=1}^q \nabla f(y_t)^\top u_i \cdot u_i + \nabla f(y_t)^\top v_{t-1} \cdot v_{t-1}; v_t \leftarrow \overline{g_1(y_t)}$;
8:  $\quad g_2(y_t) \leftarrow \frac{d-1}{q} \sum_{i=1}^q \nabla f(y_t)^\top u_i \cdot u_i + \nabla f(y_t)^\top v_{t-1} \cdot v_{t-1}$;
9:  $\quad \theta_t \leftarrow \frac{D_t + \frac{q}{d-1}(1 - D_t)}{\hat{L}\left(D_t + \frac{d-1}{q}(1 - D_t)\right)}$, where $D_t$ is estimated using Eq. (158) with $p_t = v_{t-1}$;
10: $\quad x_{t+1} \leftarrow y_t - \frac{1}{\hat{L}}g_1(y_t), m_{t+1} \leftarrow m_t - \frac{\theta_{t-1}}{\alpha_t}g_2(y_t)$;
11: **end for**
12: **return** $x_T$.

---

**Algorithm 6** Extended accelerated random search framework for $\tau \geq 0$

---

**Input:** $L$-smooth and $\tau$-strongly convex function $f$; initialization $x_0$; $\hat{L} \geq L$; $\hat{\tau}$ such that $0 \leq \hat{\tau} \leq \tau$; iteration number $T$; a positive $\gamma_0 \geq \hat{\tau}$.
**Output:** $x_T$ as the approximate minimizer of $f$.
1: $m_0 \leftarrow x_0$;
2: **for** $t = 0$ to $T - 1$ **do**
3:  $\quad$ Find a $\theta_t > 0$ such that $\theta_t \leq \frac{\mathbb{E}_t\left[(\nabla f(y_t)^\top v_t)^2\right]}{\hat{L} \cdot \mathbb{E}_t[\|g_2(y_t)\|^2]}$ where $\theta_t$, $y_t$ and $g_2(y_t)$ are defined in following steps:
4:  $\quad$ Step 1: $y_t \leftarrow (1 - \beta_t)x_t + \beta_t m_t$, where $\beta_t := \frac{\alpha_t \gamma_t}{\gamma_t + \alpha_t \hat{\tau}}$, $\alpha_t$ is a positive root of the equation $\alpha_t^2 = \theta_t((1 - \alpha_t)\gamma_t + \alpha_t \hat{\tau}); \gamma_{t+1} \leftarrow (1 - \alpha_t)\gamma_t + \alpha_t \hat{\tau}$;
5:  $\quad$ Step 2: Let $v_t$ be a random vector s.t. $\|v_t\| = 1$; $g_1(y_t) \leftarrow \nabla f(y_t)^\top v_t \cdot v_t$;
6:  $\quad$ Step 3: Let $g_2(y_t)$ be an unbiased estimator of $\nabla f(y_t)$, i.e. $\mathbb{E}_t[g_2(y_t)] = \nabla f(y_t)$;
7:  $\quad \lambda_t \leftarrow \frac{\alpha_t}{\gamma_{t+1}}\hat{\tau}$;
8:  $\quad x_{t+1} \leftarrow y_t - \frac{1}{\hat{L}}g_1(y_t), m_{t+1} \leftarrow (1 - \lambda_t)m_t + \lambda_t y_t - \frac{\theta_t}{\alpha_t}g_2(y_t)$;
9: **end for**
10: **return** $x_T$.

---

**Theorem 7.** *In Algorithm 6, if $\theta_t$ is $\mathcal{F}_{t-1}$-measurable, we have*

$$\mathbb{E}\left[(f(x_T) - f(x^*))\left(1 + \frac{\sqrt{\gamma_0}}{2}\sum_{t=0}^{T-1}\sqrt{\theta_t}\right)^2\right] \leq f(x_0) - f(x^*) + \frac{\gamma_0}{2}\|x_0 - x^*\|^2. \tag{172}$$

*and*

$$\mathbb{E}\left[(f(x_T) - f(x^*))\exp\left(\sqrt{\hat{\tau}}\sum_{t=0}^{T-1}\sqrt{\theta_t}\right)\right] \leq f(x_0) - f(x^*) + \frac{\gamma_0}{2}\|x_0 - x^*\|^2. \tag{173}$$

*Proof.* Let $L_e := \frac{\hat{L}}{2\hat{L}-L} \cdot \hat{L}$. We still have Eq. (18), so

$$\mathbb{E}_t[f(x_{t+1})] \leq f(y_t) - \frac{\mathbb{E}_t\left[(\nabla f(y_t)^\top v_t)^2\right]}{2L_e} \tag{174}$$

$$\leq f(y_t) - \frac{\mathbb{E}_t\left[(\nabla f(y_t)^\top v_t)^2\right]}{2\hat{L}}. \tag{175}$$

For an arbitrary fixed $x$, define $\rho_t(x) := \frac{\gamma_t}{2}\|m_t - x\|^2 + f(x_t) - f(x)$. Let $r_t := (1 - \lambda_t)m_t + \lambda_t y_t$. We first prove a lemma.

Since $(1 - \beta_t)x_t + \beta_t m_t = y_t = (1 - \beta_t)y_t + \beta_t y_t$, we have $m_t - y_t = \frac{1-\beta_t}{\beta_t}(y_t - x_t)$. So

$$r_t = (1 - \lambda_t)m_t + \lambda_t y_t = y_t + (1 - \lambda_t)(m_t - y_t) = y_t + (1 - \lambda_t)\frac{1 - \beta_t}{\beta_t}(y_t - x_t). \tag{176}$$

By $\beta_t = \frac{\alpha_t \gamma_t}{\gamma_t + \alpha_t \hat{\tau}}$, $\gamma_{t+1} = (1 - \alpha_t)\gamma_t + \alpha_t \hat{\tau}$ and $\lambda_t = \frac{\alpha_t}{\gamma_{t+1}}\hat{\tau}$, after eliminating $\gamma_t$ and $\gamma_{t+1}$, we have $(1 - \lambda_t)\frac{1-\beta_t}{\beta_t} = \frac{1-\alpha_t}{\alpha_t}$. Hence $r_t = y_t + \frac{1-\alpha_t}{\alpha_t}(y_t - x_t)$, which means

$$y_t = (1 - \alpha_t)x_t + \alpha_t r_t. \tag{177}$$

Now we start the main proof.

$$\rho_{t+1}(x) = \frac{\gamma_{t+1}}{2}\|m_{t+1} - x\|^2 + f(x_{t+1}) - f(x) \tag{178}$$

$$= \frac{\gamma_{t+1}}{2}\|r_t - x\|^2 - \frac{\gamma_{t+1}\theta_t}{\alpha_t}g_2(y_t)^\top(r_t - x) + \frac{\gamma_{t+1}\theta_t^2}{2\alpha_t^2}\|g_2(y_t)\|^2 + f(x_{t+1}) - f(x) \tag{179}$$

$$= \frac{\gamma_{t+1}}{2}\|r_t - x\|^2 - \alpha_t g_2(y_t)^\top(r_t - x) + \frac{\theta_t}{2}\|g_2(y_t)\|^2 + f(x_{t+1}) - f(x). \tag{180}$$

We make sure in Remark 2 that $\mathcal{F}_{t-1}$ always includes all the randomness before iteration $t$. Therefore, $\gamma_t$, $m_t$ and $x_t$ are $\mathcal{F}_{t-1}$-measurable. The assumption in Theorem 5 requires that $\theta_t$ is $\mathcal{F}_{t-1}$-measurable. Since $\alpha_t$, $\beta_t$, $y_t$ and $r_t$ are determined by $\gamma_t$, $x_t$, $m_t$ and $\theta_t$ (through Borel functions), they are also $\mathcal{F}_{t-1}$-measurable. Since $\theta_t$, $\alpha_t$ and $r_t$ are $\mathcal{F}_{t-1}$-measurable, we have $\mathbb{E}_t[\alpha_t g_2(y_t)^\top(r_t - x)] = \alpha_t \mathbb{E}_t[g_2(y_t)]^\top(r_t - x)$ and $\mathbb{E}_t[\theta_t\|g_2(y_t)\|^2] = \theta_t \mathbb{E}_t[\|g_2(y_t)\|^2]$. Hence

$$\mathbb{E}_t[\rho_{t+1}(x)] = \frac{\gamma_{t+1}}{2}\|r_t - x\|^2 - \alpha_t \mathbb{E}_t[g_2(y_t)]^\top(r_t - x) + \frac{\theta_t}{2}\mathbb{E}_t[\|g_2(y_t)\|^2] + \mathbb{E}_t[f(x_{t+1})] - f(x) \tag{181}$$

$$= \frac{\gamma_{t+1}}{2}\|r_t - x\|^2 - \alpha_t \nabla f(y_t)^\top(r_t - x) + \frac{\theta_t}{2}\mathbb{E}_t[\|g_2(y_t)\|^2] + \mathbb{E}_t[f(x_{t+1})] - f(x) \tag{182}$$

$$\leq \frac{\gamma_{t+1}}{2}\|r_t - x\|^2 - \alpha_t \nabla f(y_t)^\top(r_t - x) + \frac{\mathbb{E}_t\left[\left(\nabla f(y_t)^\top v_t\right)^2\right]}{2\hat{L}} + \mathbb{E}_t[f(x_{t+1})] - f(x) \tag{183}$$

$$\leq \frac{\gamma_{t+1}}{2}\|r_t - x\|^2 - \alpha_t \nabla f(y_t)^\top(r_t - x) + f(y_t) - f(x) \tag{184}$$

$$= \frac{\gamma_{t+1}}{2}\|r_t - x\|^2 - \nabla f(y_t)^\top(\alpha_t r_t - \alpha_t x) + f(y_t) - f(x) \tag{185}$$

$$= \frac{\gamma_{t+1}}{2}\|r_t - x\|^2 + \nabla f(y_t)^\top(-y_t + (1 - \alpha_t)x_t + \alpha_t x) + f(y_t) - f(x) \tag{186}$$

$$= \frac{\gamma_{t+1}}{2}\|r_t - x\|^2 + \alpha_t\left(f(y_t) + \nabla f(y_t)^\top(x - y_t)\right) \tag{187}$$

$$+ (1 - \alpha_t)\left(f(y_t) + \nabla f(y_t)^\top(x_t - y_t)\right) - f(x) \tag{188}$$

$$\leq \frac{\gamma_{t+1}}{2}\|r_t - x\|^2 + (1 - \alpha_t)f(x_t) - (1 - \alpha_t)f(x) - \frac{\alpha_t \tau}{2}\|x - y_t\|^2. \tag{189}$$

We also have

$$\frac{\gamma_{t+1}}{2}\|r_t - x\|^2 = \frac{\gamma_{t+1}}{2}\|(1 - \lambda_t)m_t + \lambda_t y_t - x\|^2 \tag{190}$$

$$= \frac{\gamma_{t+1}}{2}\|(1 - \lambda_t)(m_t - x) + \lambda_t(y_t - x)\|^2 \tag{191}$$

$$\leq \frac{\gamma_{t+1}(1 - \lambda_t)}{2}\|m_t - x\|^2 + \frac{\gamma_{t+1}\lambda_t}{2}\|y_t - x\|^2 \tag{192}$$

$$= \frac{\gamma_{t+1}(1 - \lambda_t)}{2}\|m_t - x\|^2 + \frac{\alpha_t \hat{\tau}}{2}\|y_t - x\|^2 \tag{193}$$

$$= (1 - \alpha_t)\frac{\gamma_t}{2}\|m_t - x\|^2 + \frac{\alpha_t \hat{\tau}}{2}\|x - y_t\|^2, \tag{194}$$

where the inequality is due to Jensen's inequality applied to the convex function $\|\cdot\|^2$, and the third equality is obtained after substituting $\lambda_t\gamma_{t+1} = \alpha_t\hat{\tau}$ by the definition of $\lambda_t$. Since $\gamma_{t+1} = (1-\alpha_t)\gamma_t + \alpha_t\hat{\tau} = (1-\alpha_t)\gamma_t + \lambda_t\gamma_{t+1}$, we have $\gamma_{t+1}(1-\lambda_t) = (1-\alpha_t)\gamma_t$, which leads to the last equality.

Hence

$$\mathbb{E}_t[\rho_{t+1}(x)] \leq \frac{\gamma_{t+1}}{2}\|r_t - x\|^2 + (1-\alpha_t)f(x_t) - (1-\alpha_t)f(x) - \frac{\alpha_t\tau}{2}\|x - y_t\|^2 \tag{195}$$

$$= (1-\alpha_t)\rho_t(x) + \frac{\alpha_t(\hat{\tau} - \tau)}{2}\|x - y_t\|^2 \tag{196}$$

$$\leq (1-\alpha_t)\rho_t(x). \tag{197}$$

Therefore,

$$\rho_0(x) = \mathbb{E}[\rho_0(x)] \geq \mathbb{E}\left[\frac{\mathbb{E}_0[\rho_1(x)]}{1-\alpha_0}\right] = \mathbb{E}\left[\mathbb{E}_0\left[\frac{\rho_1(x)}{1-\alpha_0}\right]\right] = \mathbb{E}\left[\frac{\rho_1(x)}{1-\alpha_0}\right]$$

$$\geq \mathbb{E}\left[\frac{\mathbb{E}_1[\rho_2(x)]}{(1-\alpha_0)(1-\alpha_1)}\right] = \mathbb{E}\left[\mathbb{E}_1\left[\frac{\rho_2(x)}{(1-\alpha_0)(1-\alpha_1)}\right]\right] = \mathbb{E}\left[\frac{\rho_2(x)}{(1-\alpha_0)(1-\alpha_1)}\right]$$

$$\geq \ldots$$

$$\geq \mathbb{E}\left[\frac{\rho_T(x)}{\prod_{t=0}^{T-1}(1-\alpha_t)}\right].$$

We have $\rho_T(x) \geq f(x_T) - f(x)$. To prove the theorem, let $x = x^*$. The remaining is to give an upper bound of $\prod_{t=0}^{T-1}(1-\alpha_t)$. Let $\psi_k := \prod_{t=0}^{k-1}(1-\alpha_t)$ and $a_k := \frac{1}{\sqrt{\psi_k}}$, we have

$$a_{k+1} - a_k = \frac{1}{\sqrt{\psi_{k+1}}} - \frac{1}{\sqrt{\psi_k}} = \frac{\sqrt{\psi_k} - \sqrt{\psi_{k+1}}}{\sqrt{\psi_k\psi_{k+1}}} = \frac{\psi_k - \psi_{k+1}}{\sqrt{\psi_k\psi_{k+1}}(\sqrt{\psi_k} + \sqrt{\psi_{k+1}})} \tag{198}$$

$$\geq \frac{\psi_k - \psi_{k+1}}{\sqrt{\psi_k\psi_{k+1}}(\sqrt{\psi_k})} \tag{199}$$

$$= \frac{\psi_k - (1-\alpha_k)\psi_k}{2\psi_k\sqrt{\psi_{k+1}}} = \frac{\alpha_k}{2\sqrt{\psi_{k+1}}} = \frac{\sqrt{\gamma_{k+1}\theta_k}}{2\sqrt{\psi_{k+1}}} = \frac{\sqrt{\theta_k}}{2}\sqrt{\frac{\gamma_{k+1}}{\psi_{k+1}}} \tag{200}$$

$$\geq \frac{\sqrt{\gamma_0\theta_k}}{2}. \tag{201}$$

The last step is because $\gamma_{t+1} \geq (1-\alpha_t)\gamma_t$, so $\frac{\gamma_{k+1}}{\gamma_0} \geq \prod_{t=0}^{k}(1-\alpha_t) = \psi_{k+1}$. Since $\psi_0 = 1$, $a_0 = 1$. Hence $a_T \geq 1 + \frac{\sqrt{\gamma_0}}{2}\sum_{t=0}^{T-1}\sqrt{\theta_t}$. Therefore,

$$\psi_T \leq \frac{1}{\left(1 + \frac{\sqrt{\gamma_0}}{2}\sum_{t=0}^{T-1}\sqrt{\theta_t}\right)^2}. \tag{202}$$

Meanwhile, since $\gamma_0 \geq \hat{\tau}$ and $\gamma_{t+1} = (1-\alpha_t)\gamma_t + \alpha_t\hat{\tau}$, we have that $\forall t, \gamma_t \geq \hat{\tau}$. Then $\alpha_t^2 = \theta_t((1-\alpha_t)\gamma_t + \alpha_t\hat{\tau}) \geq \theta_t\hat{\tau}$, then we have that $\alpha_t \geq \sqrt{\hat{\tau}\theta_t}$. Therefore,

$$\psi_T \leq \prod_{t=0}^{T-1}\left(1 - \sqrt{\hat{\tau}\theta_t}\right) \leq \exp\left(-\sqrt{\hat{\tau}}\sum_{t=0}^{T-1}\sqrt{\theta_t}\right). \tag{203}$$

The proof is completed. $\qquad\square$

# D   Supplemental materials for Section 5

## D.1   More experimental settings in Section 5.1

In experiments in this section, we set the step size $\mu$ used in finite differences (Eq. (1)) to $10^{-6}$, and the parameter $\gamma_0$ in ARS-based methods to $\hat{L}$.

### D.1.1 Experimental settings for Fig. 1

**Prior** We adopt the setting in Section 4.1 of [9] to mimic the case that the prior is a biased version of the true gradient. Specifically, we let $p_t = \overline{\nabla f(x_t) + (b + n_t)}$, where $b$ is a fixed vector and $n_t$ is a random vector uniformly sampled each iteration, $\|b\| = 1$ and $\|n_t\| = 1.5$.

**Test functions** Our test functions are as follows. We choose $f_1$ as the "worst-case smooth convex function" used to construct the lower bound complexity of first-order optimization, as used in [12]:

$$f_1(x) = \frac{1}{2}(x^{(1)})^2 + \frac{1}{2}\sum_{i=1}^{d-1}(x^{(i+1)} - x^{(i)})^2 + \frac{1}{2}(x^{(d)})^2 - x^{(1)}, \text{ where } x_0 = \mathbf{0}. \tag{204}$$

We choose $f_2$ as a simple smooth and strongly convex function with a worst-case initialization:

$$f_2(x) = \sum_{i=1}^{d}\left(\frac{i}{d} \cdot (x^{(i)})^2\right), \text{ where } x_0^{(1)} = d, x_0^{(i)} = 0 \text{ for } i \geq 2. \tag{205}$$

We choose $f_3$ as the Rosenbrock function ($f_8$ in [5]) which is a well-known non-convex function used to test the performance of optimization problems:

$$f_3(x) = \sum_{i=1}^{d-1}\left(100\left((x^{(i)})^2 - x^{(i+1)}\right)^2 + (x^{(i)} - 1)^2\right), \text{ where } x_0 = \mathbf{0}. \tag{206}$$

We note that ARS, PARS-Naive and PARS could depend on a strong convexity parameter (see Section C.6) when applied to a strongly convex function. Therefore, for $f_2$ we set this parameter to the ground truth value. For $f_1$ and $f_3$ we set it to zero, i.e. we use Algorithm 2.

### D.1.2 Experimental settings for Fig. 2

In this part we set $d = 500$ and set $q$ such that each iteration of each algorithm costs 11 queries. Since when using the historical prior, we aim to build algorithms agnostic to parameters of the objective function, we set the strong convexity parameter in ARS-based methods to 0 even though we know that e.g. $f_2$ is strongly convex. Correspondingly, we adopt adaptive restart of function scheme [13] to reach the ideal performance. We introduce our implementation here. In each iteration (suppose that currently it is iteration $t$) of Algorithm 5, we check whether $f(y_t) \leq f(y_{t-1})$. If not, we set $m_{t+1} \leftarrow x_{t+1}$ and $\gamma_{t+1} \leftarrow \gamma_0$ as the restart.

### D.2 More experimental settings in Section 5.2

We perform targeted attacks under the $\ell_2$ norm with the perturbation bound set to $3.514 (= 32/255 \times \sqrt{784})$ if each pixel value has the range $[0, 1]$. The objective function to maximize for attacking image $x$ is the C&W loss [3], i.e. $f(x) = Z(x)_t - \max_{i \neq t} Z(x)_i$, where $t$ is the target class and $Z(x)$ is the logits given the input $x$. The network architecture is from the PyTorch example (https://github.com/pytorch/examples/tree/master/mnist).

We set the step size $\mu$ used in finite differences (Eq. (1)) to $10^{-4}$, and the parameter $\gamma_0$ in ARS-based methods to $\hat{L}$. To deal with the constraints in optimization, in each iteration we perform projection after the update to ensure that the constraints are satisfied. In historical-prior-guided methods, to prevent the prior from pointing to the infeasible region (where the constraints are not satisfied), we let the prior $p_t$ be $\overline{x_t - x_{t-1}}$ for History-PRGF and $\overline{x_t - y_{t-1}}$ for History-PARS. In unconstrained optimization, this is equivalent to the original choice of $p_t$ ($p_t = \overline{g_{t-1}}$ for History-PRGF and $p_t = \overline{g_1(y_{t-1})}$ for History-PARS) up to sign. But in constrained optimization, since $x_t$ is further projected to the feasible region after the update from $x_{t-1}$ or $y_{t-1}$, they are not equivalent.

We note that the number of queries for each image does not count queries (one query per iteration) to check whether the attack has succeeded.

## E  Potential negative societal impacts

As a theoretical work, we think this paper can provide valuable insights on understanding existing algorithms and may inspire new algorithms for zeroth-order optimization, while having no significant

potential negative societal impacts. One may pay attention to its application to query-based black-box adversarial attacks.