# OpenReview forum: "On the Convergence of Prior-Guided Zeroth-Order Optimization Algorithms"
_NeurIPS.cc/2021/Conference — NeurIPS 2021 Poster_

### Official Review · Reviewer_xyD2 · 2021-07-07

**Rating:** 7
**Confidence:** 3

**Summary:**

This work analyzes the convergence rates of several existing prior guided zero order optimization methods. The zero order methods studied use statistical methods to provide an estimate of the function gradient using only function evaluations. This works studies the trade offs between gradient estimation accuracy and function evaluations as well as the effect of learning rate selection for the various estimates. This work also proposes a prior guided version of accelerated random search and studies its convergence. All algorithms studied are then numerically compared on benchmark functions as well as black box adversarial attacks.

**Limitations And Societal Impact:**

Limitations and negative social impact are adequately discussed.



**Main Review:**

The authors here focus on the effects of history dependent gradient estimates on zero order optimization.  The setting studied here is challenging because gradient estimates themselves are not statistically independent. This requires significant modifications to the standard convergence arguments and to the derived bounds that the authors clearly explain.  To the best of my knowledge, the technical arguments used here are novel.

The authors also try to explain the empirical success of history based gradient estimators. The authors prove that convergence can be accelerated by using previous gradient information when a suboptimal learning rate is used, which is in agreement with past empirical findings. I found this insight very helpful and relevant given that learning rate selection remains more art than science even for first order methods.

Finally, the analysis of the prior guided version of Accelerated Random Search method requires  even more work as additional quantities beyond directional derivatives need to be estimated. The experiments showcase that the prior guided algorithms can be more efficient than their simpler counterparts. I believe that this work both explains the success of existing works but can also have standalone theoretical and practical impact.

I acknowledge the authors' responses to all reviewers. I believe the additions proposed by the authors will make the submission stronger so I continue to be in favor of accepting this submission.

**Time Spent Reviewing:**

4

---

> ### Author Response · Authors · 2021-08-10
> **Response to Reviewer xyD2**
>
> Thank you for the appreciation of our novelty and technical contributions.

---

### Official Review · Reviewer_1ukB · 2021-07-15

**Rating:** 7
**Confidence:** 3

**Summary:**

This work studies the problem of zeroth-order optimization in a deterministic setting, where the exact function values at each point can be observed but the gradient of the function is not accessible. The main approach for solving such problems is first to estimate the gradient of the function and then apply a gradient method by plugging in this estimator instead of the gradient. The main goal of this work is to provide convergence guarantee for prior guided zeroth order algorithms.

Assumptions on the objective function: The objective function is (strongly) convex and smooth.

Approach: Since the paper assumes a deterministic setting, at each step by two function evaluations and choosing a small enough perturbation term, roughly they can observe the gradient of the function towards any arbitrary direction (page 3, equation 1). Also, they assume that at each step they can observe $q$ random directions of the gradient, which are given priorly (or they estimate them by $2q$ additional function evaluations). Then, they take advantage of these $q$ given directions and $g_{I}$s to construct $g_t$, where $g_t$ is the gradient estimator at step $t$, and $I<t$.

Contributions: Their results in Theorems 3 and 4 outline the convergence guarantees and robustness to the learning rate.

**Limitations And Societal Impact:**

The authors briefly discuss the limitations of their work. I also pointed out some, in the Main Review section.

Since the paper is quite technical, I except no direct social impact at this point.

**Main Review:**

Before the authors' responses, some parts were not clear to me. They provide useful justification and clear the misunderstanding. I edited the review and raised my score.

1. Theoretical claims are well supported.
2. The technical aspects are well written. Also, in the appendix, they tried to cover the most general cases, which was quite nice.
3. I encourage the authors to discuss the computational cost of Gram-Schmidt procedure diagonalization.
4. Their results provide intuition about the practical success of History based algorithms and outline the robustness of History-PRGF against the choices of learning rate.
5. The authors also provide convergence analysis for an accelerated greedy method, which is outlined in Theorem 5.

**Time Spent Reviewing:**

8 hours

---

> ### Author Response · Authors · 2021-08-10
> **Response to Reviewer 1ukB**
>
> Thank you for the valuable comments. Below, we address your main concerns in detail, especially clarifying some potential misunderstandings.
> We hope you might view this as sufficient reason to further raise your score.
>
> ### Q1: About the comment "The main goal of this paper is to use prior information on the gradient at each step to develop a more efficient gradient estimator":
>
> We'd like to clarify this potential misunderstanding. Although our analysis indeed inspired a new algorithm (i.e., prior-guided ARS), the main goal of this paper is on the convergence analysis of zeroth order optimization algorithms where gradient estimation uses prior information. Such analysis is lacking, leaving a gap between empirical success and theoretical understanding. We provide a first attempt with insightful results (please see Lines 39-45), as agreed by all the other reviewers (gEjB, c85q and xyD2). In our view, offering insights into and explanations of widely-used methods is just as important an aspect of science and engineering research as producing new methods.
>
> ### Q2. Effect of choosing different $q$ in Theorems 3 and 4:
>
> Thanks for this insightful comments. Indeed, we mainly consider the second scenario as you commented, and we agree that the key metric is the total number of function evaluations.
> We think that the usefulness of our propositions is not eliminated, as explained below:
>
> * The main purpose of Theorems 3 and 4 is not to show the role how $q$ plays. Instead, they show that History-PRGF admits a convergence guarantee which is robust to the learning rate (i.e., the r.h.s. of Eq. (6) and (7) do not depend on $\\hat{L}$; see Remarks 5 and 6). This is also mentioned by Reviewers c85q and xyD2. Therefore, these theorems are also meaningful even if we let $q=1$, but we deal with the cases $q>1$ to make the statements more general.
>
> * In fact, one iteration of PRGF requires $q+1$ (instead of $q$ as you said in the comments) queries to the directional derivative oracle due to one additional query related to the prior, as explained in Example 2 (Line 155, letting $k=1$). Therefore, if the iteration complexity is propositional to $1/q$, the total query complexity is proportional to $(q+1)/q$, which is indeed dependent on $q$. Therefore, in History-PRGF, under certain circumstances, choosing a moderate $q$ can in theory save nearly one half of the queries compared with choosing $q=1$. This is further confirmed in experiments: e.g., in Section 5.1 we adopt $q=10$, and when using History-PRGF (with $\\hat{L}=L$) for optimization of $f_2$, it requires ~96 queries to halve the error $f(x_{\\mathrm{current}})-f(x^*)$, but if $q=1$ it requires ~173 queries.
> We will clarify this in our final version.
>
> * Finally, our paper discusses the convergence properties of more general algorithms besides History-PRGF in Theorems 1, 2 and 5, which have standalone non-trivial insights besides Theorems 3 and 4 (e.g., inspiring the development of prior-guided ARS), as agreed by all the reviewers (gEjB, c85q and xyD2).
>
> ### Q3. Cost in orthogonalization:
>
> We agree that although the orthogonalization procedure does not require additional queries, it requires additional computation. Nevertheless, the extra computational burden is usually not intense, due to the following reasons:
>
> * We'd like to first clarify that the time complexity of a Gram-Schmidt procedure (or other orthogonalization algorithms) over $q$ vectors in $\\mathbb{R}^d$ is $O(q^2 d)$ (see e.g. [*1], pp. 24) instead of $O(qd^2)$ as you wrote. In practice $q$ is typically chosen to be a moderate value such that $q\\ll d$ (e.g., in our experiments $q=10$ or $20$, while $d = 256$, $500$, or $784$), and our theoretical results also allow $q\\ll d$.
> With a small $q$, the computational burden of orthogonalization is small compared to $O(q)$ function evaluations used to approximate the directional derivatives since the computational complexity of each function evaluation could be much larger than $O(d)$ (e.g. in black-box attacks ($d$ is the input dimension) or hyperparameter tuning). Similar discussion also appears in [14], pp. 349.
>
>     In our adversarial attack experiments, we found that for any $q$ satisfying $q\\leq d$ (i.e. allowing $q$ to be large), using the QR decomposition module to perform orthogonalization, the orthogonalization took less than 20% of the total time, which is acceptable.
>
> * When $q\\ll d$, since two uniformly random distributed high-dimensional vectors are nearly orthogonal, in practice we can omit orthogonalization of $\\{u_1,u_2,\\ldots,u_q\\}$. We observed that in all of our experiments, such change has little influence on the results.
>
> We will include the discussion in our final version.
>
>
> ========================================================
>
> [*1] http://www.seas.ucla.edu/~vandenbe/133A/lectures/qr.pdf

---

> > ### Comment · Reviewer_1ukB · 2021-08-21
> > **Updated review**
> >
> > I would like to thank the authors for their useful explanations. I edited the review and raised my score.

---

> > > ### Author Response · Authors · 2021-08-21
> > > **Thanks for the feedback**
> > >
> > > Thank you very much for the update. We highly appreciate that.

---

### Official Review · Reviewer_c85q · 2021-07-17

**Rating:** 7
**Confidence:** 3

**Summary:**

This paper studies the performance of zeroth order optimization algorithms where gradient estimation uses prior information. They provide a bound on the convergence analysis for such prior guided algorithms in terms of expected alignment of the gradient estimates to the true gradient. They use this framework to study historical priors and show that such a prior may counter the effects of using a sub-optimal learning rate. They also provide a new accelerated random search algorithm that incorporates prior information.

**Limitations And Societal Impact:**

The paper does address the limitation of using the directional gradient oracle.

**Main Review:**

This paper studies an important and interesting problem. Their insights into the performance of the historical prior are useful. The paper is clearly written and well organized.
My concerns with this paper are as follows:
- the difference between the bounds in the convex and strongly convex case are not well highlighted or discussed. Moreover, zeroth order algorithms are not typically applied to only convex functions so some discussion on possible extensions or the difficulty of should be included in this work.
- while the bounds are used to study historical priors, it would benefit to work to either study other examples of priors or to demonstrate why their bounds in terms of the expected correlation between the prior and the true gradient are useful for other priors. Can we expect that such bounds are easy to compute for other priors?

**Time Spent Reviewing:**

3 hours

---

> ### Author Response · Authors · 2021-08-10
> **Response to Reviewer c85q**
>
> Thank you for the appreciation of our contributions and the valuable comments. We address them in detail below and will further improve in the final version.
>
> ### Q1. Differences between the convex case and the strongly-convex case; Extension to non-convex case:
>
> Thank you for the suggestions. We will add related discussion on differences of results and proofs between Theorems 1 and 2 in our final version. The extension to non-convex case will be our future work, and our preliminary attempts of theoretical analysis face some difficulty regarding how to deal with dependency between $\\mathbb{E}_t[C_t]$ and $\\|\\nabla f(x_t)\\|^2$ as mentioned in Line 18, which is addressed better in the convex case. We will include the discussion in our final version.
>
> ### Q2. Analysis on other priors
>
> Thanks for the suggestion. While it is difficult to lower-bound the usefulness of general priors such as the transfer prior in adversarial attacks (the gradient of a surrogate model), when the prior is an approximate gradient (such cases appear in [15]), it may be possible to bound its cosine similarity to the true gradient. We leave this meaningful task as future work.

---

> > ### Comment · Reviewer_c85q · 2021-08-20
> > **Acknowledgement of Authors' Response**
> >
> > I have read the other reviews and the authors' response. I do think that their bounds and the resulting insight that historical priors may counter the effects of using a sub-optimal learning rate is useful. My only concern with this work is the applicability of the bounds to other priors. In particular, it seems challenging to compute the expected correlation between the prior and the true gradient for other priors.

---

> > > ### Author Response · Authors · 2021-08-20
> > > **Thanks for the Feedback on Our Response**
> > >
> > > Thank you very much for the feedback on our response and the further acknowledgement of our contributions. We'll include a discussion on priors in the final version.

---

### Official Review · Reviewer_gEjB · 2021-07-19

**Rating:** 7
**Confidence:** 3

**Summary:**

The paper provides a convergence analysis of a class of prior-guided zeroth-order (ZO) optimization algorithms and gives a prior-guided variant of the accelerated random search (ARS) algorithm.
Contribution:
1. Provides a complete convergence on some prior-guided zeroth-order optimization algorithms.
2. Provides a prior-guided variant of the ARS algorithm.

**Limitations And Societal Impact:**

Yes.

**Main Review:**

Strengths:
1. The presentation of the convergence analysis is very nice, very well-written, and easy to follow. It is shown through the analysis that a good prior is one with a larger correlation with the gradient which is intuitively true.
2. The prior-guided ARS is a good addition to the class of prior-guided ZO algorithms. In addition, the new algorithm performs well on some convex functions.

Weaknesses:
1. It seems the analysis is just writing down traditional analysis of ZO algorithms with some quantities waiting to be instantiated to specific forms for different algorithms. I do not really see any unexpected new insights distilled from the analysis.
2. The proposed prior-guided ARS seems does not to perform well on black-bok attacks which is a key application to ZO algorithms in machine learning. This limited the impact of the paper.



**Time Spent Reviewing:**

2

---

> ### Author Response · Authors · 2021-08-10
> **Response to Reviewer gEjB**
>
> Thank you for the appreciation of our contributions and the valuable comments. We address them in detail below and will further improve in the final version.
>
> ### Q1. Non-triviality of our analysis and insights:
>
> Our main motivation is to establish the relationship between convergence and gradient estimation given that previous related works only focus on gradient estimation. Indeed, the theoretical analysis is non-trivial.
>
> The main challenge is briefly explained in Lines 115-119, i.e., $\\mathbb{E}_t[C_t]$ is not independent of $\\|\\nabla f(x_t)\\|^2$ (note that in traditional analysis $\\mathbb{E}_t[C_t]$ is a constant instead of a random variable, e.g., $\\mathbb{E}_t[C_t]=q/d$ for the RGF baseline (see Line 150)). This requires novel technical arguments (as agreed by Reviewer xyD2) and leads to more sophisticated statements in Theorems 1 and 2: when applying these theorems to History-PRGF to reach Theorems 3 and 4, besides bounding $\\mathbb{E}[C_t]$, we need to consider concentration of $C_t$ in the proofs (see Remarks 13 and 14 in Appendix B.5), which give us technical insights.
>
> Moreover, in Section 4, Algorithm 2 (the extended ARS framework) is designed based on its proof. So the proof provides meaningful and inspiring insights.
>
> We will make it clearer in the final version.
>
> ### Q2. The performance of PARS on black-box attacks:
>
> Due to space limitation, we did not include the full results of ARS-based methods, and we list them here: the median numbers of query for ARS\_0.2, ARS\_0.1, PARS\_0.2, PARS\_0.1 and PARS\_0.05 are respectively 736, 1387, 485, 551 and 727 (PARS refers to History-PARS).
>
> We can see that ARS-based methods perform comparably to RGF-based ones: there is no significant improvement by switching RGF to ARS (and PRGF to PARS), in contrast to results in Section 5.1. The possible reasons are as follows: 1) The number of iterations until success of attack is too small to show the advantage of ARS; 2) the theoretical analysis of ARS is limited to the convex case, so its performance under nonconvex problems is not well understood.
>
> Nevertheless, we note that as desired, our proposed PARS accelerates over the ARS baseline.
>
> We leave more evaluation of ARS-based methods in adversarial attacks and further improvement of their performance as future work. We'll make this clearer in the final version.

---

> > ### Comment · Reviewer_gEjB · 2021-08-24
> > **Thanks for the reply, my concerns are addressed**
> >
> > Thanks for providing a detailed feedback. My concerns are addressed and I would raise my score.

---

> > > ### Author Response · Authors · 2021-08-25
> > > **Thanks for the feedback**
> > >
> > > Thank you very much for the update. We highly appreciate that.

---

### Decision · Program_Chairs · 2021-09-27

**Decision:**

Accept (Poster)

**Comment:**

While there were some initial concerns from the reviewers, these were all clearly resolved following the author rebuttal, with the reviewers in question acknowledging the depth and clarity of the responses.  The reviewers are now in agreement that this paper is a valuable addition to the literature on zero-order optimization, and is suitable for publication at NeurIPS.  I do not have any specific reviewer points to highlight here, but I do ask that the authors carefully consider all of the reviewer comments when forming the camera-ready version.